# Multivariate regression trees as an 'explainable machine learning' approach to explore relationships between hydroclimatic characteristics and agricultural and hydrological drought severity: Case of study Cesar River basin.

Ana Paez-Trujilo[1,2,3], Jeffer Cañon[3], Beatriz Hernandez[3], Gerald Corzo1, Dimitri Solomatine[1,2,4]

[1]IHE Delft Institute for Water Education, P.O. Box 3015, 2601 DA Delft, the Netherlands
[2]Delft University of Technology, Water Resources Section, P.O. Box 5048, 2600 GA Delft, the Netherlands
[3]Fundacion Natura Colombia, P.O. 111311, Carrera 21 No. 39–43 Bogotá D.C., Colombia
[4]Water Problems Institute of RAS, 119333, Gubkina 3, Moscow, Russia

*Correspondence to*: Ana M. Paez-Trujilo[1] (A.M.PaezTrujillo@tudelft.nl)

**Abstract**

The typical drivers of drought events are lower than normal precipitation and/or higher than normal evaporation. The region's characteristics may enhance or alleviate the severity of these events. Evaluating the combined effect of the multiple factors influencing droughts requires innovative approaches. This study applies hydrological modelling and a machine learning tool

to assess the relationship between hydroclimatic characteristics and the severity of agricultural and hydrological droughts. The Soil Water Assessment Tool (SWAT) is used for hydrological modelling. Model outputs, soil moisture and streamflow, are used to calculate two drought indices, namely the Soil Moisture Deficit Index and the Standardized Stream Flow Index. Then, drought indices are utilised to identify the agricultural and hydrological drought events during the analysis period, and the indices categories are employed to describe their severity. Finally, the Multivariate regression tree technique is applied to

assess the relationship between hydroclimatic characteristics and the severity of agricultural and hydrological droughts.

Our research indicates that multiple parameters influence the severity of agricultural and hydrological droughts in the Cesar River Basin. The upper part of the river valley is very susceptible to agricultural and hydrological drought. Precipitation

shortfalls and high potential evapotranspiration drive severe agricultural drought, whereas limited precipitation influences severe hydrological drought. In the middle part of the river, inadequate rainfall partitioning and an unbalanced water cycle that favours water loss through evapotranspiration and limits percolation cause severe agricultural and hydrological drought conditions. Finally, droughts are moderate in the basin's southern part (Zapatosa marsh and the Serrania del Perijá foothills). Moderate sensitivity to agricultural and hydrological droughts is related to the capacity of the subbasins to retain water, which

lowers evapotranspiration losses and promotes percolation. Results show that the presented methodology, combining

hydrological modelling and a machine learning tool, provides valuable information about the interplay between the hydroclimatic factors that influence drought severity in the Cesar River basin.

## 1 Introduction

Projections indicate that drought frequency, severity and duration are expected to increase globally in the twenty-first century
(UNDRR, 2021). Upcoming soil moisture drought scenarios predict statically significant large-scale drying, especially in scenarios with strong radiative forcing in Central America and tropical South America (Lu et al., 2019). A similar trend is predicted for hydrological drought severity, which is expected to increase by the end of the twenty-first century, with regional hotspots in central and western Europe and South America, where the frequency of hydrological drought may increase by more than 20 % (Prudhomme et al., 2014). The intensification of drought characteristics (in combination with other factors) could
force the migration of up to 216 million people by 2050 (The World Bank, 2021), increase wildfire risk and tree mortality, and negatively affect regional air quality, among other ecosystem impacts (Vicente-Serrano et al., 2020).

It is essential that we better understand drought drivers if we are to foster preparedness and resilience to projected drought events. Remarkable progress has been achieved in understanding drought propagation through the hydrological cycle (Van
Loon et al., 2012). Drought occurs due to climatic extremes, which may be enhanced or alleviated by region characteristics and anthropogenic influence (Hao et al., 2022; Seneviratne et al., 2012; Tijdeman et al., 2018). Typically, droughts are triggered by atmospheric circulation and weather systems that combine to cause lower-than-normal precipitation and/or higher-than-normalevaporation in a region (Destouni & Verrot, 2014; Sheffield & Wood, 2011a). Reduced precipitation leads to a decrease in soil moisture, causing agricultural drought. When soil moisture depletion is high, it is restored in the wet season,
thus reducing subsurface flow and groundwater recharge and giving rise to hydrological drought (Iglesias et al., 2018). Regional characteristics such as soil type, elevation, slope, vegetation cover, drainage networks, water bodies and groundwater systems play a relevant role in response to the climate anomalies that affect drought propagation and contribute to different levels of agricultural and hydrological drought (Sheffield & Wood, 2011a; Zhang et al., 2022). Equally important, human interventions in the hydrological cycle (e.g. reservoirs, water diversion, deforestation, over-pumping groundwater,
overgrazing, urbanisation) can reduce water supplies, triggering a drought situation or exacerbating a climate-driven drought (Rangecroft et al., 2019; Wang et al., 2021).

Drought planning also uses research progress on drought characterisation. Using drought indices is a widespread methodology for drought characterisation. (Zargar et al., 2011). Drought indices are computed numerical representations of drought severity
(Hao & Singh, 2015; Keyantash & Dracup, 2002). Severity refers to the drought strength, also described as the deficit degree (Cavus & Aksoy, 2020), soil moisture deficit in the case of agricultural droughts and streamflow deficit in the case of hydrological droughts. Generally, severity is divided into different categories (e.g. moderate, severe, extreme), providing a

qualitative assessment of the drought state in a region during a given period. Drought indices (and their categories) are crucial for tracking or anticipating drought-related damage and impacts (WMO & GWP, 2016).


Despite remarkable progress achieved in understanding the drought-generating process and drought characterisation, there is still a need for studies that assess the complex interplay between the different drivers of droughts and how their combined effect influences drought characteristics (e.g. duration, severity, intensity) (Valiya Veettil & Mishra, 2020). Previous studies focus on the influence of one driver (Margariti et al., 2019; Mastrotheodoros et al., 2020; Shah et al., 2021; Xu et al., 2019),

and some of the methodologies applied cannot adequately address the non-linear relationship between climate, basin processes and droughts characteristics (Peña-Gallardo et al., 2019; Saft et al., 2016; Van Loon, 2015).

We have found two studies employing machine learning to assess the non-linear relationship between climate and basin processes and droughts (Konapala & Mishra, 2020; Valiya Veettil & Mishra, 2020). The studies reported relevant findings on

the parameters driving droughts; however, the selected techniques showed a limitation for the drought analysis since they allow only one output variable. In both cases, it was necessary to apply the chosen technique multiple times to find the relationships between hydroclimatic parameters and the different categories of the evaluated drought characteristics. For example, Valiya Veettil et al. (2020) used a classification and regression tree (CART) to identify the variables influencing drought duration. CART allows one output variable; then, the authors applied the approach three times to evaluate the variables

influencing short-term, medium-term and long-term drought events. Meanwhile, Konapala et al. (2020) used a random forest (RF) algorithm to identify the climate and basin parameters influencing the characteristics (duration, frequency and intensity) of three different drought regimes (long duration and mild intensity, moderate duration and intensity, short duration and high intensity). As the core of RF is a decision tree that allows one output variable (in this case, each characteristic of each drought regime), the authors repeated the procedure nine times, one for each drought regime and characteristic.


The aforementioned research shows the potential of machine learning techniques for drought-related analysis; nevertheless, it also suggests that assessing the parameters driving drought characteristics requires techniques capable of simultaneously handling the different categories of drought characteristics. Commonly used in ecology to relate independent environmental conditions to populations of multiple species, the Multivariate Regression Tree (MVRT) arises as a suitable technique for this

purpose. MVRT is a constrained clustering technique that links explanatory variables to multiple response variables while maintaining the individual characteristics of the responses. Significantly, the technique does not assume a linear relationship between explanatory and response variables. Furthermore, it allows for the so-called "interpretable machine learning" algorithms that make decisions and predictions understandable to humans (Molnar, 2022). MVRT interpretably is a relevant attribute for drought researchers and planners since it allows them to identify the parameters influencing severe (or mild)

drought conditions.

To understand the relationship between the drivers of droughts and the individual categories of agricultural and hydrological droughts severity, this study employs a three-step methodology. The first is hydrological modelling. We used SWAT to simulate the hydroclimatic parameters required for analysing droughts and applying the MVRT approach. The second is the analysis of droughts. SWAT outputs, soil moisture and streamflow, are used to calculate the drought indices, i.e., the Soil Moisture Deficit Index (SMDI) and the Standardized Stream Flow Index (SSI). Drought indices are utilised to identify the agricultural and hydrological drought events in the analysis period. Then, we calculate the months for each drought severity category during the observed droughts. Finally, the MVRT approach is applied to assess the relationship between hydroclimatic characteristics (represented by the simulated parameters in each subbasin) and drought severity categories (represented by the total number of months for each drought severity category in each subbasin). The analyses for agricultural and hydrological droughts were conducted separately; thus, two MVRTs were obtained. A concrete application of this methodology is developed in the Cesar River basin (Colombia, South America).

## 2 Study location and methods

### 2.1 Case study

Figure 1 presents the Cesar River basin's location, topography (Fig. 1a) and land use (Fig. 1b). The basin is located between 72º53'W 74º04'W longitude and 10º52'00'N 7º41'00''N latitude (Colombia). It extends for an area of 22,312 km². The basin's topography defines three distinct climatic regions (Universidad del Atlantico, 2014). In the north is La Sierra Nevada de Santa Marta. This sector is characterised by steeply sloped mountains reaching up to 5,700 meters above sea level (masl). The temperature ranges from 3°C to 6°C, and the mean annual precipitation is 1,000 mm. In the east is La Serranía del Perijá. This mountainous area is an extension of the eastern branch of the Andes range. In this sector, the altitude ranges from 1,000 to 2,000 masl. The average temperature is 24°C, and the average annual precipitation varies from 1,000 mm to 2,000 mm. Lastly, the valley of the Cesar River and the Zapatosa marsh are in the west and south of the basin, respectively. The valley is characterised by flat topography and a complex system of marshes formed by the Cesar River floodplains and its confluence with the Magdalena River. The average temperature is 28°C, and the mean annual precipitation is 1,500 mm. The basin's annual rainfall pattern is bimodal. The dry season occurs from December to April, followed by a rainy season from April to May. From June to July, precipitation decreases, and the main rainfall events occur between August and November.

The predominant land use is pasture, followed by agriculture (Universidad del Atlantico, 2014). The primary land use in La Sierra Nevada foothills is pastures for cattle farming. In La Serranía del Perijá, the high-altitude areas are covered by forests in very good condition; at the lower altitudes, the principal land use is agriculture, particularly subsistence crops. The Cesar River valley's soils are rich in nutrients, providing favourable conditions for agriculture. The riverbanks are covered by forests with low tree density.

The Zapatosa marsh is recognised as one of the most important wetlands in the country, and considering the relevance of this ecosystem, it was declared a Ramsar site in 2018 (Ramsar sites are wetlands of international importance for containing rare or

130 unique wetland types or for their relevance in conserving biological diversity). Nevertheless, the region is threatened by high water demand of monocrops and the overexploitation of forest resources. In addition, climate change projections indicate that by 2070, the basin's temperature may increase by 2.7°C, and precipitation may reduce by 10 % compared to the reference period 1971-2000 (Universidad del Magdalena et al., 2017). Accordingly, multiple initiatives are oriented to improve water management and create resilience to hydroclimatic extremes (Ministerio de Ambiente y Desarrollo Sostenible (Colombia),

135 2015).

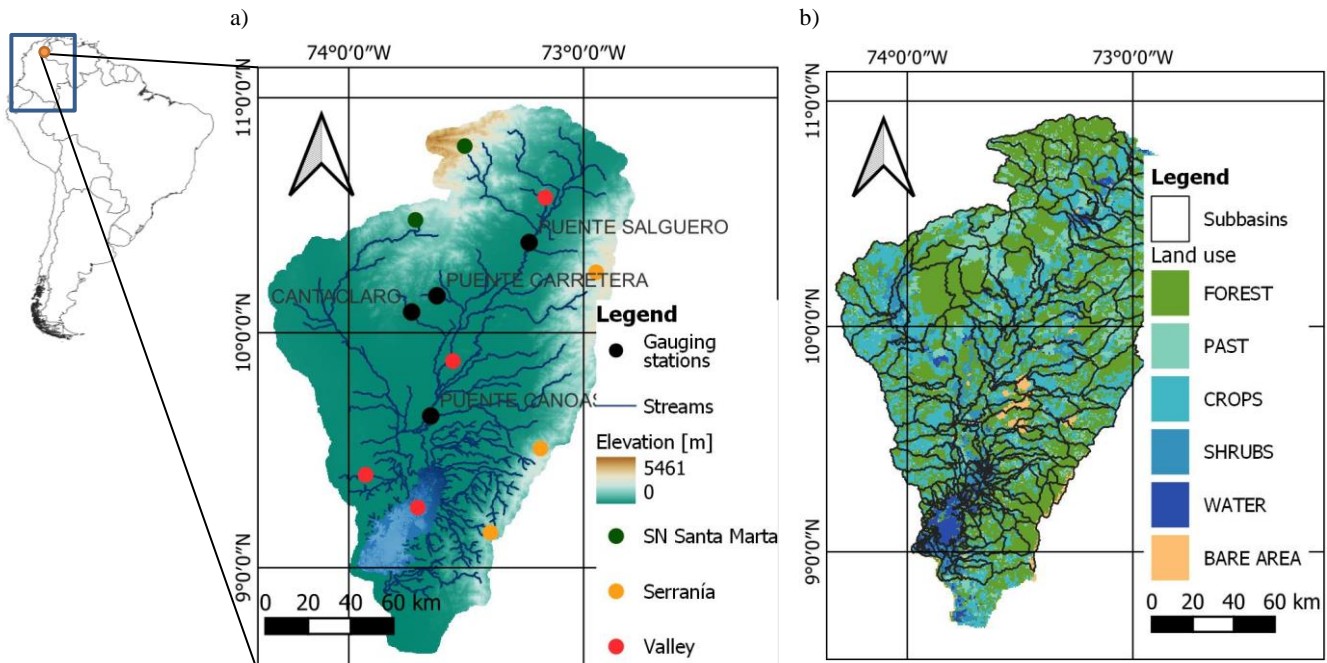

**Figure 1** Cesar River basin: a) topography and b) land use.

## 2.2 Methods

Figure 2 illustrates the three-step methodology applied in this study. Section 2.2.1 describes the hydrological modelling, and

140 2.2.2 the drought analysis. Section 2.2.3 presents the description of the MVRT technique.

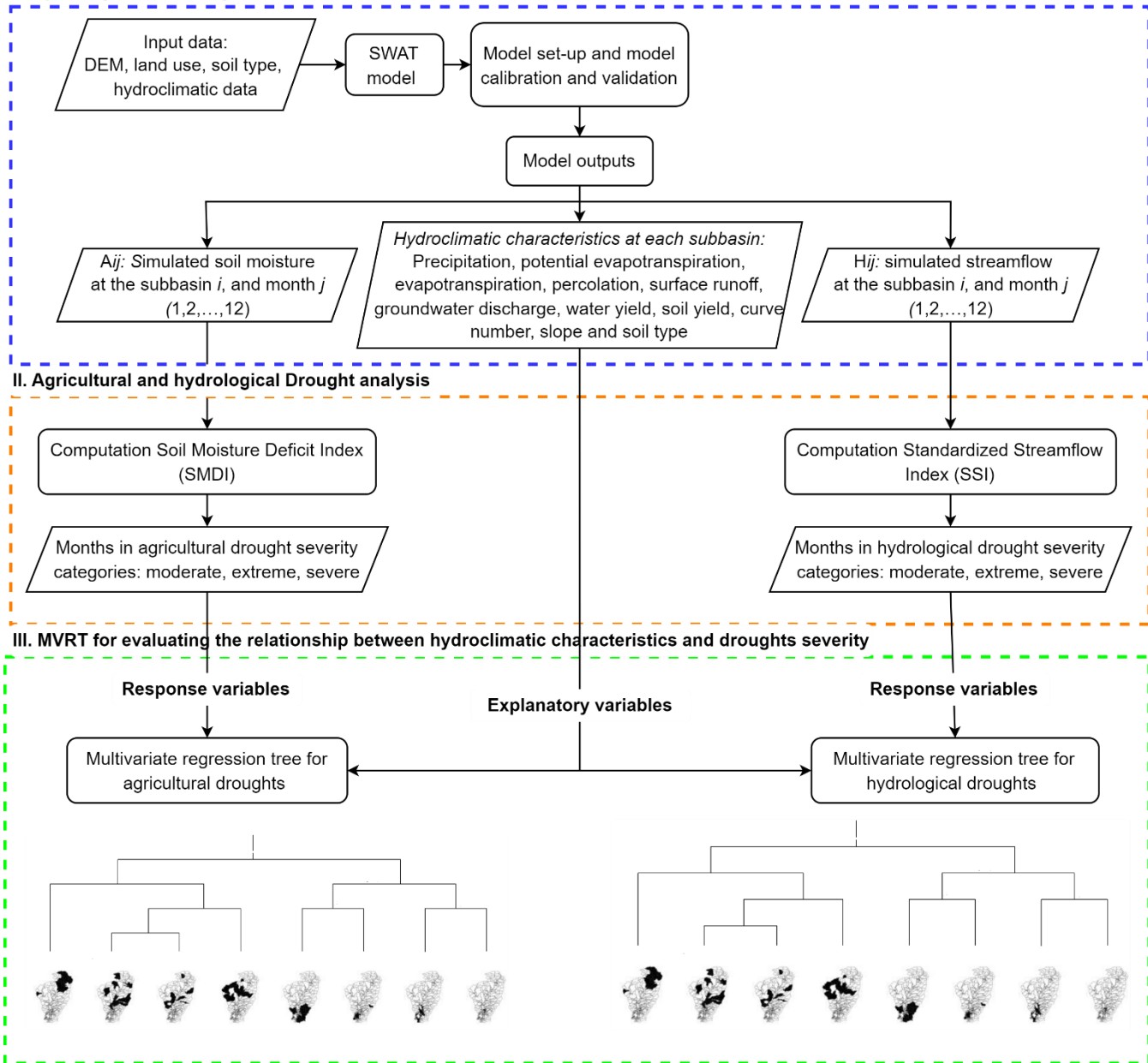

**Figure 2** Flow chart of the methodology.

### 2.2.1 Hydrological modelling

The SWAT model with an ArcSWAT extension was used to simulate the hydrological balance of the Cesar River. SWAT is a

145   continuous-time, semi-distributed and process-based river watershed scale model developed by The Agricultural Research Service of the United States Department of Agriculture (ARS-USDA). The model is designed to simulate the quality and

quantity of surface and groundwater and predict the environmental impacts of land management and climate change (Neitsch et al., 2011). SWAT divides the basin area up to the outlet point into several subbasins. Each subbasin is further split into multiple Hydrological Response Units (HRU), which are areas within the subbasin with common combinations of land cover, soil type and slope (Arnold et al., 2012).

**Model setup**

The model was built for the period from 1987 to 2018. The Cesar River basin was divided into 313 subbasins with a median area of 70 km$^2$. Four slope classes were set for the HRUs generation: flat (0–2%), gentle (2–10%), steep (10–35%) and considerably steep (>36%) (GEF et al., 2020, 2021). The following methods were used to model the principal hydrological processes: the soil conservation service-curve number (SCS-CN) was used to simulate surface runoff; potential evapotranspiration was estimated using the Hargreaves method; and water was routed through the channel network using the variable storage routing method. The details and sources of the SWAT model input data are presented in Table 1.

**Table 1.** SWAT model input data

| Data type | Details | Source |
|---|---|---|
| Digital elevation model | $25 \times 25$ m | Dataset ALOS PALSAR L1.0, Cartography 1:25000 Geographic Institute Agustín Codazzi (IGAC), Colombia |
| Soil map | $300 \times 300$ m | Soil profiles Project GEF Magdalena–Cauca VIVE, GEF, BID, Fundación Natura, Colombia |
| Land use map | $25 \times 25$ m | Land use map Geographic Institute Agustin Codazzi (IGAC), Colombia |
| Daily precipitation and daily minimum and maximum temperature | Period 1985–2018 (34 years) | Institute of Hydrology, Meteorology and Environmental Studies (IDEAM), Colombia |

**Model calibration and validation**

We used the SWAT-CUP software package with Sequential Uncertainty Fitting version 2 (SUFI-2) for automatic model calibration and validation. SUFI-2 operates by performing several iterations. The calibration parameters are sampled in each iteration using the Latin hypercube technique against the objective function values (Abbaspour et al., 2018).

Based on expert judgment and the available literature (Arnold et al., 2012; ASABE, 2017), the following SWAT parameters were used in the calibration and validation process: baseflow alpha factor (ALPHA_BF), effective hydraulic conductivity in main channel alluvium (CH_K), Manning's value for the main channel (CH_N2), SCS runoff curve number for moisture condition II (CN2), soil evaporation compensation factor (ESCO), groundwater delay (GW_DELAY), threshold depth of water in the shallow aquifer required for return flow to occur (GWQMN), deep aquifer percolation fraction (RCHRG_DP), threshold

depth of water in the shallow aquifer for percolation to the deep aquifer to occur (REVAPMN) and available water capacity of the soil layer (SOL_AWC). In the calibration process, a physically meaningful range is set for each parameter in each iteration. Then, a new parameter value (within the range) is selected and applied at each HRU or subbasin.

The model was calibrated from 1985 to 2002 and validated from 2003 to 2018 using the streamflow series from four stream
gauges (Fig. 1a). The source of the discharge data is the Institute of Hydrology, Meteorology and Environmental Studies (IDEAM), Colombia. The first two years were used as a warming-up period in both cases. Thus, performance indicators were calculated for 1987 to 2002 (calibration) and 2005 to 2018 (validation). The model's performance for simulating streamflow was evaluated using the Nash-Sutcliffe Efficiency (NSE) and percent bias (PBIAS), represented by Eq. 1 and Eq. 2, respectively:


$$NSE = 1 - \frac{\sum_{i=1}^{N}(O_i - P_i)^2}{\sum_{i=1}^{N}(O_i - \bar{O})^2} \qquad\qquad 1$$

$$PBIAS = \frac{\sum_{i=1}^{N}(O_i - P_i) \times 100}{\sum_{i=1}^{N} O_i} \qquad\qquad 2$$

where $O_i$ is the observed data, $P_i$ the predicted data, $\bar{O}$ the mean of the observed data and $N$ the number of observations during the simulation period.

The NSE is a dimensionless indicator ranging from -∞ to 1, with 1 representing a perfect match between the observed and
simulated values (Moriasi et al., 2007). The PBIAS measures the average tendency of the simulated values to be larger or smaller than the observed values. The ideal PBIAS is 0, with low-magnitude values indicating accurate model simulation (Moriasi et al., 2007).

### 2.2.2 Agricultural and hydrological drought analysis

The present study used the soil moisture deficit index (SMDI) to analyse agricultural droughts. We chose this index since it
was developed to use simulated soil moisture as the input parameter (Narasimhan & Srinivasan, 2005). SWAT calculates the soil water content of the entire soil profile. Three soil layers were identified in the Cesar River basin. The first layer thickness (vertical distance from the surface) reaches up to 350 mm, the second 1000 mm, and the third 1500 mm.

The computation procedure to determine the soil moisture deficit used the long-term soil moisture characteristics and the soil
moisture conditions during the drought period. The indicator was scaled between -4 and 4 to allow the spatial comparison of the drought index, regardless of climatic characteristics (Narasimhan & Srinivasan, 2005). Negative values of SMDI indicate

dry periods, while positive values indicate wet periods (compared to the region's normal conditions). Per the SMDI, agricultural drought severity was divided into three categories: moderate drought (SMDI -2.0 to -2.99), severe drought (SMDI -3.0 to -3.99) and extreme drought (SMDI -4). The following procedure was applied to compute the SMDI at each subbasin:


$$SD_{ij} = \frac{SW_{ij} - MSW_j}{MSW_j - minSW_j} \times 100, \qquad if \ SW_{ij} \leq MSW_j \qquad\qquad 3$$

$$SD_{ij} = \frac{SW_{ij} - MSW_j}{maxSW_j - MSW_j} \times 100, \qquad if \ SW_{ij} > MSW_j \qquad\qquad 4$$

where $SD_j$ is the soil moisture deficit (%), $SW_j$ is the monthly soil water available in the soil profile (mm) and $MSW_j$ is the long-term median available soil water in the soil profile (mm), $maxSW_j$ and $minSW_j$ are long-term median, maximum and minimum soil water available in the soil profile (mm), respectively, ($i = 1987 - 2018 \ and \ j = 1 - 12$).


The $SMDI_j$ of any given month was calculated using Eq. 5:

$$SMDI_j = 0.5 \times SMDI_{j-1} + \frac{SD_j}{50} \qquad\qquad 5$$

where $SMDI_{j-1}$ is the SMDI from the previous month.


SMDI was not calculated for the subbasins that correspond to the Zapatosa marsh. In these subbasins, the predominant land cover is water. See Fig. 5.

We used a standardised streamflow index (SSI) to represent hydrological droughts. The indicator was introduced by Modarres
(2007) and further investigated by Vicente-Serrano et al. (2011). The index is statically analogous to the commonly used standardised precipitation index (SPI) introduced by Mckee et al. (1993). SSI values mainly range from -2.0 (extremely dry) to 2.0 (extremely wet), and hydrological drought severity is divided into three categories: moderate drought (SSI -1.0 to -1.49), severe drought (SSI -1.5 to -1.99) and extreme drought (SSI -2.0 or less). The procedure to calculate SSI consists of converting streamflow values to standardised anomalies (i.e. z-scores). To this aim, the monthly simulated streamflow at each subbasin
in the analysis period (1987 to 2018) was fitted to the gamma probability distribution function.

SMDI and SSI were calculated monthly using the simulated soil water and streamflow values at each subbasin. The drought events during the period of analysis were then identified. A drought (agricultural or hydrological) event was assumed to occur in the basin when a number of subbasins (covering at least 30 % of the basin's total area) were in a drought state (moderate,

severe or extreme) for at least two consecutive time steps (i.e. in this study month). According to the spatial and temporal thresholds, a drought event began when both conditions were met and continued until one of them failed to be met. It is worth highlighting that the minimal extension of a drought is not defined, but it is accepted that droughts typically occur on a large scale (Sheffield & Wood, 2011b). Setting a spatial threshold is a common practice to maintain a minimum drought-affected and prevent identifying isolated areas experiencing dry spells as drought events (Brunner et al., 2021). The temporal threshold,
it was used to avoid including short-term droughts (i.e., daily or weekly) in the analysis (Li et al., 2020).

### 2.2.3 Multivariate regression tree approach for evaluating the relationships between hydroclimatic characteristics and droughts severity

MVRT is an extension of the popular regression tree (Breiman, 2001), but it differs in that it allows for multiple outputs (see De'ath (2002)). It recursively splits a quantitative response variable (predictand, output) controlled by a set of numerical or
categorical explanatory variables (predictors, input). The approach yields a set of non-linear models, each a piece-wise linear regression model (of zero order). An MVRT result is a tree whose terminal groups (leaves) of instances (input-output vectors) comprise subsets of samples selected to minimise the within-group sums of squares. Each successive split is given by a threshold value of the explanatory variables (Borcard et al., 2018). MVRT is applied to dataset exploration, description and prediction (De'ath, 2002). In this study, the explanatory variables are the hydroclimatic parameters at each subbasin,
represented by the average value of each parameter during the analysis period (1987 to 2018). The multivariate response is the number of months observed in the three drought severity categories (moderate, severe and extreme) at each subbasin. The analyses for agricultural and hydrological droughts were conducted separately; thus, two MVRTs were obtained.

The following MVRT attributes are relevant for this study. First, MVRT can capture the non-linear interactions between the
parameters influencing droughts and their severity. Second, the technique can handle numerical and categorical hydroclimatic parameters influencing drought severity (explanatory variables). Third, MVRT's capability to handle multiple outputs allowed us to evaluate the influence of the hydroclimatic parameters on moderate, severe and extreme drought conditions simultaneously (response variables). Simultaneous analysis of different drought categories provides a comprehensive understanding of the drought-generating process and the factors influencing severe (or mild) drought conditions. Fourth,
MVRT results can be easily visualised and interpreted. The resulting tree structure provides a clear representation of the relationship between the drivers of droughts and the severity of agricultural and hydrological droughts.

For building the MVRT, R software was used, namely, package mvpart. Before the analysis, the sets of explanatory and response variables were transformed to compare the descriptors measured in different units and to modify the variables'
weights. The matrix of explanatory variables was standardised to a mean of zero and a standard deviation of one. The matrix of response variables was standardised by the column maximum, then again by the row total (Wisconsin double standardisation).

**Datasets**

**Set of explanatory variables**

To select the set of explanatory variables, we used the outcomes of previous studies on governing drivers of droughts (Sheffield & Wood, 2011a; Zhang et al., 2022). Table 2 describes the eleven parameters selected as the potential drivers of droughts. The used values correspond to the parameters' average in the analysis period (1987 to 2018). The averages were computed using the SWAT model results at each subbasin. We used the dominant category at each subbasin for the curve number, the slope, and the soil type (categorical variables).


**Table 2.** Explanatory variables used in MVRT

| Hydroclimatic parameter | Abbreviation | Unit | Definition |
|---|---|---|---|
| Precipitation | PRECP | mm | Average precipitation at each subbasin |
| Potential evapotranspiration | PET | mm | Average potential evapotranspiration at each subbasin |
| Evapotranspiration | ET | mm | Average actual evapotranspiration at each subbasin |
| Percolation | PERC | mm | Average percolation past the root zone |
| Surface runoff | SURFQ | mm | Average surface contribution to the streamflow at each subbasin |
| Groundwater | GRWQ | mm | Average groundwater contribution to the streamflow at each subbasin |
| Water yield | WYLD | mm | Average amount of water that leaves the subbasin and contributes to the streamflow at each subbasin |
| Sediment yield | SYLD | metric tons/ha | Average sediment from the subbasin transported into the reach |
| Curve number | CN | – | Dominant curve number at each subbasin |
| Slope | SLP | – | Dominant slope at each subbasin |
| Hydrologic soil group | STY | – | Dominant hydrologic soil group (A, B, C, and D) at each subbasin. The soil hydrologic groups refer to the soil's infiltration characteristics. Properties of each soil type can be found in USDA (2007) |

**Set of response variables**

We used the drought analysis outcomes to define the response variables (Table 3). Following the methodology presented in 2.2.2, we identified the agricultural and hydrological drought events during the analysed period. After identifying the drought

events, we counted the months for each drought severity category at each subbasin. The observed months for each one of the three drought categories were used as response variables. The analyses for agricultural and hydrological droughts were conducted separately; thus, two sets of response variables were obtained.

**Table 3.** Response variables used in MVRT

| Drought category | Abbreviation | Unit | Definition |
|---|---|---|---|
| Moderate agricultural/hydrological drought | MOD | month | Number of months in the moderate agricultural drought category during the drought events identified in the simulation period at each subbasin |
| Severe agricultural/hydrological drought | SEV | month | Number of months in the severe agricultural drought category during the drought events identified in the simulation period at each subbasin |
| Extreme agricultural/hydrological drought | EXT | month | Number of months in the extreme agricultural drought category during the drought events identified in the simulation period at each subbasin |

### Building the MVRT: Constrained partitioning of the data and cross-validation

Building the MVRT consisted of two processes: (1) the constrained partitioning of the data and (2) the cross-validation of the results. The mvpart package runs both processes in parallel. The two procedures are briefly explained below, and a more detailed description can be found in Borcard et al. (2018).

The data partitioning consisted of three steps. First, for each explanatory variable were generated all possible partitions of the sites (subbasins) into two groups. Second, for each partition, it was calculated the resulting sum of within-group sums of squared distances to the group means for the response data (within-group SS). Within-group SS is equivalent to standard deviation. Lastly, the partition into two groups to minimise the within-group SS and the threshold value/level of the explanatory variable was retained. These steps were repeated within the two previously established subgroups until all the objects formed their own groups. For each tree that was computed, the relative error was calculated as the sum of the within-group SS of all leaves divided by the overall SS of the data. This procedure for MVRT is equivalent to the one originally proposed by Breiman (2001) for his regression tree technique.

A cross–validation procedure was used to prune the tree and identify the optimal tree size (Kuhn & Johnson, 2013; Legendre & Legendre, 2012). The cross-validation procedure was performed automatically using mvpart. Per this procedure, the data was randomly divided into roughly equal-sized test groups. Each test group was held out in turn while the tree was fitted using the remaining groups. The distances between the centroids of the objects at tree leaves and each object of the test group were then calculated. Finally, the objects of the test group were allocated to the closest leaf of the constructed tree. An overall relative error statistic (relative cross-validation error, CVRE) was calculated for each group using all $n$ objects, per Eq. 6:

$$CVRE_{(k)} = \frac{\sum_{i=1}^{n} \sum_{j=1}^{p} (y_{ij(k)} - \hat{y}_{j(k)})^2}{\sum_{i=1}^{n} \sum_{j=1}^{p} (y_{ij} - \bar{y}_j)^2}$$

where $y_{ij(k)}$ is the value of variable $j$ for object $i$ belonging to test group $k$, $\hat{y}_{j(k)}$ is the value of that same variable at the centroid of the leaf closest to object $i$, and the denominator is the overall sum of squares of the response data.

This cross-validation process was repeated several times for consecutive and independent divisions of the data into test groups. For each group, the mean and standard deviation of all CVRE were computed. The CVRE varied from 0 for perfect predictors to close to 1 for poor predictors (as error increases, CVRE increases indefinitely). Among the mvpart function arguments, we used ten cross-validation groups (function argument, xval = 10) and 100 iterations (function argument xmult = 100). The tree was selected using interactive cross-validation (function argument xv = 'pick').


To choose the size of the tree that retained the most descriptive partition, we used the approach suggested by De'ath (2002). According to the author the tree with the smallest CVRE offers the best combination of explanatory power and interpretability. Once the tree was built, the proportion of explained variance (EV) was calculated as $1 - RE_{tree}$ (tree relative error) (Cannon, 2012).

**3 Results**

**3.1 SWAT model calibration and validation**

Table 4 summarises the calibration and validation performance indicators for the SWAT model at each gauging station. The calibration and validation models simulated monthly stream flows with NSE values equal to or greater than 0.50 and relatively low PBIAS values (GEF et al., 2020, 2021). According to the performance ratings for calibrating and validating hydrological

models, NSE and PBIAS values indicated that the model was appropriate for simulating streamflow (Moriasi et al., 2007). Figure 3 presents the model hydrographs at each gauging station for the calibration and validation periods. The locations of the stations can be found in Fig 1.

**Table 4.** SWAT model performance simulating streamflow

| Gauging station | Calibration | | Validation | |
|---|---|---|---|---|
| | NSE | PBIAS [%] | NSE | PBIAS [%] |
| Puente Salguero | 0.61 | 4.28 | 0.52 | -8.3 |
| Puente Carretera | 0.50 | -5.34 | 0.52 | 7.6 |
| Cantaclaro | 0.58 | -11.30 | 0.50 | -11.7 |
| Puente Canoas | 0.70 | -1.34 | 0.57 | 10.64 |


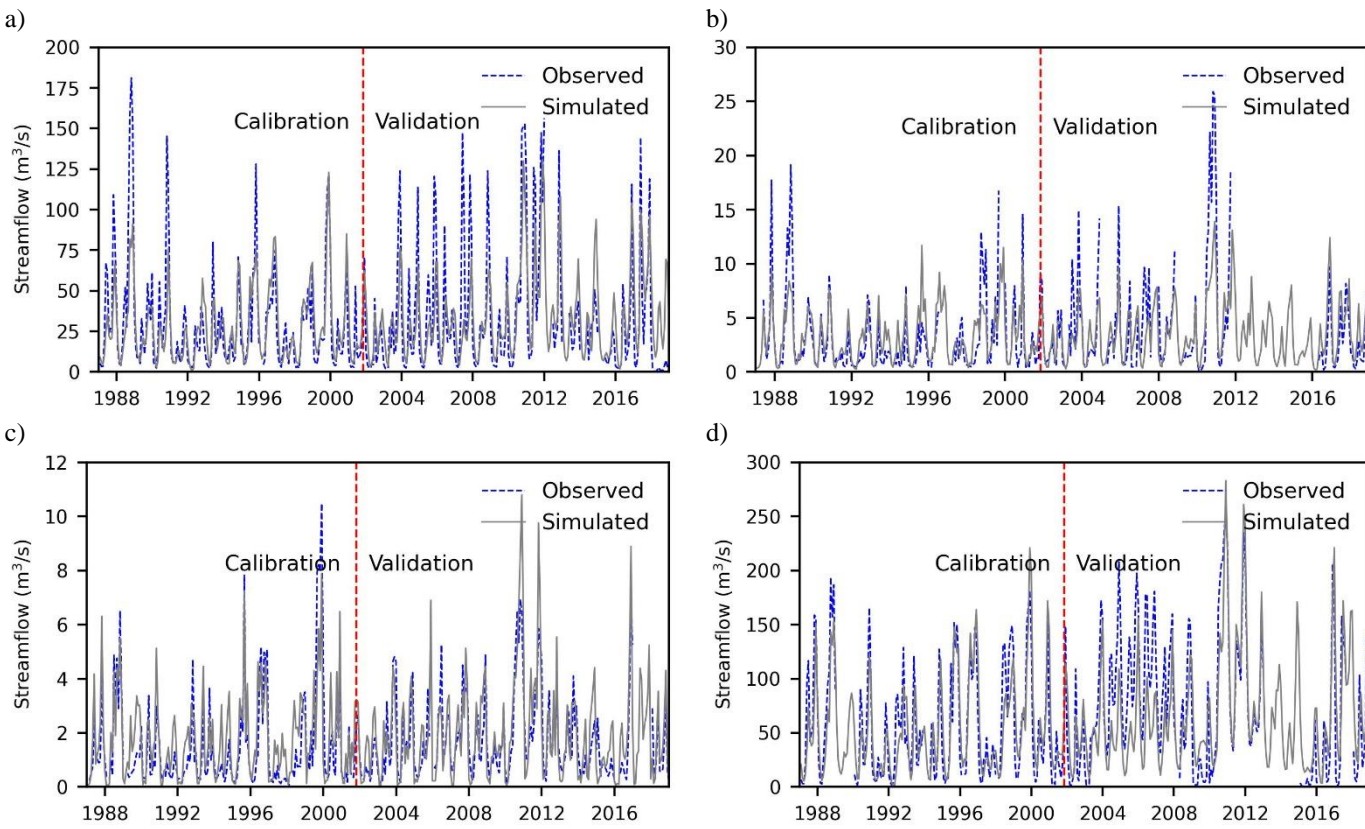

**Figure 3** Monthly calibration and validation for streamflow at: a) Puente Salguero, b) Puente Carretera, c) Cantaclaro and d) Puente Canoas.

Since the study focuses on droughts, the model performance simulating streamflow in the dry season was analysed separately. Performance indicators were calculated for the period corresponding to the basin's dry season (December to March). The intermediate period of precipitation decrease from June to July was also included in this analysis. Table 5 summarises the calibration and validation performance indicators in the dry season. According to the rating guidelines, the model performance simulating streamflow in the dry season is satisfactory (ASABE, 2017).

**Table 5.** SWAT model performance simulating flows in the dry season.

| Gauging station | Calibration | | Validation | |
|---|---|---|---|---|
| | NSE | PBIAS [%] | NSE | PBIAS [%] |
| Puente Salguero | 0.65 | -19.4 | 0.53 | -21.3 |
| Puente Carretera | 0.67 | -15.3 | 0.53 | 17.2 |
| Cantaclaro | 0.67 | -3.6 | 0.58 | 16.3 |
| Puente Canoas | 0.55 | -15.7 | 0.60 | -13.5 |

## 3.2 Hydroclimatic drivers of droughts

Figure 4 presents the numerical and categorical hydroclimatic parameters used as potential drivers of droughts. Figures 4a to h present the multi-annual average of the numerical hydroclimatic drivers of droughts at each subbasin. The average was calculated using the hydrological model's results during the simulation period (1987 to 2018). Figures 4i to k present the categorical drivers: the curve number, slope and soil type. The dominant category at each subbasin is shown in Figs. 4i to k. The dataset of explanatory variables was created from the values presented in Fig.4.

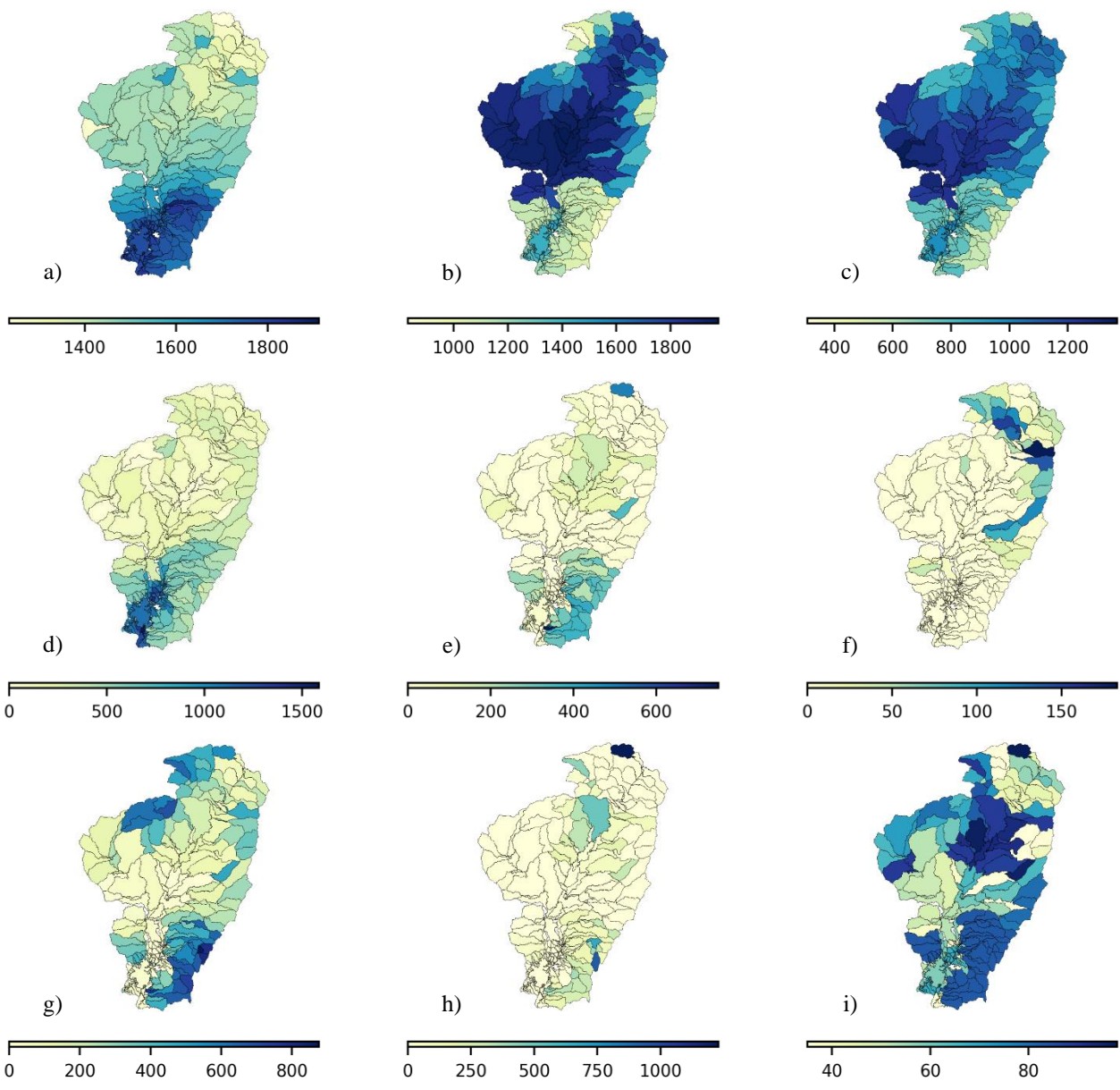

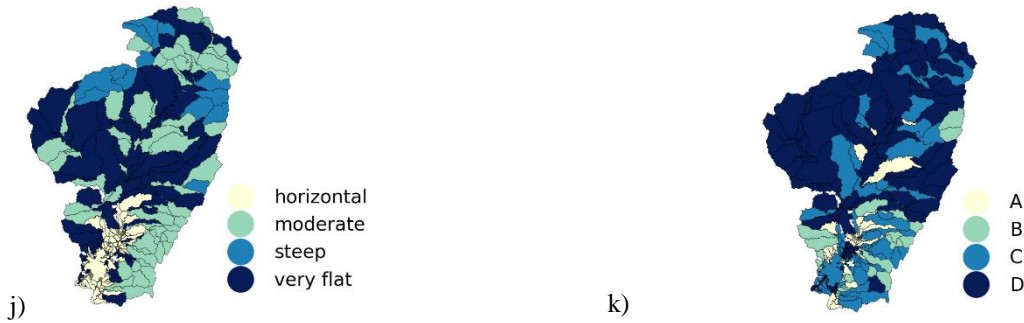

j)                                                                        k)

**Figure 4** Average value of hydroclimatic parameters during the simulation period at each subbasin: a) precipitation in mm, b) potential evapotranspiration in mm, c) actual evapotranspiration in mm, d) percolation in mm, e) surface runoff in mm, f) groundwater contribution to streamflow in mm, g) water yield in mm, h) sediment yield in metric tons/ha, i) curve number, j) slope and k) soil type, the soil hydrologic groups A, B, C and D refer to the soil's infiltration characteristics.

### 3.3 Drought events during the simulation period and their duration

We identified the drought events and estimated their duration following the definition of droughts presented in 2.2.2. A month was summed to the duration of an event when a number of subbasins, covering at least 30 % of the basin's total area, were in a drought state (moderate, severe or extreme). The identified droughts in the simulation period were in good agreement with the chronology of drought events in Colombia described at the National Study of Water (Instituto de Hidrología, 2019). Table 6 shows the dates and durations of the drought events.

**Table 6.** Agricultural and hydrological droughts during the period of analysis

| Event | Agricultural droughts | | Hydrological droughts | |
|:---:|:---:|:---:|:---:|:---:|
| | Date | Duration [months] | Date | Duration [months] |
| I | May 1991 – Jun 1992 | 13 | Apr 1991 – May 1992 | 14 |
| II | Jun 1997 – April 1998 | 11 | Apr 1997 – Feb 1998 | 11 |
| III | Jun 2001 – Aug 2001 | 3 | May 2001 – Jun 2001 | 2 |
| IV | Oct 2009 – Jan 2010 | 4 | Sep 2009 – Nov 2009 | 3 |
| V | Jun 2014 – Aug 2014 | 3 | Jun 2014 – Jul 2014 | 2 |
| VI | May 2015 – Jul 2016 | 15 | Apr 2015 – Apr 2016 | 13 |

After identifying the agricultural and hydrological drought events, it was possible to determine the number of months for each drought category in each subbasin, as represented in Figs. 5 and 6. The results presented in Figs. 5 and 6 are the response variables for the MVRT technique.

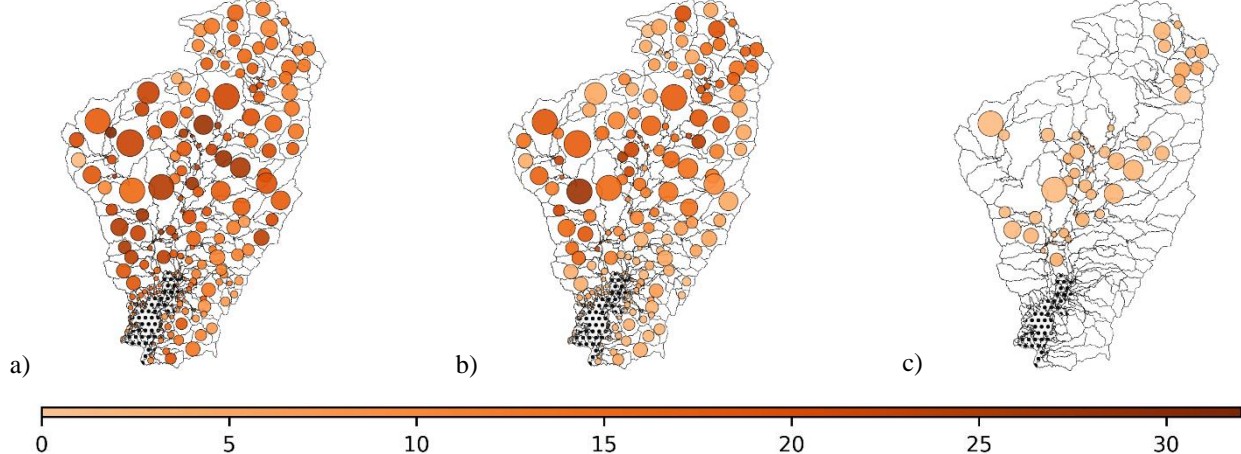

Figure 5 Months in each agricultural drought category: a) moderate, b) severe and c) extreme. SMDI was not calculated in the wetland subbasins (i.e. hatched area).

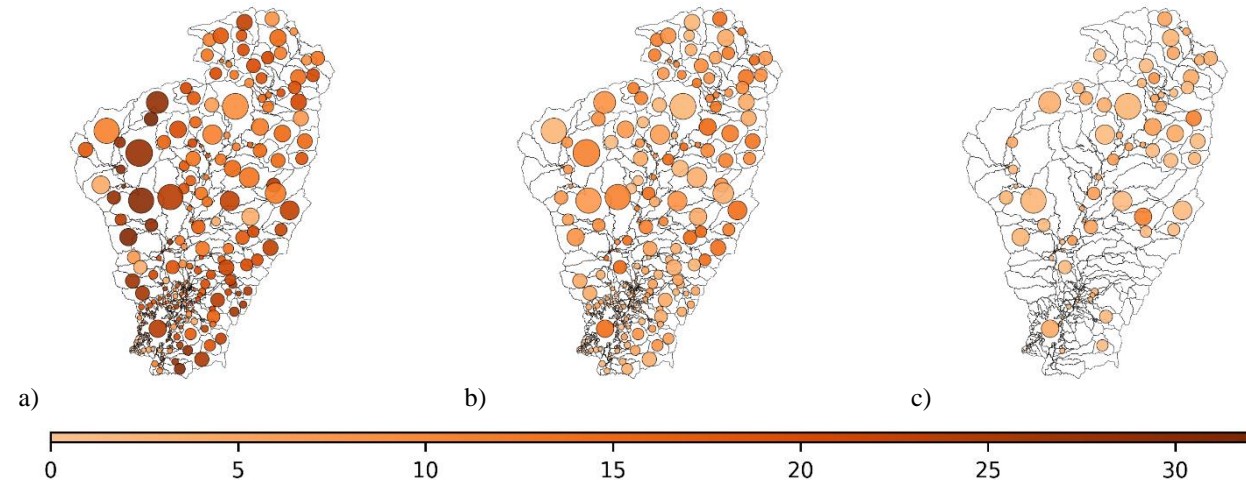

Figure 6 Months in each hydrological drought category: a) moderate, b) severe and c) extreme.

## 3.4 Multivariate regression tree

In this section, we describe the results of the MVRT technique applied to identify the governing drivers of agricultural and hydrological drought severity and their critical thresholds.

### 3.4.1 Drivers of agricultural drought

Figure 7 presents the tree generated by R software, the number of subbasins clustered at each terminal group (variable "*n*"), and the spatial distribution of these subbasins. The tree consists of five levels of split and twelve leaves. The minimum value of the cross-validation error (CVRE = 0.46) was used to select the tree size. The relative error of the MVRT was 0.19, and the

EV was 0.81. 8 hpresents the tree's numerical output: namely, the number of months for each drought category. The scattering of the outputs in each leaf allows us to identify the most susceptible subbasins to agricultural droughts.

The MVRT indicated that actual evapotranspiration was a strong driver of agricultural droughts; it appeared three times at different tree levels of split (Fig.7). The subbasins were split at the first level according to ET (924 mm). At the second level

of split, precipitation (1,318 mm) was used for the left branch of the tree and percolation (271 mm) for the right branch. Then, the left branch was recursively split as follows: at the third level, according to potential evapotranspiration (1,888 mm) and evapotranspiration (1,191 mm); at the fourth level, according to evapotranspiration (1,064 mm) and percolation (111 mm); and at the fifth level, according to sediment yield (101 tons/ha). The left branch accounts for seven out of the tree's twelve leaves. Regarding the right branch, splitting was done according to evapotranspiration (729 mm) and the curve number (67) at

the third level and according to the water yield (352 mm) at the last level. In the following, we describe agricultural drought MVRT terminal groups.

Leaf *a* clusters seven subbasins in the north part of the basin. In this area, actual evapotranspiration and potential evapotranspiration were above the basin average, while precipitation was below average (Figs. 4c, b and a, respectively).

Figure 8a shows that these subbasins experienced the highest number of months in extreme agricultural drought and a median of fifteen months in severe agricultural drought. Leaf *b* clusters two subbasins in the western part of the basin. In this leaf, there are no months in the extreme drought category. The median of months in the moderate and severe agricultural drought categories is ten months, one of the lowest among the terminal groups (Fig. 8b).

Leaves *c* and *d* cluster twenty-four and nineteen subbasins, respectively. Leaf *c* groups subbasins located in the upper part of the river course and the basin east. Precipitation was slightly below the basin average in the subbasins located in the north and close to the average in subbasins in the east (Fig. 4a). Leaf *d* groups subbasins located in the upper course of the river and in the basin's western part. The actual evapotranspiration threshold to split leaves *c* and *d* is 1,064 mm, value above the basin average (Fig. 4c). For subbasins with actual evapotranspiration below the threshold, leaf *c*, the median of months in the severe

drought category is below ten (Fig. 8c). For subbasins with actual evapotranspiration above the threshold, leaf *d,* the median of months in the severe drought category is sixteen, one of the highest among the terminal groups (Fig. 8d).

Leaves *e, f* and *g* cluster twenty-four, six and twelve subbasins, respectively. Subbasins are located in the river valley and the basin's western part. In these subbasins, precipitation was below the basin average (Fig. 4a), and actual evapotranspiration was

above the average (Fig. 4c). The percolation threshold to split leaves *e* and *f* from leaf *g* is 111 mm, a value considerably below the basin average (Fig. 4d). At the fifth level of split, the sediment yield threshold to split leaves *e* and *f* is 101 metric tons/ha, a value close to the average in the basin (Fig. 4h). Figures 8e, f and g show that subbasins clustered in these leaves are prone

to agricultural droughts. The median of months in the moderate drought category was above twenty months; the severe category was above ten months; and the three leaves exhibited months in the extreme drought category.


Leaves *h*, *i* and *j* cluster twenty-six, fifty-two and fifty-six subbasins, respectively. Subbasins are mainly located in the wetland surroundings, La Serranía (leaf *i*), and some outliers are located in the basin's north (leaves *h* and *j*). Percolation in leaves *h*, *i* and *j* was close to the basin average (Fig. 4d). Actual evapotranspiration in terminal groups *h* and *i* was relatively close to the basin average (Fig. 4c). The water yield threshold to split clusters *h* and *i* is 352 mm. Overall, subbasins clustered at leaves *h*,

*i* and *j* presented low susceptibility to severe and extreme agricultural drought conditions. The median of months in the moderate drought category was slightly higher than ten; the median for months in the severe category was the lowest for the study area and showed no months in the extreme drought category (Figs. 8h, I and j).

Leaves *k* and *l* cluster two and six subbasins, respectively. Subbasins are located towards the basin's north, and one outlier is

observed in the subbasin east (leaf *l*). In these subbasins, percolation was lower than 271 mm, value relatively low compared to other basin areas (Fig. 4d). In leaf *k*, the curve number was lower than sixty-seven, while in leaf *l,* it was higher. In leaf *k,* the median of months for the moderate category is ten, and for the severe category, it is 14. In leaf *l,* the median of months in the moderate category is above ten, and the subbasins experienced some months in severe drought. Leaves *k* and *l* show no months in the extreme drought category (Figs. 8k and l).

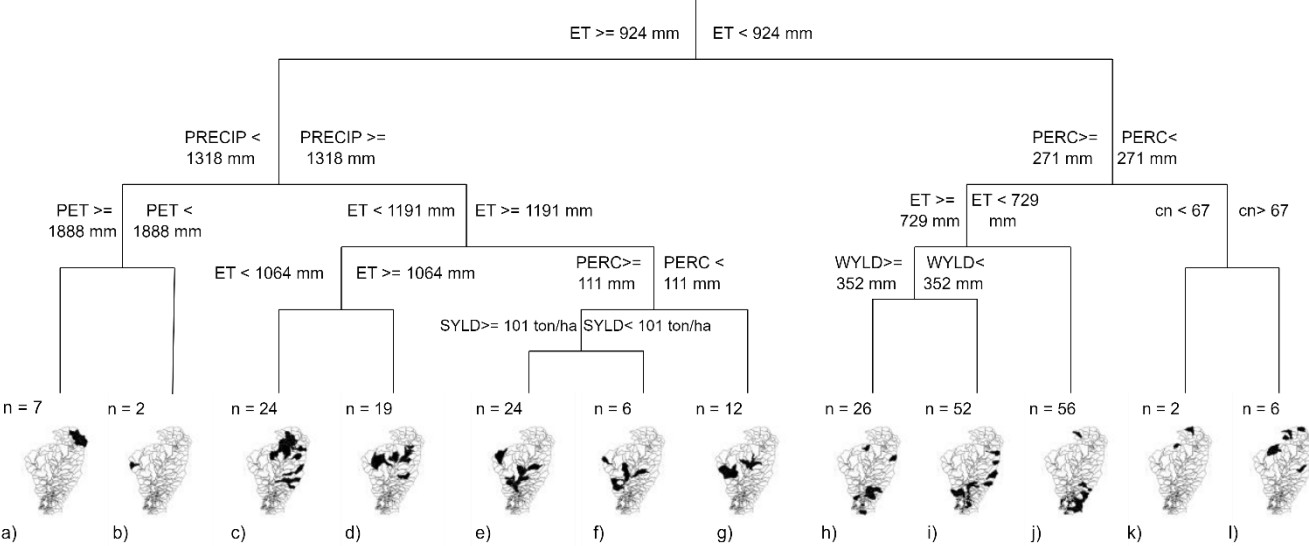


**Figure 7** MVRT of hydroclimatic drivers of agricultural droughts at the Cesar River basin and spatial distribution of the subbasins clustered at each leaf. Tree leaves are named from *a* to *l*, and *n* indicates the number of subbasins clustered at each leaf. The wetland subbasins are not included in the analysis for agricultural drought.

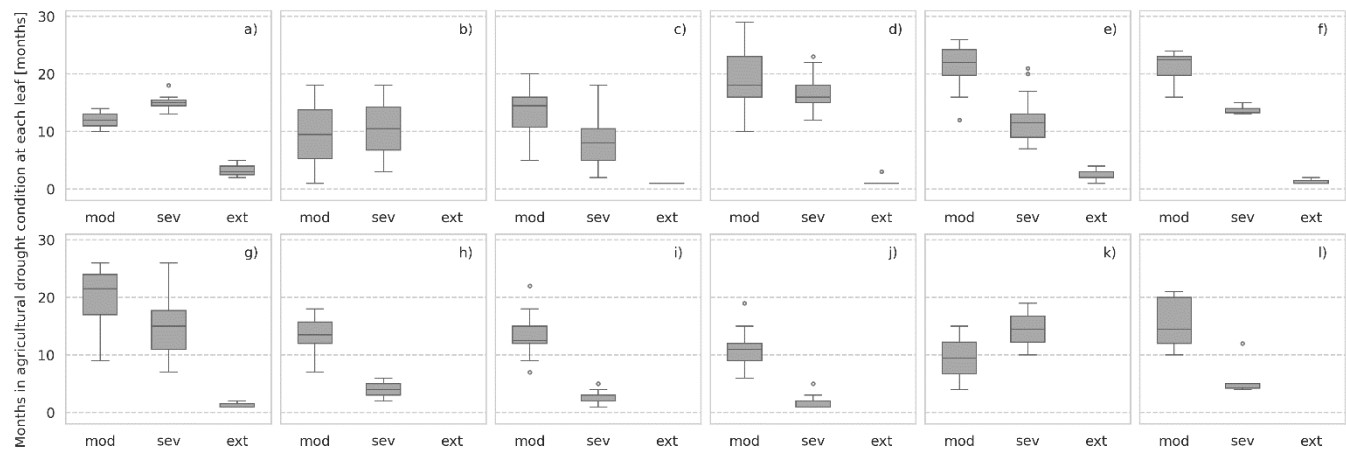

**Figure 8** Number of months in agricultural drought categories (moderate, severe, extreme) at each leaf. Tree leaves are named from *a* to *l*.

### 3.5.2 Drivers of hydrological drought

Figure 9 presents the hydrological drought MVRT, the number of subbasins clustered at each terminal group (variable "*n*") and the spatial distribution of these subbasins. The tree consists of four levels of split and eight leaves. The minimum value of the cross-validation error (CVRE = 0.67) was used to select the tree size. The relative error of the MVRT was 0.52, and the EV was 0.48. Figure 10 presents the tree's numerical output: namely, the number of months for each drought category. This information allowed us to identify the clusters of subbasins prone to hydrological droughts.

The MVRT demonstrated that precipitation was a primary driver of hydrological drought; it appeared two times at different levels of split. The subbasins were separated at the first split level according to precipitation (1632 mm). At the second split level, precipitation (1398 mm) was used as the left branch of the tree, and water yield was used as the right branch (29 mm). The left branch was then further divided according to percolation (153 mm) at the third level and according to curve number (51) at the fourth level. At the third level, the right branch was split according to evapotranspiration (833 mm) and surface runoff (0.5 mm). The MVRT terminal groups were then examined in detail.

Leaf *a* clusters twenty-eight subbasins in the upper basin and one outlier located in the western part of the subbasin (Fig. 9a). In these subbasins, precipitation was considerably below the basin average (Fig. 4a). Figure 10a shows that the subbasins in this terminal group repeatedly experienced moderate, severe and extreme hydrological drought.

Leaves *b* and *c* cluster thirty-seven and thirteen subbasins, respectively. Subbasins clustered at leaf *b* are relatively distant; most are towards the eastern part of the basin, and the rest are in the north and west of the basin. Subbasins in leaf *c* are located in the river's middle course towards the western part of the basin and some outliers in the north. Precipitation and percolation were slightly above the basin average in subbasins clustered at leaves *b* and *c* (Figs. 4a and d). The curve number threshold to

split leaves *c* and *d* is 51. Subbasins with a curve number above the threshold, leaf *b*, experience months in extreme drought and present one of the highest median of months for severe drought (Fig. 10b). For subbasins with curve number below the threshold, leaf *c,* the median of months at moderate drought is almost 20 and experience months at severe and extreme category (Fig. 10c).

Leaf *d* clusters twenty-nine subbasins in the river's middle course and the basin's eastern part. Figure 10d indicates that in this terminal group, the subbasins experienced fewer months in the severe and extreme drought categories than the other clusters in the tree's left branch; however, subbasins experienced one of the highest median of months at moderate drought.

In leaves, *e* (*n* = 72) and *f* (*n* = 23), precipitation exceeded the basin average and water yield was considerably high in the subbasins in La Serranía del Perijá (Figs. 4a and g). The actual evapotranspiration threshold to split leaves *e* and *f* is 833 mm, value below the basin average (Fig. 4c). Both terminal groups describe moderate exposure to hydrological drought. At leaf *e*, the median of months in the severe and extreme drought categories is below ten, while the median of months in the moderate drought category is twenty (Fig. 10e). The hydrological drought exposure of the subbasins clustered at leaf *f* is also mild. In these subbasins, actual evapotranspiration is above the threshold and close to the basin average. These subbasins present the lowest median of months for all drought categories (Fig. 10f). Notably, the Zapatosa marsh and upstream subbasins are clustered in this terminal group (Fig. 9f).

Leaves *g* and *h* cluster seventy-one and forty subbasins, respectively. Subbasins clustered at these leaves are located upstream of the Zapatosa marsh. The surface runoff threshold to split the leaves *g* and *h* is 0.5 mm. Figure 10g shows that the subbasins grouped at leaf *g* present the low suceptibility to hydrological drought. The median of months for all categories is the lowest in the basin. In leaf *h*, the surface runoff was lower than 0.5 mm. In these subbasins, the medians of months in the severe and extreme categories are relatively low, while the median of months in the moderate category is eighteen (Fig. 10h).

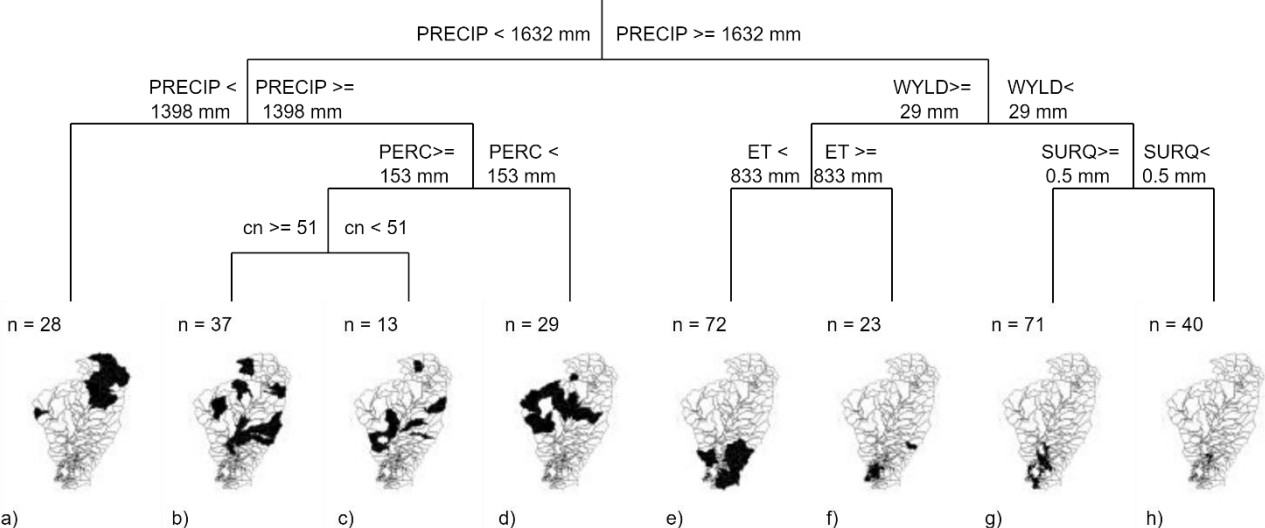

**Figure 9** MVRT of hydroclimatic drivers of hydrological drought at the Cesar River basin and spatial distribution of the subbasins clustered at each leaf. Tree leaves are named from *a* to *h,* and *n* indicates the number of subbasins clustered at each leaf.

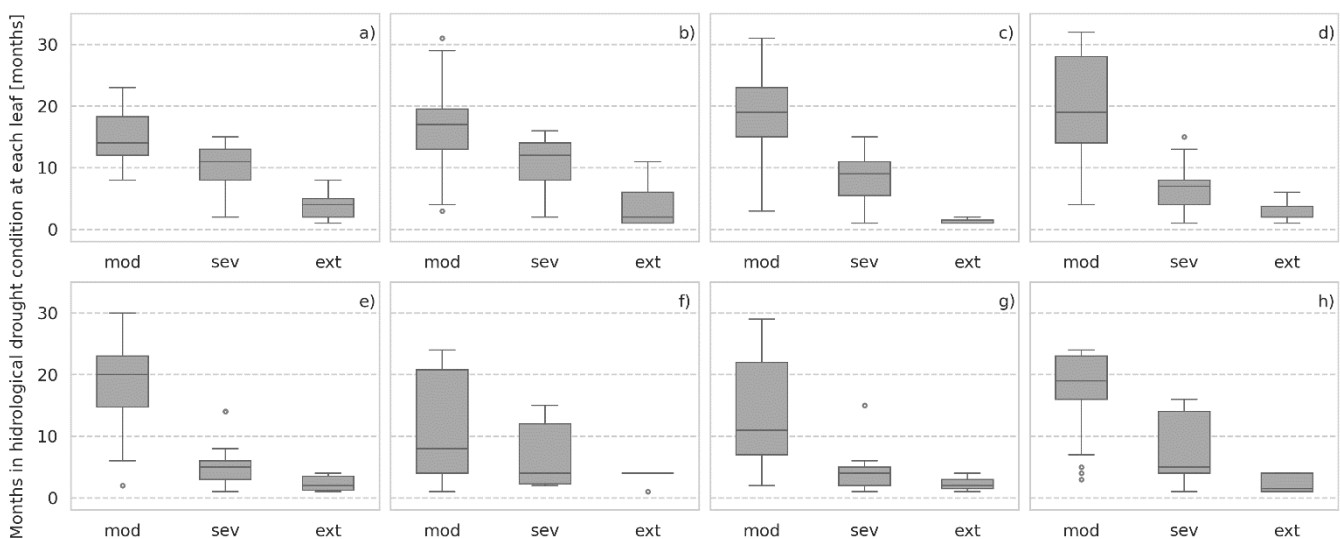

**Figure 10** Months in hydrological drought categories (moderate, severe, extreme) at each leaf. Tree leaves are named from *a* to *h.*

## 4. Discussion

### 4.1 Hydroclimatic drivers of agricultural drought

The left branch of the MVRT clusters the subbasins susceptible to severe agricultural drought (Figs. 8a, d, e, f and g). Conversely, the right branch of the MVRT clusters the subbasins experiencing moderate agricultural drought severity. The subbasins in leaves *h*, *i* and *j* predominately experienced months in the moderate drought category (Figs. 8h, i, and j).

Interestingly, agricultural drought severity in leaves *a*, *e*, *f* and *g* was comparable but governed by different parameters. For instance, leaf *a* presented the highest median of months for severe and extreme agricultural drought (Fig. 8a). The drought drivers in this terminal group, namely precipitation and potential evapotranspiration, indicate that agricultural drought results from an imbalance between the soil moisture supply (i.e. precipitation relatively close to the minimum value at the basin) and soil moisture demand (i.e. moderately high potential evapotranspiration). Leaves *b*, *c*, and *d* corroborate the significant influence of evapotranspiration on agricultural drought severity. A comparison of clusters *a* and *b*, and *c* and *d* indicates that the leaves with higher evapotranspiration are more prone to experience severe drought. It is interesting to notice that in clusters *c* and *d*, the actual evapotranspiration threshold causes a notable difference in drought severity. While the leaf *c*, clustering subbasins with actual evapotranspiration below 1064 mm presents the lowest median of months at severe category at the left branch of the tree, leaf d shows the highest median of months at the same category in the tree.

This finding aligns well with studies demonstrating that potential evapotranspiration considerably enhances the severity of agricultural droughts in water-limited areas (Ding et al., 2021; Manning et al., 2018; Teuling et al., 2013). According to such studies, potential evapotranspiration influence on agricultural drought severity may be explained by the significant increase in net radiation during droughts, as the lack of rainfall usually concurs with decreased cloud cover.

In contrast, the MVRT outcomes suggest that a lack of precipitation is not a primary driver of agricultural drought in the subbasins clustered at leaves *e*, *f* and *g*. Particularly, leaf *e* grouped the subbasins that experienced the most severe agricultural drought in the analysis period. The median of months in the moderate drought category was above twenty; the severe category was above ten; and subbasin experienced months in extreme category (Fig. 8e). The observed evapotranspiration and percolation thresholds might indicate poor precipitation partitioning and a disturbed water regime that favours water lost by runoff and evapotranspiration. Furthermore, the sediment yield threshold (notably above the median) may be linked to poor soil structure, thus compromising soil water retention capacity and enhancing drought severity.

The results from leaf *e* show that a higher sediment yield slightly increases the occurrence of extreme droughts (Fig. 8e), as compared to the results from leaf *f*. This agrees with earlier findings concluding that soil degradation enhances agricultural drought characteristics (Masroor et al., 2022; Santra & Santra Mitra, 2020; Trnka et al., 2016). Further, our results are

consistent with previous studies that indicate the incidence of droughts is not only caused by extreme weather events but also by the inefficient soil–water management associated with land and soil degradation (Cornelis et al., 2019; Wildemeersch et al., 2015).

510

The right branch of the tree provides valuable information on the hydroclimatic parameters that reduce the severity of agricultural droughts. Moderate drought susceptibility in leaves *h*, *i* and *j* is linked to relatively low evapotranspiration thresholds; accordingly, it may be asserted that evapotranspiration controlling measures (e.g. surface cover, crop rotation, agroforestry, intercropping) are relevant interventions for building resistance to agricultural drought. At terminal groups *h* and *i*, water yield was found to influence the severity of agricultural drought. Notably, the subbasins at leaf *i* were slightly more resistant to drought (Fig. 8i); this indicates that measures aimed at increasing the subbasins' water storage capacity (e.g. rainwater and floodwater harvesting techniques) are suitable interventions to reduce the severity of agricultural drought.

Some of the subbasins grouped at leaf *i* showed high exposure to hydrological drought (Figs. 10b and c). Contrasting exposure to agricultural and hydrological droughts suggests that the water retention capacity in these subbasins reduces the severity of agricultural drought events but limits the contribution of surface runoff, lateral flow and groundwater to the streamflow, thus exacerbates the water deficit and hydrological drought severity. Therefore, drought management interventions require the prior assessment of the potential effects on both types of droughts.

### 4.2 Hydroclimatic drivers of hydrological droughts

The subbasins clustered on the left branch of the tree were prone to hydrological drought (Figs. 10a, b, c, d). Leaf *a* presented the highest median for months in the severe and extreme hydrological categories. The analysis results confirmed that precipitation deficits caused the severe hydrological drought conditions in the upper part of the basin.

Conversely, the MVRT also showed that in terminal groups *b*, *c* and *d*, hydrological drought severity was linked to the inefficient partition of precipitation. Selected drivers (precipitation, percolation and curve number representing land use) are widely recognised as predominant drivers of hydrological droughts (Iglesias et al., 2018; Stoelzle et al., 2014; van Lanen et al., 2013; van Loon, 2015). The difference observed between the precipitation and percolation thresholds suggests that a large part of rainwater was lost either by evapotranspiration or surface runoff (or other water abstractions, e.g., human consumption, agriculture). Low percolation values limited the groundwater contribution to the streamflow, enhancing the streamflow deficit during drought periods.

Interestingly, the curve number was selected as a driver of hydrological drought for leaves *b* and *c* (Figs. 9b and c). The subbasins in leaf *b* presented higher curve numbers than those in leaf *c* and higher exposure to hydrological drought. High curve number values are commonly the result of anthropogenic changes in land cover, which modifies evapotranspiration and

the division of precipitation into evapotranspiration and streamflow. The present selection of the curve number at the third level of split is consistent with previous studies, which established that hydroclimatic parameters and human activities influence hydrological droughts; however, the influence of both drivers is uneven. Results indicate that hydroclimatic parameters are more influential (Jehanzaib et al., 2020; Saidi et al., 2018).

The right branch of the MVRT grouped subbasins with moderate and intermediate exposure to hydrological drought. The hydroclimatic parameters and the thresholds used to define leaves *e* and *f* (precipitation, water yield and evapotranspiration) demonstrate that in these subbasins, precipitation values compensated for the water abstraction by evapotranspiration. When we compare the severity of the hydrological droughts observed in leaves *e* and *f*, we find that lower evapotranspiration values reduce exposure to severe and extreme hydrological drought but increase the incidence of moderate hydrological drought.


     The subbasins in terminal group *g* experienced the lowest median number of months for all hydrological drought categories (Fig. 10g). The water yield threshold indicates good water retention capacity in these subbasins. It can be explained by the proximity of the subbasins to the marsh (which acted as a natural control), the low slope in the area (which reduced streamflow velocity) and the presence of water bodies (which collected and stored runoff during the rainy season). The runoff threshold

indicates that part of rainwater reaches the streamflow; nevertheless, the subbasins in cluster *g* have one of the lowest runoff potentials in the basin (Fig. 4e). On the contrary, in these subbasins, percolation is considerably high (Fig. 4d). This seems to confirm that low susceptibility to hydrological droughts is linked to subbasins water retention capacity. The present findings suggest that the water storage capacity of the Zapatosa marsh can compensate for the increased evaporation that occurs during drought events, thereby alleviating hydrological drought severity upstream. Our results concur with previous analyses

concluding that wetlands (located in different climatic regions) significantly alleviate hydrological drought severity when direct evaporation from the water body does not significantly reduce water storage (Wu et al., 2023).

     The hydrological drought conditions in the subbasins clustered at leaf *h* were mild, despite water yield values below 29 mm (Fig. 10h). Negligible surface runoff values indicated that in leaf *h,* rainfall is stored in the soil profile, lost by

evapotranspiration or percolates in an area of minimal baseflow contribution to streamflow. This limits the amount of water reaching the streamflow and enhances the severity of hydrological droughts, compared to leaf *g*.

### 4.3 Comparison of the hydroclimatic parameters influencing the severity of agricultural and hydrological droughts

     Crucial similarities and differences emerge from contrasting the parameters influencing the severity of droughts and the spatial distribution of the subbasins experiencing severe and mild drought conditions. MVRTs indicate that severe agricultural and

hydrological drought conditions occurred in the upper and middle course of the river. Nevertheless, the severe droughts were influenced by different hydroclimatic factors. Severe agricultural drought in the headwater was driven by the interaction between precipitation shortfalls and high potential evapotranspiration (Fig.7a). Conversely, severe hydrological drought

condition was solely driven by limited precipitation. It is worth highlighting that the severe hydrological situation extends from the headwater to the subbasins in the middle course (Fig. 9a).


Downstream, in subbasins located in the middle course, the agricultural and hydrological drought situation was also severe. In this area, drought severity was linked to inadequate rainfall partitioning and an unbalanced water cycle that favours water loss through evapotranspiration and low percolation values (Figs.7d, e, f and g, and Figs. 9b, c and d). Significantly, agricultural and hydrological droughts in these leaves were more severe than in leaves experiencing precipitation deficits (Fig.7a and Fig. 9a). Results also suggest that poor soil structure enhanced severe agricultural drought conditions (Fig.7e), and high curve numbers seem to increase hydrological drought severity (Fig. 9b).


MVRTs also showed subbasins experiencing mild agricultural and hydrological drought severity. Overall, these subbasins were located in the southern part of the basin. However, for agricultural drought, a few cases were observed in the north of the basin (Figs.7h, i and j). Subbasins presenting mild hydrological drought severity are allocated upstream of the Zapatosa marsh (Fig. 9g). Moderate agricultural drought severity was linked to low evapotranspiration losses and the subbasins' capacity to retain water in the soil profile, improving percolation (Fig.7j). In turn, moderate hydrological drought severity related to the subbasins' proximity to the marsh (which acted as a natural control reducing the water yield) and surface runoff contributions to the streamflow (Fig. 9g). Remarkably, some of these subbasins also showed mild agricultural drought conditions (Fig.7i).


## 4.4 Accuracy of the MVRTs


The high EV (0.81) value indicates the good explanatory power of the tree built for agricultural drought. This confirms that the selected explanatory variables significantly influence the severity of agricultural drought. Nevertheless, two potential disadvantages of the tree are identified. First, clusters $h$ and $i$ are very similar. Drought severity is alike in these leaves, and the parameters influencing droughts are the same. This suggests that these two clusters can be merged into one. Second, leaves $b$ and $k$ cluster only two subbasins. Accordingly, the distribution presented in the boxplots must be interpreted cautiously. Neither of these disadvantages compromises the study's main findings; however, further analysis is recommended to determine the size of the tree (number of clusters) that better fits the assessment of the hydroclimatic drivers of droughts.


Conversely, the explanatory power of the tree built for hydrological drought is not very high (EV = 0.48). This may be related to the inaccurate representation of groundwater contribution to the streamflow. Streams depend significantly on groundwater during droughts to maintain flow; nevertheless, groundwater contribution to the streamflow was not included as a key drought driver in the MVRT, although it was in the list of explanatory variables. It is possible that the model's simplifications for the simulation of groundwater flow and storage did not adequately represent the groundwater contribution to the streamflow (Molina-Navarro et al., 2019). The lack of adequate information about this relevant factor hydrological drought may have compromised the MVRT's accuracy. Unexplained variability may also link to factors that influence hydrological drought but



were not considered in the dataset of explanatory variables (e.g. abstractions such as water for irrigation, industry or human consumption).

## 5. Conclusions

In this study, a machine learning technique, namely multivariate regression tree (MVRT), was applied. The main aim was to
build an 'explanatory AI' model to explicitly identify relationships between a subbasin's hydroclimatic characteristics (i.e. explanatory variables) and the severity categories of agricultural and hydrological drought (i.e. response variables). The results show that the machine learning technique identifies drought severity's primary drivers and critical thresholds reasonably well. Notably, the MVRT built for agricultural drought shows a good explanatory power. The MVRT also identifies parameters which can contribute to reducing agricultural and hydrological drought severity.


The outcomes of the MVRT provide valuable information on the hydroclimatic parameters influencing the drought-generating process in the Cesar River basin. MVRTs indicate that severe agricultural and hydrological drought conditions observed in the upper and middle course of the river are influenced by different hydroclimatic factors. The interaction between precipitation shortfalls and high potential evapotranspiration drives severe agricultural drought in the headwater. Conversely, severe
hydrological drought condition is mostly caused by limited precipitation. In subbasins in the middle course, drought severity is linked to inadequate rainfall partitioning and an unbalanced water cycle, favouring water loss through evapotranspiration and low percolation values. Notably, results suggest that poor soil structure enhances severe agricultural drought conditions, and high curve numbers seem to increase hydrological drought severity. In the southern region, subbasins experience moderate agricultural and hydrological drought severity. Mild agricultural drought is linked to low evapotranspiration losses and
subbasins' capacity to retain water in the soil profile, improving percolation. In turn, moderate hydrological drought severity relates to the subbasins' proximity to the marsh (which acted as a natural control reducing the water yield) and surface runoff contributions to the streamflow. The outcomes of this study also demonstrate that the combined effect of parameters with low impact can trigger a drought situation as severe as the one produced by one or two of the most influential parameters. It is worth mentioning that the study outcomes indicate that the slope and the soil type do not influence the severity of agricultural
and hydrological droughts in the Cesar River Basin.

It can also be concluded that the MVRT (and other machine learning techniques that generate 'explainable AI' models based on progressive tree-like data partitioning and simplified models in leaves) is a relevant tool for defining drought management strategies. The tool helps to identify drought-prone areas and design management strategies that contribute to maintaining the
hydrological parameters influencing droughts above (or below) the thresholds that trigger severe and extreme drought conditions.

This study is not without limitations. First, we used a simplified approach to modelling a complex phenomenon using SWAT software (e.g. representing the groundwater components that impact hydrological drought conditions). Second, only a single ML technique was employed to build explainable models. Further extensions of this research may address these limitations.

For example, candidate ML techniques could include M5 model trees (rather than regression trees), which have shown their effectiveness in solving water-related problems (see Solomatine & Dulal, 2004; Solomatine & Xue, 2004). These result in linear models in tree leaves rather than constants like in regression trees. Additionally, there is still a need to better represent anthropogenic interventions (and other relevant parameters influencing droughts) in the set of explanatory variables, e.g. abstractions such as water for irrigation, industry or human consumption, groundwater pumping.


The issue of combining human and artificial intelligence (and knowledge of physics with machine learning) is currently a point of great interest (see Jiang et al., (2020) on 'physics-aware deep learning models', Moreido et al., (2021) and Bertels et al, (2023), on the role of experts in constraining machine-learning and hydrological models). However, the mentioned approaches directly incorporate physical knowledge into ML models, and domain experts still see the resulting models as "black boxes".

This study can be seen as the one that contributes to developing and testing tools to better incorporate 'explanatory' ML, leading to models that can be overviewed and analysed by experts and hence have better potential for inclusion into existing modelling and management practices.

*Data Availability*. The data is available on request.


*Authors contributions*. All authors contributed to the study conception and design. Material preparation, data collection and analysis were performed by Ana Paez-Trujillo. Jeffer Cañon developed the hydrological model. The first draft of the manuscript was written by Ana Paez-Trujillo, and all authors commented on previous versions of the manuscript. Ana Paez-Trujillo, Jeffer Cañon, Beatriz Hernandez, Gerald Corzo and Dimitri Solomatine read and approved the final manuscript.


*Competing interests.* The contact author has declared that neither she nor her co-authors have any competing interest.

*Acknowledgements.* The authors would like to express their gratitude to the Natura Foundation and the project GEF Magdalena–Cauca VIVE for proving the hydrological model of the Cesar River basin.


*Financial support.* This study was financially supported by the Ministry of Education Colombia, Programa Colombia Cientifica, Grant No. 3597287. The authors have no relevant financial interest to disclose.

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
