# Peer review of "Multivariate regression trees as an 'explainable machine learning' approach to explore relationships between hydroclimatic characteristics and agricultural and hydrological drought severity: Case of study Cesar River basin."

_Natural Hazards and Earth System Sciences, 2023_

## Author Comment (AC1)

**Response to Reviewer 1 Comments**

**Manuscript title:** Multivariate regression trees as an 'explainable machine learning' approach to exploring relationships between hydroclimatic characteristics and agricultural and hydrological drought severity

**Author's general response:**

The authors would like to thank the reviewer for the time given to this manuscript and for providing insightful and detailed comments to help us to improve this manuscript's overall scientific quality and readability. Your attention to detail has undoubtedly enhanced the overall strength of our study. Notably, we appreciate the comments about the missing definition of drought severity and the lack of information on the application of MVRT to this particular study. We will apply multiple changes to incorporate the reviewer's suggestions and clearly define the study's objective. In the following, you will find the answers to the general and specific comments. Some of them required a particular action or change in the manuscript. The changes we will apply in the Revised Manuscript (RM) are in italics.

**General Comments.**

1. My main concern regards the lack of clarity in the objective of the study. The title fails to mention that it is an application to a specific case study. In addition, it is not clear until deep in the "results section" what exactly the authors mean with drought severity. For almost the entire paper the readers are left wondering what exactly is modelled with the MVRT. Is it the severity of a series of events on the entire basin? Is it the spatial distribution of the severity? This should be made clear already in the objective described in the introduction, and then detailed in the methodology.

The authors thank the reviewer for pointing out that the title does not mention that the study is an application to a case study. Accordingly, we will update the RM title, including the case of study:

*Multivariate regression trees as an 'explainable machine learning' approach to exploring relationships between hydroclimatic characteristics and agricultural and hydrological drought severity. Case of study Cesar River basin.*

Regarding the second part of the comment, we agree that the introduction needs to include the definition of drought severity and how it is represented in this study. In addition, it fails to describe what is modelled by applying the MVRT technique. Since both concepts are crucial elements of this study, we will apply two changes in the RM. In the introduction, we will include a paragraph presenting the definition of drought's severity and how it is represented using drought indices. In addition, we will update the introduction indicating

that drought severity categories (moderate, severe and extreme) are the three response variables modeled with the MVRT. We present the updated version of the introduction (There are no changes in the first two paragraphs of the 
[revised manuscript text omitted]

2. Another related issue of the paper is the lack of specific details of the application of MVRT to the given study case. Most of the description is rather generic, and do not answer key questions about the specific application. The authors state that one of the advantages of MVRT is the capability to output multiple variables, but it is never clarified why this is needed here and how this is exploited.

The authors agree with the reviewer that the introduction and methodology do not explicitly present the reasons for choosing the MVRT approach and how the technique capabilities are exploited in this study. Accordingly, two changes will be included in the RM. First, we will update the introduction presenting the MVRT capabilities relevant to the study.

*Despite remarkable progress achieved in understanding the drought-generating process and drought characterisation, there is still a need for studies that assess the complex interplay between the different drivers of droughts and how their combined effect influences drought characteristics (e.g. duration, severity, intensity) (Valiya Veettil & Mishra, 2020). Previous studies focus on the influence of one driver (Margariti et al., 2019; Mastrotheodoros et al., 2020; Shah et al., 2021; Xu et al., 2019), and some of the methodologies applied cannot*

*adequately address the non-linear relationship between climate, basin processes and droughts (Peña-Gallardo et al., 2019; Saft et al., 2016; van Loon, 2015).*

*We have found two studies that employ machine learning to analyse the non-linear relationship between climate and basin processes and droughts. Valiya Veettil et al. (2020) used a classification and regression tree (CART) to identify the variables influencing drought duration. Since CART allows one output variable (drought duration), the authors applied the technique three times to evaluate the variables influencing short-term, medium-term and long-term drought events. Meanwhile, Konapala et al. (2020) used a random forest (RF) algorithm to identify the climate and basin parameters influencing the characteristics (duration, frequency and intensity) of three different drought regimes (long duration and mild intensity, moderate duration and intensity, short duration and high intensity). As the core of RF is a decision tree that allows one output variable (in this case, each characteristic of each drought regime), the authors repeated the procedure for each drought regime and characteristic. Both studies focused on drivers of hydrological drought and were developed in the continental United States.*

*Mentioned research shows the potential of machine learning techniques for drought-related analysis; nevertheless, there is still a need for testing a technique capable of simultaneously assessing the influence of drought drivers on the individual categories of drought severity. Commonly used in the field of ecology to relate independent environmental conditions to populations of multiple species, Multivariate Regression Tree (MVRT) arises as a suitable technique for this purpose. MVRT is a supervised clustering technique that links explanatory variables to multiple response variables while maintaining the individual characteristics of the responses. Significantly, the technique does not assume a linear relationship between explanatory and response variables. Furthermore, it allows for the so-called "interpretable machine learning" algorithms that make decisions and predictions understandable to humans (Molnar, 2022). MVRT interpretably is a relevant attribute for drought researchers and planners since*

*the method allows them to identify the parameters influencing severe (or mild) drought conditions.*

Second, in the methodology, we will update the introductory paragraph of Section ***Multivariate regression tree approach for evaluating the relationships between hydroclimatic characteristics and droughts severity*** and include a paragraph describing the reasons for selecting the technique.

*MVRT is an extension of a regression tree (Breiman, 2001), but it differs in that it allows for multiple outputs (see De'ath, 2002). It allows the recursive split of a quantitative response variable (predictand, output) controlled by a set of numerical or categorical explanatory variables (predictors, input). The technique approach yields a set of non-linear models, each a piece-wise linear regression model (of zero order). An MVRT result is a tree whose terminal groups (leaves) of instances (input-output vectors) comprise subsets of instances selected to minimise the within-group sums of squares. Each successive split is given by a threshold value of the explanatory variables (Borcard et al., 2018). MVRT applies to dataset exploration, description and prediction (De'ath, 2002). In this study, explanatory variables are the hydroclimatic parameters at each subbasin, represented by the average value of each parameter during the analysis period (1987 to 2018). The number of months observed at each drought severity category (Categories are given by the drought indices) are the response variables. The analyses for agricultural and hydrological droughts were conducted separately; thus, two MVRTs were obtained.*

*Four technique attributes are relevant to this study. First, MVRT can capture the non-linear interactions between the parameters influencing droughts and their severity. Second, the technique can handle numerical and categorical hydroclimatic parameters influencing drought severity (explanatory variables). Third, MVRT's capability to handle multiple outputs allowed us to evaluate the influence of the hydroclimatic parameters on moderate, severe and extreme drought conditions simultaneously (response variables). The drought indicators give these three categories to represent the drought severity. Simultaneous analysis of different drought categories provides a comprehensive understanding*

*of the drought-generating process and the factors influencing severe (or mild) drought conditions. Fourth, MVRT results can be easily visualised and interpreted. The resulting tree structure provides a clear representation of the relationship between the drivers of droughts and the severity of agricultural and hydrological droughts.*

3. A lot more can be said on the "explainable" portion of the study. The authors provide some comments on the outcomes of the two MVRTs, but the link between these outputs and a physical interpretation is lacking. In both the discussion and the conclusion sections (as well as in the abstract), the authors stress how a main finding is the division of the domain in 3 macro regions. However, it is not clear how this conclusion is drawn from the outputs of MVRT, and how MVRT are "explained" to derive this conclusion. At the moment, it is seems that this conclusion is derived from previous knowledge of the area rather than the actual outcomes of the study.

Regarding the reviewer's concern about dividing the basin into three regions, the authors realized that the analysis results should be summarised differently. It is more precise to say that we identify different sets of parameters that govern drought severity in the basin. First, severe agricultural and hydrological drought conditions are driven by precipitation shortfalls and high potential evapotranspiration. This interaction is observed in the upper part of the river valley. Second, severe agricultural and hydrological drought conditions are caused by inadequate rainfall partitioning and an unbalanced water cycle favouring water loss through percolation and evapotranspiration. According to the results, the middle part of the river valley is affected by the interplay of these parameters. Finally, moderate exposure to agricultural and hydrological droughts is related to the capacity of the subbasins to retain water, which lowers evapotranspiration losses and promotes percolation. Moderate drought severity is observed in the Zapatosa marsh and the Serrania del Perijá foothills.

To improve the description of our results and ensure readers' clarity, we will not include the reference to the three regions in the RM. Following Reviwer's General Comment 4, we will compare the results from the two MVRT trees (See answer to General Comment 4). We agree that this is a better way to describe differences and similarities between the parameters influencing the severity of agricultural and hydrological droughts and present the spatial distribution of the areas experiencing severe and mild drought conditions. In the RM, the abstract and the conclusion will be updated accordingly.

*5. Conclusion (Second paragraph)*

*Our results provide valuable information on the hydroclimatic parameters influencing the drought-generating process in the Cesar River basin. MVRTs indicate that severe agricultural and hydrological drought conditions occurred in the upper and middle course of the river. Nevertheless, the severe droughts were influenced by different hydroclimatic factors. The interaction between precipitation shortfalls and high potential evapotranspiration drove severe agricultural drought in the headwater. Conversely, severe hydrological drought condition was solely caused by limited precipitation. In subbasins in the middle course, droughts' severity was linked to inadequate rainfall partitioning and an unbalanced water cycle favouring water loss through evapotranspiration and low percolation values. Notably, results suggest that poor soil structure enhances severe agricultural drought conditions, and high curve numbers seem to increase hydrological drought severity. Subbasins in the basin's southern part experienced moderate agricultural and hydrological drought severity. Mild agricultural drought was linked to low evapotranspiration losses and basin capacity to retain water in the soil profile, improving percolation. In turn, moderate hydrological drought severity relates to the subbasins' proximity to the marsh (which acted as a natural control reducing the water yield) and surface runoff contributions to the streamflow. The outcomes of this study demonstrate that the combined effect of parameters with low impact can trigger a drought situation as severe as the one produced by one or two of the most influential hydroclimatic parameters.*

4. In addition, the outcomes of the two MVRTs are rather different, and it would be interesting to discuss the analogies and differences between the two (in spatial patterns, explanatory variables, etc.). In the current version, the two analyses are almost independent from each other. Is the division in 3 macro regions valid for both agricultural and hydrological droughts? Is yes, how it is so given the differences in the trees?

To improve the description of our results and ensure readers' clarity, we will not include the reference to the three regions in the RM. In the RM, we will have a section highlighting similarities and differences between the MVRTs.

*4.3 Comparison of the hydroclimatic parameters influencing the severity of agricultural and hydrological droughts*

*Crucial similarities and differences emerge from contrasting the parameters influencing the severity of droughts and the spatial distribution of the subbasins experiencing severe and mild drought conditions. MVRTs indicate that severe agricultural and hydrological drought conditions occurred in the upper and middle course of the river. Nevertheless, the severe droughts were influenced by different hydroclimatic factors. Severe agricultural drought in the headwater was driven by the interaction between precipitation shortfalls and high potential evapotranspiration (Figure 7a). Conversely, severe hydrological drought condition was solely driven by limited precipitation. It is worth highlighting that the severe hydrological situation extends from the headwater to some subbasins in the middle course (Figure 9a).*

*Downstream, in subbasins located in the middle course, the agricultural and hydrological drought situation was also severe. In this area, droughts' severity was linked to inadequate rainfall partitioning and an unbalanced water cycle that favours water loss through evapotranspiration and low percolation values (Figure 7d, e, f and g, and Figure 9b, c and d). Significantly, agricultural and hydrological droughts in these leaves were more severe than in leaves experiencing precipitation deficits (Figure 7a and Figure 9a). Results suggest that poor soil structure enhances severe agricultural drought conditions (Figure 7e), and high curve numbers seem to increase hydrological drought severity (Figure 9b).*

*MVRTs also showed subbasins experiencing mild agricultural and hydrological drought severity. Overall, these subbasins were in the southern part of the basin. However, for agricultural drought, a few cases were observed in the north of the basin (Figure 7h, i and j). Subbasins presenting mild hydrological drought severity allocate upstream of the Zapatosa marsh (Figure 9g). Moderate agricultural drought severity was linked to low evapotranspiration losses and basin capacity to retain water in the soil profile, improving percolation (Figure*

*7j). In turn, moderate hydrological drought severity relates to the subbasins'*

*proximity to the marsh (which acted as a natural control reducing the water*

*yield) and surface runoff contributions to the streamflow (Figure 9g).*

*Remarkably, some of these subbasins also showed mild agricultural drought*

*conditions (Figure 7i).*

5. Finally, given the focus on drought, I would have expected a validation of the model also in term of drought

   quantities, especially low-flow conditions. The validation of the SWAT model should be expanded to

   highlight reasonable performances during drought conditions, and possibly expanded to soil moisture as

   well.

The authors agree with the referee that given the focus of the study on droughts, it is appropriate to evaluate the

model performance simulating low-flows. In the RM manuscript, we will include the model performance

indicators for the dry season.

*Considering the study focus is on droughts, the model performance*

*simulating low flows was analysed separately. Performance indicators were*

*calculated for the dry season, which lasts from December to March. The*

*intermediate period of precipitation decrease from June to July was also*

*included in this analysis. Table 5 summarises the calibration and validation*

*performance indicators in the dry season. According to the rating guidelines, the*

*model performance simulating low flows is satisfactory (Transactions of the*

*ASABE (American Society of Agricultural and Biological Engineers), 2018).*

**Table 1.** *SWAT model performance simulating low flows.*

| Gauging station | Calibration | | Validation | |
|---|---|---|---|---|
| | NSE | PBIAS [%] | NSE | PBIAS [%] |
| Puente Salguero | 0.65 | -19.4 | 0.53 | -21.3 |
| Puente Carretera | 0.67 | -15.3 | 0.53 | 17.2 |
| Cantaclaro | 0.67 | -3.6 | 0.58 | 16.3 |
| Puente Canoas | 0.55 | -15.7 | 0.60 | -13.5 |

Regarding the comment about expanding the validation to soil moisture, the authors agree with the reviewer that

calibration and validation of the model using soil moisture may contribute to reducing the uncertainty for the

drought analysis; nevertheless, monthly soil moisture data is needed for calibrating and validating the model,

either in-situ measurements, satellite-derived soil moisture, or reanalysis soil moisture, at subbasin level. There

are no in-situ soil moisture measurements in the study area, and the spatial resolution of the available datasets of

satellite-derived soil moisture or reanalysis soil moisture is coarse (0.25°×0.25°). Accordingly, data availability constraints that analysis. In the absence of data to conduct that calibration, good performance simulating streamflow indicates that the model adequately reproduces the land phase of the water cycle in the basin.

**Specific Comments**

1. L12-13. You mention anthropogenic interventions and region's characteristics, but those are factors that are barely included in your analysis. If this is a key point of your study, it should be better reflected in the analysis.

2. L51. "MAY play…" Actually, I have the impression from your results that some of these quantities do not play a major role, at least in your study region.

3. L53-55. Again, you stress the role of human interventions but only marginally included them in the study.

The authors highlight that "the region's characteristics" refer to hydroclimatic parameters recognised as potential drought drivers. We consider that the region's characteristics are adequately reflected in the analysis. The manuscript's introduction presents different hydroclimatic parameters that influence the drought-generating process and the characteristics of droughts. These parameters include soil type, stratigraphy, elevation, slope, vegetation cover, drainage networks, water bodies and groundwater systems. In the methodology section, Table 2 presents the hydroclimatic parameters used in this study as potential drivers of droughts (percolation, surface runoff, groundwater, water yield, sediment yield, curve number, slope and soil type). Comparing the parameters presented in the introduction with the parameters in the methodology confirms that both are in good agreement. Furthermore, the results and discussion section show that most of the parameters included in the analysis influence the drought's severity.

The authors agree with the reviewer that hydroclimatic parameters selected at the first split levels have more influence on droughts than those at lower levels. However, a relevant outcome of this study is that the combined effect of parameters with low impact can trigger a drought situation as severe as the one produced by one or two of the most influential parameters.

Regarding comments 1 and 3, the authors agree that the representation of anthropogenic interventions is limited. Land use change (represented by the CN2) is the only anthropogenic intervention included in the analysis. At the initial stage of the study, the authors asked local and regional authorities about the available information on irrigation systems and groundwater pumping in the area. The authorities confirm that the activities are developed in the region, but there was no consolidated information on these systems' location and operation

characteristics. Accordingly, it was not possible to represent these interventions in the study and evaluate the impact on drought severity.

Although the influence of anthropogenic activities is not widely analysed in this study (due to the lack of data), it is relevent to mention them in the introduction. In that section, we aim to provide an overall picture of all the potential drivers of droughts and various studies have demonstrated that human activities can enhance a drought situation.

4. L76. This is the right place to highlight why a multivariate approach may be needed here.

Indeed, the authors agree with the reviewer that the introduction needs to indicate why a multivariate approach is relevant to this study. As shown in answer to General Comment 2, we will improve the introduction to explicitly present why we opted for this technique and how its capabilities are used and relevant for this work.

5. L87. Please better link this line and figure to the rest of the text reported later (description of the methodology).

To improve the structure of the section and better link Figure 1 to the description of the methodology, we will apply the following changes in the RM. The section title and subtitles will be updated: Section 2 is *Study location and methods*, and the subsections are: *2.1 Case of study and 2.2 Methods. Section 2.2 includes 2.2.1 Hydrological modelling, 2.2.3 Agricultural and hydrological drought analysis and 2.2.3 Multivariate regression tree approach for evaluating the relationships between hydroclimatic characteristics and droughts severity.* Figures 1 and 2 are swapped according to the new section's order.

**2 Study location and methods**

**2.1 Case study**

*Figure 1 presents the Cesar River basin's location, topography and land use. The basin is located between 72º53'W 74º04'W and 10º52'00'N 7º41'00''N latitude (Colombia). It extends for an area of 22,312 km2. The basin's topography is defined in three distinct climatic regions (Universidad del Atlantico, 2014). In the north is La Sierra Nevada de Santa Marta. This sector is characterised by steeply sloped mountains reaching up to 5,700 meters above sea level (masl). The temperature ranges from 3°C to 6°C, and the mean annual precipitation is 1,000 mm. In the east is La Serranía del Perijá. This mountainous area is an extension of the eastern branch of the Andes range. In this sector, the altitude ranges from 1,000 to 2,000 masl. The average temperature is 24°C, and*

*the average annual precipitation varies from 1,000 mm to 2,000 mm. Lastly, the valley of the Cesar River and the Zapatosa marsh are in the west and south of the basin, respectively. The valley is characterised by flat topography and a complex system of marshes formed by the Cesar River floodplains and its confluence with the Magdalena River. The average temperature is 28°C, and the mean annual precipitation is 1,500 mm. At the basin, the annual rainfall pattern presents a dry season from December to April, followed by a rainy season from April to May. In the intermediate period from June to July, precipitation decreases. The main rainfall events occur between August and November.*

*The predominant land use is pasture, followed by agriculture (Universidad del Atlantico, 2014). The primary land use in La Sierra Nevada foothills is pastures for cattle farming. In La Serranía del Perijá, the high altitude areas are covered by forests in very good condition; at the lower altitudes, the principal land use is agriculture, particularly subsistence crops. The Cesar River valley's soils are rich in nutrients, providing favourable conditions for agriculture. The riverbanks are covered by forest with low tree density.*

*The Zapatosa marsh is recognised as one of the most important wetlands in the country, and considering the relevance of this ecosystem, it was declared a Ramsar site in 2018. Nevertheless, the region is threatened by the overexploitation of its forest resources and overfishing. In addition, climate change projections indicate that the basin's temperature may increase by 2.7°C, and precipitation may reduce by ten percent by 2070 (Universidad del Magdalena et al., 2017). Accordingly, multiple initiatives are oriented to improve water management and create resilience to hydroclimatic extremes (Ministerio de Ambiente y Desarrollo Sostenible (Colombia), 2015).*

[Figure]

**Figure 1** *Cesar River basin: a) topography and b) land use.*

**2.2 Methods**

Figure 2 illustrates the three steps methodology applied in this study. Section 2.2.1 describes the hydrological modelling, and 2.2.2 the drought analysis. Section 2.2.3 presents the application of the MVRT technique.

[Figure]

*Figure 2 Flow chart of the methodology*

6.  L91. Please mark these three sub-regions in the map for the people not familiar with the region.

See answer to Specific Comment 5, Figure 1a

7.  L105. You mention pasture here, but no "pasture" class is reported in the Figure. Please align the text with the figure.

There was an error in the figure. The category "GRASS" is actually "PASTURE". The authors apologize for the mistake. The figure is corrected in the RM. See answer to Specific Comment 5, Figure 1b.

8.  L115. Reference?

The reference will be included in the RM.

> *Accordingly, multiple initiatives are oriented to improve water management and create resilience to hydroclimatic extremes (Ministerio de Ambiente y Desarrollo Sostenible (Colombia), 2015).*

9.  L121. I would link this sentence to the next.

The sentence will be link to next in the RM.

*A SWAT model with an ArcSWAT extension was used to develop the Cesar River basin model used in this research. SWAT is a continuous-time, semi-distributed, process-based river watershed-scale model developed by The Agricultural Research Service of the United States Department of Agriculture (ARS-USDA). The model is designed to simulate the quality and quantity of surface and groundwater and predict the environmental impacts of land use, land management and climate change (Neitsch et al., 2011).*

10. L142. I assume that CN2 is the initial CN for soil moisture condition 2, since the actual CN is a variable. Please clarify.

The authors apologize for the mistake. Indeed, CN2 is the initial SCS runoff curve number for moisture condition II. The CN2 definition will be corrected in the RM.

*CN2 (initial SCS runoff curve number for moisture condition II).*

11. L143. No calibration on the Manning factor?

The manning factor was used in the calibration of the model. It was not included in the *Section Model Calibration and Validation* by mistake. The parameter will be included in the RM.

*Based on expert judgment and the available literature (Arnold et al., 2012; Transactions of the ASABE (American Society of Agricultural and Biological Engineers), 2018), the following SWAT parameters were used in the calibration and validation process: baseflow alpha factor (ALPHA_BF), effective hydraulic conductivity in main channel alluvium (CH_K), Manning's value for the main channel (CH_N2), SCS runoff curve number for moisture condition II (CN2), soil evaporation compensation factor (ESCO), groundwater delay (GW_DELAY), threshold depth of water in the shallow aquifer required for return flow to occur (GWQMN), deep aquifer percolation fraction (RCHRG_DP), threshold depth of water in the shallow aquifer for percolation to the deep aquifer to occur (REVAPMN) and available water capacity of the soil layer (SOL_AWC).*

12. L152. Since your focus is on hydrological drought, I suggest adding some evaluation metrics focused specifically on low flow. It is a well-known issue that NSE may return high values even when low flow conditions are not well represented due to a good matching of flood values. Also, given the relevance of soil

moisture in your study, some kind of validation/evaluation of the performances in terms of soil moisture is needed.

See the answer to General Comment 5.

13. L162. No details are provided on the soil profile. Is it a single soil layer? How depth? Please clarify.

More details on the soil profile will be provided in the RM.

> *According to the soil profiles and the secondary information used to elaborate the soil map, three soil layers were identified in the Cesar River basin. The soil layers' thickness (vertical distance from the surface) varies. The first layer reaches up to 350 mm, the second 1000 mm, and the third 1500 mm.*

14. L190. What is the reference period? 1987-2018? Clarify.

The reference period will be included in the RM

> *To this aim, the monthly simulated streamflow at each subbasin in the analysis period (1987 to 2018) was fitted to the gamma probability distribution function.*

15. L194. This sentence is not clear to me. Does the 30% refer to the total area of the basin, meaning that a minimum number of sub-basins (covering at least 30% of the total area) need to be in moderate drought?

Indeed, the reviewer's description of the sentence is correct. The sentence will be updated in the RM to prevent the reader's confusion,

> *SMDI and SSI were calculated monthly using the simulated soil water and streamflow values at each subbasin. The drought events during the period of analysis were then identified. A drought (agricultural or hydrological) event was assumed to occur in the basin when a number of subbasins (covering at least 30 % of the basin's total area) were in a moderate drought state for at least two consecutive time steps (i.e. in this study month). According to the spatial and temporal thresholds, a drought event began when both conditions were met and continued until one of them failed to be met. We set a minimum spatial extension threshold because droughts typically extend regionally (Sheffield & Wood, 2011b). By setting the temporal threshold, we avoided identifying periods of water shortage or scarcity as drought events.*

16. L196. You mention short periods, but I do not see any constrains on the duration of an event. Please better clarify the definition of drought event used here (i.e. starts when at least 30%...., and end when…). Also, if any kind of spatial or temporal pooling is performed please clarify.

The authors agree with the reviewer that the paragraph fails to adequately describe the temporal threshold used to identify droughts (agricultural and hydrological). As indicated in the answer to Comment 15, this paragraph will be improved in the RM.

17. L198. The PCA has a very limited role in this study. I suggest reevaluating the need to include this section and this analysis in the study.

The authors thank the reviewer for questioning the relevance of the PCA results. Before applying the MVRT, we used PCA to explore the dataset of explanatory variables. Our goal was to identify the most influential parameters of the dataset and discard non-influential parameters. The PCA results showed that all the parameters considerably influenced at least one of the PC retained; thus, for the MVRT technique, we used all the parameters initially selected. Reviewer's Specific Comments 17 and 31 make us reevaluate the relevance of the PCA results since the method did not produce changes in the set of explanatory variables. We concluded that using PCA was a good strategy for explanatory variables exploration, but the outcome of the analysis is not relevant to the objective of this study. Accordingly, we will remove the PCA analysis in the RM.

18. L216-221. This a rather generic description of the methodology. Please contextualize the method to your study. This section should answer the questions: What is a predictand (see comment below)? Why are they multiple? Why do you need MVRT instead of simple RT?

In the RM, we will update the paragraph to indicate the response variables explicitly and include a new paragraph to contextualize the technique in the study.

> *MVRT is an extension of a regression tree (Breiman, 2001), but it differs in that it allows for multiple outputs (see De'ath, 2002). It recursively splits a quantitative response variable (predictand, output) controlled by a set of numerical or categorical explanatory variables (predictors, input). The technique approach yields a set of non-linear models, each a piece-wise linear regression model (of zero order). An MVRT result is a tree whose terminal groups (leaves) of instances (input-output vectors) comprise subsets of samples selected to minimise the within-group sums of squares. Each successive split is given by a threshold value of the explanatory variables (Borcard et al., 2018).*

*MVRT is applied to dataset exploration, description and prediction (De'ath, 2002). In this study, the explanatory variables are the hydroclimatic parameters at each subbasin, represented by the average value of each parameter during the analysis period (1987 to 2018). The response variables are the number of months observed at each drought severity category (the drought indices give categories). The analyses for agricultural and hydrological droughts were conducted separately; thus, two MVRTs were obtained.*

*Four technique attributes are relevant to this study. First, MVRT can capture the non-linear interactions between the parameters influencing droughts and their severity. Second, the technique can handle numerical and categorical hydroclimatic parameters influencing drought severity (explanatory variables). Third, MVRT's capability to handle multiple outputs allowed us to evaluate the influence of the hydroclimatic parameters on moderate, severe and extreme drought conditions simultaneously (response variables). The drought indicators give these three categories to represent the drought severity. Simultaneous analysis of different drought categories provides a comprehensive understanding of the drought-generating process and the factors influencing severe (or mild) drought conditions. Fourth, MVRT results can be easily visualised and interpreted. The resulting tree structure provides a clear representation of the relationship between the drivers of droughts and the severity of agricultural and hydrological droughts.*

19. L223. The response variables need to be better identified here. The generic "drought severity" used here leaves a lot of questions to the readers. Is it a time series of event severity for each sub-basin? A time series over the entire basin? Just a single value (average or similar)? This need to be clarified here (and eventually detailed later) in order to justify the multivariate dimension of the problem.

In the RM, we will update the paragraph to indicate the response variables explicitly (Se answer to Specific Comment 18). In addition, we will improve the description of the set of response variables.

**Set of response variables**

*We used the drought analysis outcomes to define the response variables (Table 3). Following the methodology presented in 2.3, we identified the*

*agricultural and hydrological drought events during the analysed period. After identifying the drought events, we counted the months for each drought severity category at each subbasin. The observed months for each one of the three drought categories were used as response variables. The analyses for agricultural and hydrological droughts were conducted separately; thus, two sets of response variables were obtained.*

20. L223-229. Related to the previous point. Here you first give the impression that agrological and hydrological drought severities are the two "multivariate" variables. Then, you clarify that the two are studied separately, leaving the question on what is the "multivariate" variable then. This can be only indirectly inferred from the results section, but it must be clearly stated already here.

We will update the introduction and methodology in the RM to define the response variables clearly. See the answer to General Comment 1 and Specific Comments 18 and 19.

21. Since section 2.5 is supposed to be the main methodology section, you need to significantly extend this section and add all the needed clarifications. Also link to the flow chart should me reported here.

The following we summarize the changes we will apply to the Section 2.5 (Section 2.2.3 in the RM).

- We will define the sets of explanatory and response variables. See answer to Specific Comment 18.
- We will include a new paragraph to properly contextualize the MVRT technique in the study and highlight the attributes relevant for this study. See answer to Specific Comment 18.
- We will improve the description of the set of explanatory variables. See answer to Specific Comment 26.
- We will improve the description of the set of response variables. See answer to Specific Comment 19.

22. L234. Again, similarly to the previous section, it is not clear what average means here. Is it a spatial average? A temporal average? Do you use time series of spatial-average values for each sub-basin or just a single value. This can be indirectly inferred from the results, but it should be made clear here.

In this study, the explanatory variables are the hydroclimatic parameters at each subbasin, represented by the average value of each parameter during the analysis period (1987 to 2018). The introduction and the methodology will be updated in the RM to improve the description of the explanatory variables. See answer to General Comment 1, and Specific Comments 18 and 26.

23. L240. Following the previous comment: so, do you have 3 values for each sub-basin as response variables? Are then the frequency in the 3 categories the "multivariate"?

Indeed, the drought severity categories were the multivariate response. We agree with the reviewer that the manuscript needed more clarity about the application of the MVRT technique and why the drought severity was considered a multivariate output. To improve the description of the methodology, we will apply the changes presented in the answers to General Comment 1 and Specific Comments 18 and 19.

24. L251.Which two groups?

The sentence will be rewritten in the RM.

> *The data partitioning consisted of three steps. First, for each explanatory variable were generated all possible partitions of the sites (subbasins) into two groups.*

25. L276. This sentence seems to imply that two methods are used to choose the size, which is in contrast with the next sentence. Please clarify.

The authors apologize for the mistake. In the RM, we will update the paragraph to indicate the approach we used to choose the tree size. In the RM, we will not include information on the method we did not use.

> *To choose the tree size that retained the most descriptive partition, we used the approach suggested by De'ath (2002). According to the author, a tree with the smallest CVRE offers the best explanatory power and interpretability combination. Once the tree was built, the proportion of explained variance (EV) was calculated as 1- ⟦RE⟧_tree (tree relative error) (Cannon, 2012).*

26. L294-295. This should be clarified in the methodology and not here.

In the RM, we will update the methodology description to indicate the explanatory variables explicitly; see the answer to Specific Comment 18. In addition, we will improve the description of the set of explanatory variables.

> ***Set of explanatory variables***
>
> *To select the set of explanatory variables, we used the outcomes of previous studies on governing drivers of droughts (Sheffield & Wood, 2011a; Zhang et al., 2022). Table 2 describes the eleven parameters selected as the potential drivers of droughts. The used values correspond to the parameters' average in the analysis period (1987 to 2018). The averages were computed using the SWAT model outputs at each subbasin. We used the dominant category at each subbasin for the curve number, the slope, and the soil type (categorical variables).*

27. I am not 100% sure that the data reported in sections 3.1 and 3.2 are results of the study. They may fit better in the "Data and method section", since they do not bring much to the discussion on the use of MVRT.

We thank the reviewer for the suggestion but prefer to maintain Sections 3.1 and 3.2 in the results. We consider that model calibration results and simulated hydroclimatic parameters are results of this study and fit best the in that section.

28. Section 3.3. It is not clear how these 6 events are derived from the methodology described in section 2.3. There, only a minimum fraction of the area in the sub-basin is defined, and nothing is said on duration/continuity of an event. Is there any constrain on duration? Did you remove the minor events? Please clarify.

Indeed, the minor events were not included in this analysis. The authors agree with the reviewer that the methodology fails to provide details on how the drought events identified during the simulation period were derived from the methodology. In the RM, we will adequately describe the temporal threshold used to identify droughts (agricultural and hydrological). See answer to Specific Comment 15.

29. Table 5. There is a typo on event 4 (IV).

The authors apologize for the mistake. The typo error will be corrected in the RM.

*Table 2. Agricultural and hydrological droughts during the period of analysis*

| Event | Agricultural droughts | | Hydrological droughts | |
|---|---|---|---|---|
| | Date | Duration [months] | Date | Duration [months] |
| I | May 1991 – Jun 1992 | 13 | Apr 1991 – May 1992 | 14 |
| II | Jun 1997 – April 1998 | 11 | Apr 1997 – Feb 1998 | 11 |
| III | Jun 2001 – Aug 2001 | 3 | May 2001 – Jun 2001 | 2 |
| IV | Oct 2009 – Jan 2010 | 4 | Sep 2009 – Nov 2009 | 3 |
| V | Jun 2014 – Aug 2014 | 3 | Jun 2014 – Jul 2014 | 2 |
| VI | May 2015 – Jul 2016 | 15 | Apr 2015 – Apr 2016 | 13 |

30. L310. This should be made clear much sooner in the text, and clearly highlight that the multivariate of the MVRT is referring to the 3 categories.

We improved the description of the response variables in the introduction and the methodology. See answers to General Comment 1 and Specific Comments 18 and 19.

31. 3.4 As a said before, this has very marginal impacts on the analysis. At the end, you included all the variables in the MVRT analysis, but some of them where not actually used in the final trees (and some very marginally). What does this say on the usefulness of the PCA in this case? I suggest removing this part and

focus more on analyzing the variables used in the two final MVRTs and the differences between the two trees.

We will not include the PCA analysis in the RM. See answer to Specific Comment 17.

32. L334-342. Was an analysis on a limited number of explanatory variables also performed? As an example: how different are the results if only ET and PREC are used? Are some leaves really necessary? As an example, h) and i) are separated only at the end and based on WYLD, but the plots in Fig. 9 are quite similar. Are all 12 leaves relevant, considering that you then discuss only 3 macro regions? Some leaves are also quite small (just 2 basins for b) and k) for instance); if these are relevant, then they shouldn't be grouped in the 3 macro regions in the discussion and conclusion sections.

33. The same considerations are true for the results on hydrological drought.

Answer to Specific Comments 32 and 33

To build the MVRT, All the explanatory variables are used to recursively generate the partitions resulting in the three's final leaves. We did not perform the analysis using fewer explanatory variables because it may result in MVRTs with lower explanatory power. Including multiple explanatory allows the technique to produce the partitions that maximize the explanatory power of the three (maximize the proportion of the explained variance). In addition, before applying the MVRT technique, we used PCA to explore the dataset of explanatory variables (As explained in answer to the Specific Comment 17). Our goal was to identify the most influential parameters of the dataset and discard non-influential parameters. The PCA results showed that all the parameters considerably influenced at least one PC retained. It indicates that all the parameters included in the set of explanatory variables are relevant to this study. Accordingly, for the MVRT technique, we used all the parameters initially selected. It is worth mentioning that we chose the threes with the lowest CVRE. According to De'ath (2002), these trees offer the best explanatory power and interpretability combination.

Regarding the importance of all the leaves retained, we consider that all leaves provide relevant information on the different hydroclimatic parameters influencing droughts' severity. Figures 7, 8, 9 and 10 show that the severity of doughs (agricultural and hydrological) is different in each leaf and influenced by different parameters.

Regarding the three regions mentioned in the abstract and the conclusion, the authors realized that the statement does not properly summarize the study results. It is more precise to say that we identify different sets of parameters that govern drought severity in the basin (See answer to General Comment 3). The RM will not include the paragraphs referring to these three regions.

34. L424-426. This should be better supported by some synthetic results, rather than leaving the extraction of meaningful information to the readers.

The reviewer refers to the first paragraph of Section 4.1. In that paragraph, we summarize the information presented in Section 3.5.1 and link the tree description (results) with the discussion. In addition, in the following paragraphs of Section 4.1, we provide a detailed discussion about the parameters influencing the droughts and the severity in each leaf. We consider that the reviewer's comment may arise from the expression "subbasins most exposed to agricultural droughts". We will update the sentence in the RM to ensure readers' clarity.

> *The left branch of the MVRT clusters the subbasins exposed to severe agricultural drought (Figure 8a, e, f, g). Conversely, the right branch of the MVRT clusters the subbasins experiencing moderate agricultural drought severity. The subbasins in leaves h, i and j predominately experienced months in the moderate drought category (Figure 8i, j, k).*

35. L514-521. This explanation is a little lacking, since the explanatory variables and the targets are both derived from the same modelling framework. I am wondering if some variables that are relevant for the hydrological drought were not included in the analysis.

The authors agree with the reviewer that the three's explanatory power may also be linked to relevant parameters for the hydrological drought not included in the analysis. In the last part of this paragraph, we refer to this limitation.

> *Conversely, the explanatory power of the tree built for hydrological drought is not very high (EV = 0.48). This may be related to the inaccurate representation of groundwater contribution to the streamflow. Streams depend significantly on groundwater during droughts to maintain flow; nevertheless, groundwater contribution to the streamflow was not included as a key drought driver in the MVRT, although it was in the list of explanatory variables. It is possible that the model's simplifications for the simulation of groundwater flow and storage did not adequately represent the groundwater contribution to the streamflow (Molina-Navarro et al., 2019). The lack of adequate information about this relevant factor hydrological drought may have compromised the MVRT's accuracy. Unexplained variability may also link to factors that influence hydrological drought but were not considered in the dataset of explanatory*

*variables (e.g. abstractions such as water for irrigation, industry or human consumption).*

In addition, in the limitations of the study we mentioned that parameters influencing droughts were not included in this analysis.

*Additionally, there is still a need to better represent anthropogenic interventions (and other relevant parameters influencing droughts) in the set of explanatory variables (e.g. abstractions such as water for irrigation, industry or human consumption, groundwater pumping).*

36. L523-529. Even if 9/11 were included, some have a very limited role and appears only in hydrological drought. This discussion needs to be expanded, and a more in-depth comparisons of the two trees need to be added.

Regarding the first part of the comment, the authors considered that the relevance of parameters is not given by the number of times it was selected at different split levels in one or both threes. We evaluated a parameter's relevance by contrasting the drought's severity in the different leaves. For instance, in the MVRT for hydrological droughts, precipitation and water yield are alike for leaves g and h. Surface runoff is selected at the third split level, dividing the subbasins into two groups. Figure 10 shows that in the leave g, the median of months in the moderate drought category is ten, while at h is eighteen. Furthermore, each leave shows different number of months in severe and extreme drought categories. Although surface runoff was used at the third split level (and not included in the MVRT for agricultural droughts), results show that the parameter is utilized to divide subbasins presenting different agricultural drought severity. Similar analysis can be developed for sediment yield (Figure 8 leaves e and f) and curve number (Figure 8 leaves k and l, and Figure 10 leaves b and c).

About the second part of the comment, the comparison of the two trees was included in the RM. See answer to General Comment 4.

37. L542. Is this true also for hydrological drought?

In the line indicated by the reviewer both types of droughts are mentioned.

*This study applied the MVRT technique, which served as an explanatory approach (in the line of 'explanatory AI') to assess the relationship between a subbasin's hydroclimatic characteristics (i.e. explanatory variables) and the severity categories of agricultural and hydrological drought (i.e. response*

*variables). The results show that the machine learning technique successfully identified drought severity's primary drivers and critical thresholds. The MVRT also provided valuable information on which parameters can contribute to reducing agricultural and hydrological drought severity.*

38. L546-547. This subdivision in three sub-areas is never highlighted in the results, and it is not evident how and why these three sub-areas are the same for agricultural and hydrological droughts, given that different trees and explanatory variables are identified.

Agreed. See answer to General Comments 3 and 4.

---

## Author Comment (AC2)

**Response to Reviewer 2 Comments**

**Manuscript title:** Multivariate regression trees as an 'explainable machine learning' approach to exploring relationships between hydroclimatic characteristics and agricultural and hydrological drought severity

**Author's general response:**

The authors would like to thank the reviewer for thoroughly reviewing the manuscript. Your insightful and specific comments helped us to improve the manuscript's scientific quality. We are particularly grateful for your meaningful observations on the application of MVRT to this particular study and for providing constructive feedback on the manuscript's overall readability. We will apply multiple changes to incorporate the reviewer's suggestions. In the following, you will find the answers to the general and specific comments. Some of them required a particular action or change in the manuscript. The changes we will apply in the Revised Manuscript (RM) are in italics.

**General **Comments**

1. I am wondering why the authors used SMDI and SSI to identify soil moisture/agriculture and streamflow droughts, respectively. These are two different methods. Why don't the authors use the Standardized Soil Moisture Index (SSMI) in order to have a comparable method with SSI since both are the standardized indices. I suggest to write a clarification of why the authors decide to use SMDI instead of SSMI.

We evaluated different drought indices to select the most appropriate for this study. The SSMI is an agricultural drought index derived from daily satellite data and applicable for short-term agricultural drought monitoring (and prediction) across large areas. In validating the SSMI, the index authors found a moderate correlation with the Palmer Drought Severity Index (PDSI), an effective index in determining long-term droughts. We concluded that SSMI was unsuitable for this study for two main reasons. First, the index is developed for short-term drought monitoring. Our study focuses on past drought events, particularly severe, long-lasting droughts. Second, there is no previous assessment of the index performance using simulated soil moisture as the input parameter.

On the other hand, SMDI was developed to use simulated soil moisture with SWAT as the input parameter. In addition, in the SMDI validation, the index authors found a good correlation with the PDSI. We agree with the reviver that SMDI and SSI apply different methods to estimate agricultural drought severity. Nevertheless, we considered that difference had no implications for this study. Despite the method to estimate the drought severity, both indices successfully represented the past drought events in the region. Additionally, they allowed

us to determine the number of months for each drought category (moderate, severe, extreme) at each subbasin during the analysis period. That information was used to define the set of response variables to apply the MVRT technique.

2. Another I do not get the importance of PCA analysis in your study. Here the authors used the PCA to further confirm the key drivers of droughts obtained from the MVRT. However, more explanation in the text about the PCA results and how these confirm the MVRT results is lacking. For example, from the 11 variables, which variables have the higher explained variances, and how to read the loading factors in the sense of what positive and negative signs mean? From the MVRT, I can see that ET, precipitation, and percolation are key drivers for agriculture drought (correct me if I am wrong) and the key drivers for streamflow drought are precipitation and water yield.

The authors thank the reviewer for questioning the relevance of the PCA results. Before applying the MVRT, we used PCA to explore the dataset of explanatory variables. Our goal was to identify the most influential parameters of the dataset and discard non-influential parameters. The PCA results showed that all the parameters considerably influenced at least one of the PC retained; thus, for the MVRT technique, we used all the parameters initially selected. Reviewer's comment makes us reevaluate the PCA results' relevance since the technique application does not produce changes in the set of explanatory variables. We conclude that using PCA is a good strategy for explanatory variables exploration, but the outcome of the analysis is not relevant to the objective of this study. Accordingly, we will remove the PCA analysis in the RM.

Although the analysis is not included in the RM, we consider it essential to answer the questions posed.

Regarding the first question, when applying PCA, explained variance indicates how much of the total variance in the dataset is "explained" by each principal component. In this case, the first component explained 36% of the total variance in the set of explanatory variables, the second 29% and the third 12%. The cumulative explained variance of the three principal components retained was 77%.

Regarding the second question, loading factors indicate which individual variables contribute the most to the principal components. The sign (+ or -) indicates whether a variable and a principal component are positively or negatively correlated. In this case, the first component was heavily influenced by precipitation, potential evapotranspiration, evapotranspiration, percolation, curve number and slope.

The reviewer's interpretation of the MVRTs is correct, and it is possible to confirm that all these individual variables heavily influenced at least one of the three components retained.

3. An explanation of why the authors used different CVRE, relative error, and EV values for agriculture and streamflow droughts is needed.

We highlight that CVRE, relative error, and EV values are given by the MVRTs explanatory power. The explanatory power refers to the proportion of the variance explained by the tree. Our values are different because the trees for agricultural and hydrological droughts have distinct explanatory power. Section 2.2.3 (Building the MVRT) presents the parameters definitions and the equations to calculate them. In Section 4.3 (Accuracy of the MVRTs), we discuss the accuracy of the trees and present possible causes of the different EVs obtained. The original sections mentioned are included below:

**Cross-validation of the partitions and tree pruning**

*This cross-validation process was repeated several times for consecutive and independent divisions of the data into test groups. For each group, the mean and standard deviation of all CVRE were computed. The CVRE varied from 0 for perfect predictors to close to 1 for poor predictors (for large errors, CVRE may reach +∞). Among the mvpart function arguments, we used ten cross-validation groups (function argument, xval = 10) and 100 iterations (function argument xmult = 100). The tree was selected using interactive cross-validation (function argument xv = 'pick').*

*To choose the size of the tree that retained the most descriptive partition, we used the approach suggested by De'ath (2002). According to the author a tree with the smallest CVRE offers the best combination of explanatory power and interpretability. Once the tree was built, the proportion of explained variance (EV) was calculated as 1- ⟦RE⟧ _tree (tree relative error) (Cannon, 2012).*

**4.4 Accuracy of the MVRTs**

*The high EV (0.81) value reflects the good explanatory power of the tree built for agricultural drought. This confirms that the selected explanatory variables significantly influence the severity of agricultural drought.*

*Conversely, the explanatory power of the tree built for hydrological drought is not very high (EV = 0.48). This may be related to the inaccurate representation of groundwater contribution to the streamflow. Streams depend significantly on groundwater during droughts to maintain flow; nevertheless, groundwater*

*contribution to the streamflow was not included as a key drought driver in the MVRT, although it was in the list of explanatory variables. It is possible that the model's simplifications for the simulation of groundwater flow and storage did not adequately represent the groundwater contribution to the streamflow (Molina-Navarro et al., 2019). The lack of adequate information about this relevant factor hydrological drought may have compromised the MVRT's accuracy. Unexplained variability may also link to factors that influence hydrological drought but were not considered in the dataset of explanatory variables (e.g. abstractions such as water for irrigation, industry or human consumption).*

**Specific Comments**

1.  P1L17: Maybe add the word "such as" -> ….model outputs, such as soil moisture and streamflow….

The authors thank the reviewer for the suggestion, but, after reflecting on it, we consider that the adverb "such us" is not suitable for this sentence, because we are not introducing examples. Instead, soil moisture and streamflow are the only two model outputs we used to calculate the drought indices.

2.  P1L26: the authors may replace the word "brought on" with "caused"

We like the suggestion. The word will be replaced in the RM.

3.  P2L46-50: Here the authors describe drought propagation. I suggest stating this clearly thus the readers understand what is drought propagation. Moreover, the authors also used the term propagation a few times in the next paragraph. The authors may also add a drought propagation study by Van Loon et al. (2012).

We like the suggestion. In the RM, we will indicate that we refer to drought propagation and will include the reference.

> *Remarkable progress has been achieved in understanding drought propagation through the hydrological cycle (Van Loon et al., 2012).*

4.  P3L66-68: Write the references (two studies) directly after the sentence.

In the RM, we will update the paragraph as shown below. Editor's suggestion will be included.

> *We have found two studies that employ machine learning to analyse the non-linear relationship between climate and basin processes and droughts (Konapala & Mishra, 2020; Valiya Veettil & Mishra, 2020).*

5.  P3L79-80: Same, write such as or the authors may re-write it as: "Soil moisture and streamflow obtained from the SWAT model are used to……."

In the RM, we will update the paragraph as shown below. Editor's suggestion will be included.

> *To understand the relationship between the drivers of droughts and the individual categories of agricultural and hydrological droughts severity, this study employs a methodology that consists of three steps. The first is hydrological modelling. We used Soil Water Assessment Tool (SWAT) to simulate the hydroclimatic parameters required for analysing droughts and applying the MVRT approach. The Second is the analysis of droughts. SWAT outputs, soil moisture and streamflow, are used to calculate the drought indices Soil Moisture Deficit Index (SMDI) and the Standardized Stream Flow Index (SSI). Drought indices were utilised to identify the agricultural and hydrological drought events during the period of analysis and describe their severity. Finally, the MVRT approach is applied to assess the relationship between hydroclimatic characteristics (represented by the simulated parameters at each subbasin, see Table 2) and droughts severity categories (represented by the observed number of months for each drought severity category at each subbasin, see Table 3). The analyses for agricultural and hydrological droughts were conducted separately; thus, two MVRTs were obtained. A concrete application of this methodology is developed in the Cesar River basin (Colombia, South America).*

6.  P3L80-81: The authors mention "other simulated hydroclimatic parameters…." -> mention them already.

The authors thank the reviewer for the suggestion; however, we consider mentioning all the parameters (11 in total) can affect the readability of the paragraph. Please note that they are already presented in Figure 2 (Flowchart of the methodology), and Table 2 explicitly describes each one of the hydroclimatic parameters used as explanatory variables. In the RM, we will refer to Table 2 (See answer to Specific Comment 5).

7.  P3L86: I suggest restructuring section 2. Section 2 will be Study location and methods and thus section 2.1 will be study location and section 2.2 will be methods. Swap figures 1 and 2 accordingly. In the present form, the authors mention first the flowchart describing the data and method but then no explanation is followed. Study location is placed after this 1 sentence about data and method, and then section 2.2 back to method again.

The authors thank the reviewer for the suggestion since it improves the structure of the section. We will apply the following changes in the RM. The section title and subtitles will be updated. Section 2 is *Study location and methods*, and the subsections are 2.1 Case of study and 2.2 Methods. Section 2.2 includes *2.2.1 Hydrological modelling, 2.2.3 Agricultural and hydrological drought analysis* and *2.2.3 Multivariate regression tree approach for evaluating the relationships between hydroclimatic characteristics and droughts severity*. Figures 1 and 2 are swapped according to the new section's order. The relevant sections will read as follows:

**2 Study location and methods**

**2.1 Case study**

Figure 1 presents the Cesar River basin's location, topography and land use. The basin is located between 72º53'W 74º04'W and 10º52'00'N 7º41'00''N latitude (Colombia). It extends for an area of 22,312 km2. The basin's topography is defined in three distinct climatic regions (Universidad del Atlantico, 2014). In the north is La Sierra Nevada de Santa Marta. This sector is characterised by steeply sloped mountains reaching up to 5,700 meters above sea level (masl). The temperature ranges from 3°C to 6°C, and the mean annual precipitation is 1,000 mm. In the east is La Serranía del Perijá. This mountainous area is an extension of the eastern branch of the Andes range. In this sector, the altitude ranges from 1,000 to 2,000 masl. The average temperature is 24°C, and the average annual precipitation varies from 1,000 mm to 2,000 mm. Lastly, the valley of the Cesar River and the Zapatosa marsh are in the west and south of the basin, respectively. The valley is characterised by flat topography and a complex system of marshes formed by the Cesar River floodplains and its confluence with the Magdalena River. The average temperature is 28°C, and the mean annual precipitation is 1,500 mm. At the basin, the annual rainfall pattern presents a dry season from December to April, followed by a rainy season from April to May. In the intermediate period from June to July, precipitation decreases. The main rainfall events occur between August and November.

The predominant land use is pasture, followed by agriculture (Universidad del Atlantico, 2014). The primary land use in La Sierra Nevada foothills is pastures for cattle farming. In La Serranía del Perijá, the high altitude areas are

*covered by forests in very good condition; at the lower altitudes, the principal land use is agriculture, particularly subsistence crops. The Cesar River valley's soils are rich in nutrients, providing favourable conditions for agriculture. The riverbanks are covered by forest with low tree density.*

*The Zapatosa marsh is recognised as one of the most important wetlands in the country, and considering the relevance of this ecosystem, it was declared a Ramsar site in 2018. Nevertheless, the region is threatened by the overexploitation of its forest resources and overfishing. In addition, climate change projections indicate that the basin's temperature may increase by 2.7°C, and precipitation may reduce by ten percent by 2070 (Universidad del Magdalena et al., 2017). Accordingly, multiple initiatives are oriented to improve water management and create resilience to hydroclimatic extremes (Ministerio de Ambiente y Desarrollo Sostenible (Colombia), 2015).*

[Figure]

***Figure 1*** *Cesar River basin: a) topography and b) land use.*

**2.2 Methods**

*Figure 2 illustrates the three steps methodology applied in this study. Section 2.2.1 describes the hydrological modelling, and 2.2.2 the drought analysis. Section 2.2.3 presents the application of the MVRT technique.*

[Figure]

*Figure 2 Flow chart of the methodology*

8. P5L117: Figure 2. Please label this figure into 2a and 2b and refer these figures in the text above. I also strongly suggest the authors change the color label for Figure 2b. Using red color for water bodies, blue for grass, and black color for forest are not common. Change the color codes into the most commonly used colors to represent the land use.

We will label Figure 2 as suggested and change the color map into the most commonly used colors to represent land use. See answer to Specific Comment 7.

9. P6L143: Maybe reverse the abbreviations and full names. I suggest to write the full name first and then the abbreviation.

The new text will read as follows:

> *Based on expert judgment and the available literature (Arnold et al., 2012; Transactions of the ASABE (American Society of Agricultural and Biological Engineers), 2018), the following SWAT parameters were used in the calibration*

*and validation process: baseflow alpha factor (ALPHA_BF), effective hydraulic conductivity in main channel alluvium (CH_K), SCS runoff curve number for moisture condition II (CN2), soil evaporation compensation factor (ESCO), groundwater delay (GW_DELAY), threshold depth of water in the shallow aquifer required for return flow to occur (GWQMN), deep aquifer percolation fraction (RCHRG_DP), threshold depth of water in the shallow aquifer for percolation to the deep aquifer to occur (REVAPMN) and available water capacity of the soil layer (SOL_AWC).*

10. P8L195: Please write the minimum threshold. Is it 30%?

Indeed, the threshold is 30%. The paragraph will be updated in the RM.

*SMDI and SSI were calculated monthly using the simulated soil water and streamflow values at each subbasin. The drought events during the period of analysis were then identified. A drought (agricultural or hydrological) event was assumed to occur in the basin when a number of subbasins (covering at least 30 % of the basin's total area) were in a moderate drought state for at least two consecutive time steps (i.e. in this study month).*

11. P9L198: PCA analysis. Please describe in this section that the authors only used the first order until the third order only.

Please see answer to General Comment 2.

12. P9L215: MVRT method. Some explanations why the authors only used this single method are encouraged.

We agree with the reviewer that the introduction and methodology do not explicitly present the reasons for choosing the MVRT approach and how the technique capabilities are exploited in this study. Accordingly, two changes will be included in the RM. First, we will update the introduction indicating that drought severity categories (moderate, severe and extreme) are the three response variables modeled with the MVRT and summarising the technique capabilities relevant to the study, as shown below: We present the updated version of the introduction (There are no changes in the first two paragraphs of the 
[revised manuscript text omitted]

Second, in the methodology, we will update the introductory paragraph of the Section 2.2.3 and will include a paragraph describing the reasons considered in the selection of the technique, as follows:

*MVRT is an extension of a regression tree (Breiman, 2001), but it differs in that it allows for multiple outputs (see De'ath, 2002). It allows the recursive split of a quantitative response variable (predictand, output) controlled by a set of numerical or categorical explanatory variables (predictors, input). The technique approach yields a set of non-linear models, each a piece-wise linear regression model (of zero order). An MVRT result is a tree whose terminal groups (leaves) of instances (input-output vectors) comprise subsets of instances selected to minimise the within-group sums of squares. Each successive split is given by a threshold value of the explanatory variables (Borcard et al., 2018). MVRT applies to dataset exploration, description and prediction (De'ath, 2002). In this study, explanatory variables are the hydroclimatic parameters at each subbasin, represented by the average value of each parameter during the analysis period (1987 to 2018). The number of months observed at each drought severity category (Categories are given by the drought indices) are the response variables. The analyses for agricultural and hydrological droughts were conducted separately; thus, two MVRTs were obtained.*

*Four technique attributes are relevant to this study. First, MVRT can capture the non-linear interactions between the parameters influencing droughts and their severity. Second, the technique can handle numerical and categorical hydroclimatic parameters influencing drought severity (explanatory variables).*

*Third, MVRT's capability to handle multiple outputs allowed us to evaluate the influence of the hydroclimatic parameters on moderate, severe and extreme drought conditions simultaneously (response variables). The drought indicators give these three categories to represent the drought severity. Simultaneous analysis of different drought categories provides a comprehensive understanding of the drought-generating process and the factors influencing severe (or mild) drought conditions. Fourth, MVRT results can be easily visualised and interpreted. The resulting tree structure provides a clear representation of the relationship between the drivers of droughts and the severity of agricultural and hydrological droughts.*

13. P10L234: value -> values

We regret this mistake. The error will be corrected in the RM, as shown below:

*To select the set of explanatory variables, we used the outcomes of previous studies on governing drivers of droughts (Sheffield & Wood, 2011a; Zhang et al., 2022). Table 2 describes the eleven parameters selected as the potential drivers of droughts. The used values correspond to the parameters' average in the analysis period (1987 to 2018). The averages were computed using the SWAT model outputs at each subbasin. We used the dominant category at each subbasin for the curve number, the slope, and the soil type (categorical variables).*

14. P10L2242-243: I am wondering why the authors use the total number of months for each drought category and not monthly. By doing this then the response variables are only 1 total number of SM drought month and 1 total number of streamflow drought month? I thought the input variables for both explanatory and response are monthly data or at least yearly data.

This comment makes us realize that the introduction and the methodology could be more precise, which is an excellent opportunity to improve the paper. First, please note that this study's objective is to evaluate the relationship between droughts' drivers and the severity of agricultural and hydrological droughts. Generally, severity is divided into different categories (e.g. moderate, severe, extreme), providing a qualitative assessment of the drought state in a region during a given period. Drought categories are crucial for tracking or anticipating drought-related damage and impacts. The MVRT approach is applied to assess the relationship between

hydroclimatic characteristics (represented by the simulated parameters at each subbasin) and droughts severity categories (represented by the observed number of months for each drought severity category at each subbasin). Regarding the reviewer's question, the response variables (in this case, the drought severity) were not aggregated in one category because the MVRT allowed us to evaluate the relationship between the hydroclimatic parameters and the severity of the drought while maintaining its individual categories. In our study, the explanatory variables are the hydroclimatic parameters at each subbasin, represented by the average value of each parameter during the analysis period (1987 to 2018). The months observed at each drought severity category (The drought indices give categories) are the response variables.

Therefore, we will update the RM in two sections:

- The introduction, as shown in response to Specific Comment 12.

- The methodology, improving the description of the sets of explanatory and response variables.

**Set of explanatory variables**

*To select the set of explanatory variables, we used the outcomes of previous studies on governing drivers of droughts (Sheffield & Wood, 2011a; Zhang et al., 2022). Table 2 describes the eleven parameters selected as the potential drivers of droughts. The used values correspond to the parameters' average in the analysis period (1987 to 2018). The averages were computed using the SWAT model outputs at each subbasin. We used the dominant category at each subbasin for the curve number, the slope, and the soil type (categorical variables).*

**Set of response variables**

*We used the drought analysis outcomes to define the response variables (Table 3). Following the methodology presented in 2.3, we identified the agricultural and hydrological drought events during the analysed period. After identifying the drought events, we counted the months for each drought severity category at each subbasin. The observed months for each one of the three drought categories were used as response variables. The analyses for agricultural and hydrological droughts were conducted separately; thus, two sets of response variables were obtained.*

15. P11L251: What are these two groups?

The new sentence will be clearer:

*The data partitioning consisted of three steps. First, for each explanatory variable were generated all possible partitions of the sites (subbasins) into two groups.*

16. P13L292: Figure 3. I suggest to write the alphabet a, b, c, and so on at the top of the figure. Moreover, please use different colors for observed and simulated for better visibility.

Agreed.

[Figure]

*Figure 3 Monthly calibration and validation for streamflow at: a) Puente Salguero, b) Puente Carretera, c) Cantaclaro and d) Puente Canoas.*

17. P13L296: The authors may re-write the sentence into "……of the parameters, which are the curve number, slope, and soil type at…..

We will edit paragraph in the following way:

*Figure 4 a to h presents the average value of the numerical hydroclimatic drivers of droughts at each subbasin. The average was calculated using the hydrological model's outputs during the simulation period (1987 to 2018). Figure 4 i to k presents the categorical drivers: the curve number, slope and soil type. The dominant category at each subbasin is shown in Figure 4 i to k. The dataset of explanatory variables was created from the values presented in Figure 4.*

18. P14L300: Figure 4. Please describe the soil types. What is soil type a, b, c, and d? I could not find it

    everywhere.

We regret this mistake. The soils type definition and the corresponding reference will be included in the Table 2.

*Table 1. Explanatory variables used in MVRT*

| Hydroclimatic parameter | Abbreviation | Unit | Definition |
|---|---|---|---|
| Precipitation | PRECP | mm | Average precipitation at each subbasin |
| Potential evapotranspiration | PET | mm | Average potential evapotranspiration at each subbasin |
| Evapotranspiration | ET | mm | Average actual evapotranspiration at each subbasin |
| Percolation | PERC | mm | Average percolation past the root zone |
| Surface runoff | SURFQ | mm | Average surface contribution to the streamflow at each subbasin |
| Groundwater | GRWQ | mm | Average groundwater contribution to the streamflow at each subbasin |
| Water yield | WYLD | mm | Average amount of water that leaves the subbasin and contributes to the streamflow at each subbasin |
| Sediment yield | SYLD | metric tons/ha | Average sediment from the subbasin transported into the reach |
| Curve number | CN | – | Dominant curve number at each subbasin |
| Slope | SLP | – | Dominant slope at each subbasin |
| Hydrologic soil group | STY | – | Dominant hydrologic soil group (A, B, C, and D) at each subbasin. The U.S. Department of Agriculture (USDA) classify soils in four hydrologic groups based on the soil's infiltration characteristics. Properties of each soil type can be found in USDA (2007) |

19. P16L316: PCA. Please see my general comment.

Please see answer to General Comment 2.

20. P17L347-348: Please re-write this sentence: "This leaf contains no instance of severe…" It is unclear what

    do the authors mean with no instance? Also, write Figure 9b after the sentence.

Instance refers to "months". The paragraph will be updated in the RM to prevent the reader's confusion.

> *In this leaf, there are no months in the extreme drought category.*

21. P18L374: Figure 8. Please mention a, b, c, d, and so on are the number of n in each decision tree. Same for

    all figures. The figure caption should be self-explanatory and detailed.

In the RM, Figures 7 and 8 captions will indicate that the tree leaves are named from a to l, and the variable n

refers to the number of subbasins clustered at each terminal group. In the Figures 9 and 10 captions, we will

indicate that the tree leaves are named from a to h, and the variable n refers to the number of subbasins clustered

at each terminal group.

[Figure]

*Figure 4* *MVRT of hydroclimatic drivers of agricultural droughts at the Cesar River basin, and spatial distribution of the subbasins clustered at each leaf. Tree leaves are named from a to l and n indicates the number of subbasins clustered at each leaf.*

[Figure]

*Figure 5* *Number of months in agricultural drought categories (moderate, severe, extreme) at each leaf. Tree leaves are named from a to l.*

[Figure]

*Figure 6* MVRT of hydroclimatic drivers of hydrological drought at the Cesar River basin, and spatial distribution of the subbasins clustered at each leaf. Tree leaves are named from a to h and n indicates the number of subbasins clustered at each leaf.

[Figure]

*Figure 7* Months in agricultural drought categories (moderate, severe, extreme) at each leaf. Tree leaves are named from a to h.

22. P20L411: I see that the MVRT has higher (g) and lower (h) than 0.5 mm.

Indeed, we double-checked the value, and the runoff threshold at the third level of split surface is 0.5 mm. We will improve the explanation in the RM for the tree's leaves, *g* and *h*, as follows:

> *The subbasins in terminal group g experienced the lowest median number of months for all hydrological drought categories (Figure 10g). The selected drought drivers and thresholds indicate that surface runoff contributes to the streamflow, and the amount of water that leaves the subbasins is limited. Both characteristics reduced their exposure to hydrological drought. It can be*

*explained by the subbasins' proximity to the marsh (which acted as a natural control), the low slope in the area (which reduced streamflow velocity) and the presence of water bodies (which collected and stored runoff during the rainy season) may have enhanced the water retention capacity in these areas. The observed moderate exposure of these subbasins fits the results of earlier analyses, which found that wetlands exert significant impacts on the alleviation of hydrological drought severity when direct evaporation from the water body does not significantly reduce water storage (Wu et al., 2023). Thus, the present findings indicate that the water storage capacity of the Zapatosa marsh can compensate for the increased evaporation that occurs during drought events, thereby alleviating hydrological drought severity upstream.*

*The hydrological drought conditions in the subbasins clustered at leaf h were mild, despite water yield values below 29 mm (Figure 10h). Negligible surface runoff values indicated that in leaf h, rainfall is either stored in the soil profile, lost by evapotranspiration or percolates in an area of minimal baseflow contribution to streamflow. This limits the amount of water reaching the streamflow and enhances the severity of hydrological droughts, compared to leaf g.*

23. P22L476: Please mention the selected drivers.

Selected drivers will be included in the RM.

*Conversely, the MVRT also showed that in terminal groups b, c and d, hydrological drought severity was linked to the inefficient partition of precipitation. Selected drivers (precipitation, percolation and curve number representing land use) are widely recognized as predominant drivers of hydrological droughts (Iglesias et al., 2018; Stoelzle et al., 2014; van Lanen et al., 2013; van Loon, 2015).*

24. P23L484: The authors stated "previous studies". Please mention those studies.

We have reformulated the paragraph as follows:

*The present selection of the curve number at the third level of split suggests that hydroclimatic parameters and human activities influence hydrological droughts; however, the influence of both drivers is uneven. This is consistent with previous studies concluding that hydroclimatic parameters are more influential (Jehanzaib et al., 2020; Saidi et al., 2018).*

25. P24L524: What do the authors mean with eleven out of nine potential drivers? Usually 9 out of 11 and not vice versa.

We really regret this mistake. It is 9 out of 11. However, we have removed this paragraph as discussed in the response to General comment 2.

26. P25L555: Here the authors mention other ML techniques. This is the reason I suggest the authors to describe why the MVRT was selected compared to others.

Agreed. See our comprehensive answer to Specific Comment 12

---

## Referee Report (RR1)

**Title:** Multivariate regression trees as an 'explainable machine learning' approach to exploring relationships between hydroclimatic characteristics and agricultural and hydrological drought severity. Case of study Cesar River basin.

**Authors:** Paez-Trujilo et al.

**Recommendation:** minor revision

**Assessment**

I appreciate the authors' careful consideration of my main concerns in their revised manuscript. The changes made have significantly improved the clarity and overall quality of the paper. The chapter 2 is now well structured compared to previous version. For the general comments, I disagree with authors response on SSMI although this is irrelevant with the manuscript goal. I also notice that the revised manuscript was not written carefully. I identified numerous mismatches between figures and their corresponding explanation text and figure references (chapter 3.4). I kindly request the authors to carefully review the revised manuscript. I look forward to seeing the final version of the paper.

**General Comments**

1. Thanks for addressing my comments. However, I would like to raise a concern regarding the explanation provided on the use of the Standardized Soil Moisture Index (SSMI or SSI and ESMI in some papers) that is not correct. The authors stated that 1) SSMI is an agricultural drought index derived from daily satellite data, 2) the index is developed for short-term drought monitoring, and 3) there is no previous assessment of the index performance using simulated SM as the input parameter. All these three arguments are misleading. First, the SSMI is a standardized drought index like SPI (for precipitation), SSI (for streamflow), SGI (for groundwater). It uses monthly data instead of daily data (e.g., Ndehedehe et al., 2016; Carrão et al., 2016; Das et al., 2022). These publications used monthly soil moisture data to derive SSMI. Second, SSMI has been employed not only for drought monitoring but also for drought forecasting in some studies (e.g., AghaKouchak, 2014; Xu et al., 2018). Last, some of the aforementioned publications have indeed utilized models to simulate soil moisture variable used in drought identification (SSMI). The second paragraph is accepted.

2. Thanks for your explanation. The authors may consider to move the PCA analysis in the appendix or supplementary material instead of delete it. PCA analysis is still useful to indicate the variance.

3. OK

**Line by line comments**

L refers to line and P refers to page.

I acknowledge the responses from the authors for line-by-line comments. However, the new colors used in the figures for the revised manuscript can be improved. My remarks and suggestions are underlined below.

**P5L117**: Figure 2. Thanks for changing the colors in the legend. I suggest to further improve the distinction between different land cover types. The colors for pasture, crops, and shrubs have bluish colors. The authors may use color dark green for forest, light gray for pasture, color light green for cropland, and brow for shrubs as commonly used in many land use map, or just see some examples from the published land use map.

**P13L292**: Figure 3. Same suggestion. The use of gray and blue colors in the graph is not contrast. The authors may use blue color for observed and red color for simulated. The vertical line to divide calibration and validation periods can be black.

**New line by line comments**

**P1L1**: Suggestion for title: "Multivariate regression trees as an 'explainable machine learning' approach to explore relationships between hydroclimatic characteristics and agricultural and hydrological drought severity: Case of study Cesar River basin"

**P1L17**: The authors may write "(SWAT)" here.

**P1L18**: Suggested text revision: ….the drought indices namely Soil Moisture….

**P2L59-60**: Rephrase this sentence. It is unclear.

**P3L83**: Mentioned -> The aforementioned research

**P4L98**: Between indices and Soil Moisture Deficit Index, the author my write either "which are" or "i.e.,"

**P10L231**: Instead of "(the drought indices give categories)" -> "(moderate, severe, and extreme)"

**P10L234**: What do the authors mean with four technique attributes are relevant to this study?

**P10L238**: The authors may remove "The drought indicators give these three categories to represent the drought severity". It is redundant.

**P12L274**: What is SS?

**P13L310**: Please give a low flow definition here. How do the authors identify low flow? Is it using a threshold method?

**P15L324**: Figure 4. It is annual average right? Also, readers need an explanation about soil type A, B, C, and D. What are those?

**P15L328**: What drought category is represented in Table 6? Is it severe drought, extreme, or moderate drought?

**Starting from here, please read carefully and do comprehensive check.**

**P17L357**: Here the authors say: potential evapotranspiration (1,679 mm). However, I cannot see this number in Figure 7.

**P18L369**: I think it is lower and not above.

**P18L372-373**: It is not Figure 8b and also please check your statement about "highest median of months in the severe drought category"

**P18L375,376**: It is not above but lower.

**P18377**: Here the authors stated that moderate drought category was above 20 months and severe category was above 10 motnhs -> but this is not for Figure 8h.

**P20L404**: Check the number 1362 mm, cannot see this number.

**P20L421**: I think it is higher and not lower.

**P20L426**: Check if it is Figure 9f?

**P20L428**: Check if it is 37?

**P21L442**: Why figure 8e? 8d is higher

**P21L444**: Clusters h, i, and j are seen in Figure 8i, j, and k?

**P22L475**: It think it is not groups I and j but h and i.

**P23L479**: This sentence is confusing. Subbasins grouped at leaf i showed in Figure 10b and c?

**P25L544**: were located

**P26L591**: software -> model

**Reference**

Carrão et al.: An empirical standardized soil moisture index for agricultural drought assessment from remotely sensed data, https://doi.org/10.1016/j.jag.2015.06.011, 2016.

Ndehedehe et al.: On the potential of multiple climate variables in assessing the spatio-temporal characteristics of hydrological droughts over the Volta Basin, https://doi.org/10.1016/j.scitotenv.2016.03.004, 2016.

Das et al.: A non-stationary based approach to understand the propagation of meteorological to agricultural droughts, https://doi.org/10.1007/s11269-022-03297-9, 2022.

AghaKoucak, A.: A baseline probabilistic drought forecasting framework using standardized soil moisture index: application to the 2012 United States drought, doi:10.5194/hess-18-2485-2014, 2014.

Xu et al.: Standardized soil moisture index for drought monitoring based on soil moisture active passive observations and 36 years of north American land data assimilation system data: A case study in the Southeast United States, https://doi.org/10.3390/rs10020301, 2018.

---

## Author Response (AR2)

**Response to Reviewer 1**

**Manuscript title:** Multivariate regression trees as an 'explainable machine learning' approach to explore relationships between hydroclimatic characteristics and agricultural and hydrological drought severity: Case of study Cesar River basin.

**Author's general response:**

We would like to express our gratitude for your constructive suggestions and critical analysis in this second round of review. We appreciated the time you dedicate to the paper to ensure the clarity our research. We applied different changes to incorporate the reviewer's suggestions and adequately present the results of our study. In the following, you will find the answers to the general and specific comments. Some of them required a particular action or change in the manuscript. The changes we apply in the Revised Manuscript (RM) are in italics.

**General Comments.**

1.  In my opinion, the definition of "drought severity" and the role of "multivariate" is still unclear for most of the paper. Even after reading the paper multiple times (in two rounds of reviews), the fact that "multivariate" refers to 3 "drought severity" values for each basin representing the total number of months in each drought class becomes clear to me only halfway through the text. Since modelling these quantities is the key goal of the study, I strongly recommend the author to clarify this definition early in the text. As an example, in the abstract "…drought events… and their severity". The use of classes and duration is never mentioned, and the term "events" may be misleading in conveying the message that severity of multiple events are characterized instead of multiple categories. Also, the authors use "moderate", "severe" and "extreme" as label for these classes, but the term "severe" severity is quite confusing, and it make the results and discussion sections difficult to read.

We thank the reviewer for this comment because it gave us the opportunity to clarify our manuscript. The definitions of drought severity and drought severity categories are now introduced early in the paper, and in consequence the abstract and the introduction have been modified. In the revised manuscript the abstract reads now as follows:

> *The Soil Water Assessment Tool (SWAT) is used for hydrological modelling.*
> *Model outputs, soil moisture and streamflow, are used to calculate the drought*
> *indices, namely the Soil Moisture Deficit Index and the Standardized Stream*

*Flow Index. Then, drought indices are utilised to identify the agricultural and hydrological drought events during the analysis period, and the indices categories are employed to describe their severity. From the identified droughts, the number of months for each drought severity category are sum up. Lastly, the Multivariate regression tree technique is applied to assess the relationship between hydroclimatic characteristics (represented by different simulated hydroclimatic parameters) and the severity of agricultural and hydrological droughts (represented by the number months in different drought severity categories).RM Ln 17 to Ln 24.*

*Drought planning also uses research progress on drought characterisation. Using drought indices is a widespread methodology used for drought characterisation. (Zargar et al., 2011). Drought indices are computed numerical representations of drought severity (Hao & Singh, 2015; Keyantash & Dracup, 2002). Severity refers to the drought strength, also described as the deficit degree (Cavus & Aksoy, 2019), soil moisture deficit in the case of agricultural droughts and streamflow deficit in the case of hydrological droughts. Generally, severity is divided into different categories (e.g. moderate, severe, extreme), providing a qualitative assessment of the drought state in a region during a given period. Drought indices (and their categories) are crucial for tracking or anticipating drought-related damage and impacts (WMO & GWP, 2016).* RM Ln 61 to Ln 69.

Regarding the comment on the severity categories, the authors agree with the reviewer that referring to severe severity may be confusing; however, this terminology comes from the indices, SMDI and SSI proposed by Narasimhan & Srinivasan (2005) and Modarres (2007), respectively. The authors named one of the indices' categories "severe", even when the indices measure "drought severity". We do not like this, but these categories are consistently used in drought studies, and we prefer not to modify the terminology.

2.  The results and discussion sections are still lacking in the "explainable ML" component. The results section is mostly a mere description of Figs. 7-10, without any added values beside what can be inferred directly by the readers by looking at the plots. As an example, no connections are made between the leaves extracted by the MVRT and the obtained spatial patter. Is there any explanation on the grouping? Are all the

separations meaningful? Why some quantities are relevant for some basins on not for others? These are some of the insights that the authors should provide, in my opinion, to the readers to support the value of "explainable ML".

We would like to emphasise that Explainable ML refers to data-driven techniques that provide information on the model's decision-making process. We can say that a model is explainable if a human can tell how a model comes to a decision. MVRT is considered an explainable clustering technique since the method's outcome includes the clusters and the parameters used in the decision-making process of creating them. In this study, MVRT explainability presented in Figures 7 and 9 refers to the fact that the parameters (and the values) to create each one of the clusters are explicit. The authors consider that the explainable component of the MVRT was fundamental for establishing the relationship between the areas experiencing severe (and mild) drought and the parameters influencing that condition.

Regarding the description of Figures 7-10 in the results section, we agree with the reviewer that the description of the MVRT branches can be enhanced. Following the positive remark by the reviewer in specific comment 22, the description of Figures 7-10 is improved in the RM (Please see the answer to specific comment 22). We address the specific comments on the "explainable ML component" in the following.

*No connections are made between the leaves extracted by the MVRT and the obtained spatial patter*
Assuming that by "connections", the reviewer refers to potential similarities or differences between the basins clustered at the same leaf, from the MVRT results, we can determine two main similarities: 1) it is possible to conclude that subbasins clustered at the same group have a comparable susceptibility to droughts (agricultural or hydrological), despite these subbasins are distant from each other, 2) it is possible to conclude that the average values of the parameters influencing droughts are similar in the subbasins clustered at the same group, although these subbasins are distant from each other.

Determine other potential similarities or differences that may connect distant subbasins clustered at the same leaf cannot be obtained from MVRT results since the technique is applied explicitly to cluster in the same leaf subbasins with similar susceptibility to droughts.

*Is there any explanation on the grouping?*

The grouping is the numerical result of the MVRT computation (see section Building the MVRT). Please note that MVRT is a data-driven technique, and building the tree relies entirely on generating multiple partitions of the data, calculating the standard deviation for each group created, and iteratively retaining the groups with the lowest standard deviation. The groups with low standard deviation are those that cluster subbasins with similar susceptibility to droughts (agricultural or hydrological).

*Are all the separations meaningful?*

From a numerical point of view, all the separations are meaningful since each retained leaf contributes to reducing the tree's relative error; in other words, the variance of the data explained by the tree is influenced by the number of separations. From the physical point of view, all the separations provide information on the drought generation process in the subbasins groped at each leaf. All the generated separations allow the discovery of possible unknown links between hydroclimatic parameters and drought severity.

*Why some quantities are relevant for some basins on not for others?*

From a physical point of view some parameter are relevant for some subbasins and not for others because the parameters influencing drought severity at each group of clusters are different. From a numerical point of view, some parameters are relevant for some subbasins and not for others because the explanatory variables that produce the partitions (groups) with the low standard deviation are different for each branch and leaf.

**Specific Comments**

1. L21. Clarify here what is actually modelled (number of months in multiple drought severity classes).

We like the reviewer suggestion. The paragraph is updated in the RM.

> *Then, drought indices are utilised to identify the agricultural and hydrological drought events during the analysis period, and the indices categories are employed to describe their severity. From the identified droughts, the number of months for each drought severity category are summed up. Lastly, the Multivariate regression tree technique is applied to assess the relationship between hydroclimatic characteristics (represented by different simulated hydroclimatic parameters) and the severity of agricultural and hydrological droughts (represented by the number of months in different drought severity categories) .RM Ln 19 to Ln 24.*

2. L31. The interplay.

The authors apologize for the mistake. The error is corrected in the RM.

> *Results show that the presented methodology, combining hydrological modelling and a machine learning tool, provides valuable information about the interplay between the hydroclimatic factors that influence drought severity in the Cesar River basin.* RM Ln 33 to Ln 35.

3. L36. United States department of Agriculture?

The authors apologize for the mistake. The error is corrected in the RM.

> *Upcoming soil moisture drought scenarios predict statically significant, large-scale drying, especially in scenarios with strong radiative forcing in Central America and tropical South America (Lu et al., 2019).* RM Ln 38 to Ln 39.

4. L37. Drought severity, which is expected to…

We like the reviewer suggestion. The sentence is updated in the RM.

> *A similar trend is predicted for hydrological drought severity, which is expected to increase by the end of the twenty-first century, with regional hotspots in central and western Europe and South America, where the frequency of hydrological drought may increase by more than 20 % (Prudhomme et al., 2014).* RM Ln 39 to Ln 42.

5. L75-77. Here you should clarify why drought severity is different and why a multi variable approach is needed.

The authors thank the reviewer for suggesting that the importance of the methodology should be presented in this paragraph. For this purpose, we updated the following paragraphs in the RM.

> *We have found two studies employing machine learning to assess the non-linear relationship between climate and basin processes and droughts (Konapala & Mishra, 2020; Valiya Veettil & Mishra, 2020). The studies reported relevant findings on the parameters driving droughts; however, the selected techniques showed a limitation for the drought analysis since they allow only one output variable. In both cases, it was necessary to apply the chosen technique multiple times to find the relationships between hydroclimatic parameters and the different categories of the evaluated drought characteristics. For example,*

*Valiya Veettil et al. (2020) used a classification and regression tree (CART) to identify the variables influencing drought duration. CART allows one output variable; then, the authors applied the approach three times to evaluate the variables influencing short-term, medium-term and long-term drought events. Meanwhile, Konapala et al. (2020) used a random forest (RF) algorithm to identify the climate and basin parameters influencing the characteristics (duration, frequency and intensity) of three different drought regimes (long duration and mild intensity, moderate duration and intensity, short duration and high intensity). As the core of RF is a decision tree that allows one output variable (in this case, each characteristic of each drought regime), the authors repeated the procedure nine times, one for each drought regime and characteristic.*

*The aforementioned research shows the potential of machine learning techniques for drought-related analysis; nevertheless, it also suggests that assessing the parameters driving drought characteristics requires techniques capable of simultaneously handling the different categories of drought characteristics. Commonly used in ecology to relate independent environmental conditions to populations of multiple species, the Multivariate Regression Tree (MVRT) arises as a suitable technique for this purpose. .RM Ln 76 to Ln 98.*

6.  L85. You introduce here "individual categories of drought severity" but never clarify the concept.

The definition of drought severity is presented in the RM in Ln 61 to Ln 67. To address the reviewer's comment, we enhanced the definition of severity in the RM.

*Drought planning also uses research progress on drought characterisation. Using drought indices is a widespread methodology used for drought characterisation. (Zargar et al., 2011). Drought indices are computed numerical representations of drought severity (Hao & Singh, 2015; Keyantash & Dracup, 2002). Severity refers to the drought strength, also described as the deficit degree (Cavus & Aksoy, 2019), soil moisture deficit in the case of agricultural droughts and streamflow deficit in the case of hydrological droughts. Generally, severity is divided into different categories (e.g. moderate, severe, extreme), providing a*

*qualitative assessment of the drought state in a region during a given period. Drought indices (and their categories) are crucial for tracking or anticipating drought-related damage and impacts (WMO & GWP, 2016).* RM Ln 61 to Ln 67.

7. L87. Why is it important to highlight "supervised" What is the role of supervision here?

It is important to highlight that MVRT is a *supervised clustering* technique because this type of technique uses explanatory variables and response variables to create clusters. That is to say, it uses two datasets. The dataset of response variables is recursively divided using the set of explanatory variables to build the tree and create the clusters. Another technique, *Unsupervised clustering*, only uses one dataset to create the clusters. That is the reason why is important to highlight that this is a supervised technique.

8. L99. "drought events… describe their severity". This gives the impression that the severity of each drought event is modelled, and that the multiple variables are the severity of multiple events. Please reword.

We improved the description of the response variable in the RM

*SWAT outputs, soil moisture and streamflow are used to calculate the drought indices, i.e., the Soil Moisture Deficit Index (SMDI) and the Standardized Stream Flow Index (SSI). Drought indices are utilised to identify the agricultural and hydrological drought events in the analysis period. Then, we calculate the months for each drought severity category during the observed droughts. Finally, the MVRT approach is applied to assess the relationship between hydroclimatic characteristics (represented by the simulated parameters in each subbasin) and drought severity categories (represented by the total number of months for each drought severity category in each subbasin).* RM Ln 103 to Ln 109.

9. Table 1. Discharge is not an input data, but rather used for calibration. As it is listed here it can be misleading.

The authors agree with the reviewer that the information on the discharge should not be included in Table 1. In the RM, details on the discharge data are presented in the "Model calibration and validation section".

**Table 1.** SWAT model input data

| Data type | Details | Source |
|---|---|---|
| Digital elevation model | $25 \times 25$ m | Dataset ALOS PALSAR L1.0, Cartography 1:25000 Geographic Institute Agustín Codazzi (IGAC), Colombia |

| Data type | Details | Source |
|---|---|---|
| Soil map | 300 × 300 m | Soil profiles Project GEF Magdalena–Cauca VIVE, GEF, BID, Fundación Natura, Colombia |
| Land use map | 25 × 25 m | Land use map Geographic Institute Agustin Codazzi (IGAC), Colombia |
| Rainfall and temperature daily data | Period 1985–2018 (34 years) | Institute of Hydrology, Meteorology and Environmental Studies (IDEAM), Colombia |

> *The model was calibrated from 1985 to 2002 and validated from 2003 to 2018 using the streamflow series from four stream gauges (Figure 1). The source of the discharge data is the Institute of Hydrology, Meteorology and Environmental Studies (IDEAM), Colombia.* RM Ln 177 to Ln 178.

10. L163-164. Please correct the reference.

The authors apologize for the mistake the reference is updated in the RM.

> *Based on expert judgment and the available literature (Arnold et al., 2012; ASABE, 2017), the following SWAT parameters were used in the calibration and validation process.* RM Ln 168.

11. L182. Please report there the reasoning behind the use of SMDI rather than soil moisture anomalies (as stated in the replies to the reviewers).

Agree. The criteria to choose SMDI is included in the RM.

> *The present study used the soil moisture deficit index (SMDI) to analyse agricultural droughts. We chose this index since it was developed to use simulated soil moisture as input parameter, particularly the SWAT simulated soil moisture in the soil profile at each subbasin (Narasimhan & Srinivasan, 2005).* RM Ln 189 to Ln 190.

12. L216-217. The values adopted to define a drought event seem rather arbitrary. Please provide some support to these assumptions.

Agreed. Information about the criteria to define the spatial and temporal minimum thresholds is included in the RM.

> *It is worth highlighting that the minimal extension of a drought is not defined, but it is accepted that droughts typically occur on a large scale (Sheffield & Wood, 2011b). Setting a spatial threshold is a common practice to maintain a minimum drought-affected and prevent identifying isolated areas experiencing*

*dry spells as drought events (Brunner et al., 2021). Regarding the temporal*
*threshold, it was used to avoid including sort periods of water shortage and*
*minor and flash droughts in the analysis. These are events that occur within days*
*or weeks (J. Shah et al., 2022).* RM Ln 225 to Ln 230.

13. L231. The response multiple variables are the number of months observed in the three drought severity categories.

Indeed, the multivariate response is the number of months in the three drought severity categories. We update the sentence in the RM.

*The multivariate response is the number of months observed in the three*
*drought severity categories (moderate, severe and extreme) at each subbasin.*
RM Ln 241 to Ln 243.

14. L254. Outputs or inputs (e.g. precipitation is an input).

The sentence is updated using the word "results" to prevent readers from confusion between the outputs of the hydrological model and the inputs of the MVRT technique.

*The averages were computed using the SWAT model results at each*
*subbasin.* RM Ln 263 to Ln 264.

15. L310. Low flow conditions using SSI are not limited to the dry season (see table 6, with some events during winter months). Why not reporting correlation on SSI values?

We agree with the first part of the statement; drought events occur in the dry and rainy seasons. Regarding the correlation on SSI values, assuming the reviewer refers to the correlation between the streamflow (in the dry and rainy season) and the SSI values, the correlation is high since streamflow is the input variable used to calculate the index. The authors consider that presenting the correlation between these two variables does not provide new insights into the objective of this study.

16. L321. Or inputs. Again, precipitation and potential ET are forcing.

The sentence is updated using the word "results" to prevent readers from confusion between the outputs of the hydrological model and the inputs of the MVRT technique.

*Figure 4a to h presents the multi-annual average of the numerical*
*hydroclimatic drivers of droughts at each subbasin. The average was calculated*
*using the hydrological model's results during the simulation period (1987 to*
*2018).* RM Ln 332 to Ln 334.

17. L330. It would be good to have some more details on the events reported. Is there any info on the severity? Do they align well with your modelled classes? What about spatial patters?

The authors agree with the reviewer that it would be interesting to contrast the results; nevertheless, the authors did not find information about drought severity in the National Study of Water. This study is conducted nationally and covers diverse topics associated with water availability in the country. Regarding droughts, the study focuses on the chronology and the duration of drought events in Colombia. Accordingly, we used the available information to compare the drought periods and their duration.

18. L335. Each subbasin, as represented in Figures 5 and 6 for agricultural and hydrological droughts, respectively.

We like the suggestion. In the RM, we applied the reviewer's suggestion.

> *After identifying the agricultural and hydrological drought events, it was possible to determine the number of months for each drought category in each subbasin, as represented in Figures 5 and 6 for agricultural and hydrological droughts, respectively.* RM Ln 348 to Ln 349.

19. Figure 5. The colour scale is difficult to read, especially for the extreme category.

The color scale of Figures 5 and 6 is improved in the RM.

[Figure]

**Figure 1** Months counted in each agricultural droughts category: a) moderate, b) severe and c) extreme. SMDI was not calculated in the wetland subbasins (i.e. hatched area).

[Figure]

**Figure 2** Months counted in the hydrological droughts category: a) moderate, b) severe and c) extreme.

20. L351. Actual evapotranspiration.

The line is updated in the RM.

> *The MVRT indicated that actual evapotranspiration was a strong driver of agricultural droughts; it appeared three times at different tree levels in the splitting rules.* RM Ln 365 to Ln 366.

21. L360. More can be said on the differences between some classes. As an examples, h, I and j looks very similar in terms of frequencies in the three classes. Are the differences modelled by MVRT justified?

The authors agree with the reviewer that the severity in the subbasins clustered at groups *h, i* and *j* is similar, and the variables influencing the severity in clusters *h* and *i* are the same. Considering the severity is alike for clusters *h* and *i,* it can be asserted that these subbasins could be clustered into the same leaf. The method to merge leaves is "pruning" the tree. Tree pruning combines the leaves starting from the leaves produced at the lower levels of split. If we had pruned the tree to merge leaves *h* and *i*, the leaves at the fifth level of split (*e* and *f*) would have become one; then the leaves at the fourth level of split *c, d, h, i, k* and *l* would have been merged. The criteria to decide what leaves to merge is determining the combination of leaves that produces the group with the lowest standard deviation. The authors highlight that pruning the tree increases the CVRE and compromises the explained variance (EV); that is to say, the tree reduces its explanatory power.

The authors opted for retaining the 12 leaves of the tree because the clusters produced at the fourth and fifth level of split provide relevant information on the links between hydroclimatic parameters and the drought severity categories (e.g. leaves *e* and *f*). We consider that the similarities between the leaves *h* and *i* do not compromise the results of this study. On the contrary, retaining these two clusters allowed the authors to extract relevant information from other clusters that would have disappeared when combining leaves *h* and *i* into one cluster.

We included an observation about this two clusters in the section *4.4 Accuracy of the MVRTs*

> *The high EV (0.81) value reflects the good explanatory power of the tree built for agricultural drought. This confirms that the selected explanatory variables significantly influence the severity of agricultural drought. Nevertheless, two potential disadvantages of the tree are identified. First, clusters h and i are very similar. Drought severity is alike in these leaves, and the parameters influencing droughts are the same. This suggests that these two clusters can be merged into one cluster. Second, leaves b and k cluster two subbasins. Accordingly, the distribution presented in the boxplots must be interpreted cautiously. Neither of these disadvantages compromises the study's main findings; however, further analysis is recommended to determine the size of the tree (number of clusters) that better fits the objective of this study.* RM Ln 585 to Ln 591.

22. L361-367. This description of classes a and b does a fir job at introducing the main drivers. The results for the other classes (as well as the ones for hydrological drought) should follow a similar structure.

The authors agree with the reviewer that there is still room to improve the description of the built trees for agricultural and hydrological droughts. The description is improved in the RM.

For agricultural droughts:

[revised manuscript text omitted]

23. Figure 8. The use of box plots for some leaves is questionable when very few subbasins are included (< 6). Please revise this figure, or highlight the limitations is these cases.

The authors agree that using box plots for groups clustering a small number of subbasins may be debatable. We included an observation about these two clusters in section 4.4 Accuracy of the MVRTs (Please see the answer to specific comment 21).

24. Figures 8 and 10. Differences in term of extreme severity are difficult to judge, due to the low magnitude compared to the other classes. Please consider representing these data in a different way or to use a secondary axis.

We thank the reviewer for pointing this out since it allowed us to notice that for some clusters (7b, h, i, j, k, l) the blox-plot presented "zero" months in the extreme as a significant value. Accordingly, we corrected the error. We did not apply additional changes to the Figures. We consider that the Figures adequately present the necessary information to discuss our results.

[Figure]

*Figure 3* *Number of months in agricultural drought categories (moderate, severe, extreme) at each leaf. Tree leaves are named from a to l.*

[Figure]

*Figure 4* *Months in hydrological drought categories (moderate, severe, extreme) at each leaf. Tree leaves are named from a to h.*

25. Figures 7 and 9. Please add some comments on the spatial patters and also on outliers. Some examples from Fig. 9: what is the difference between basins in class a (in the north) and class e (in the south) in terms of hydrology? Why is an isolated basin included in class a (is it just a numeric problem or is there any hydrological explanation)?

Comments on the spatial allocation of the subbasins grouped at each leaf were included in sections *3.4.1 Drivers of agricultural drought and 3.4.2 Drivers of hydrological drought* (Please see answer to Specific comment 22).

The comments on the subbasins hydrology focus on the parameters influencing droughts at each leaf.

It is important to reiterate that according to the MVRT results, subbasins clustered in the same group have a comparable susceptibility to droughts (agricultural or hydrological) despite these subbasins being distant. In addition, the average values of the parameters influencing droughts are similar in the subbasins clustered in the

same group, although these subbasins are distant. Thus, the clusters' outliers do not result from a numerical error.

26. Figure 10. The caption refers to agricultural drought.

The authors apologize for the mistake. The typo error is corrected in the RM. Please see answer to specific comment 24.

27. Discussion: classes b, c, and d are never mentioned.

Agreed. Clusters b, c and d are mentioned in the RM.

> *Leaves b, c, and d corroborate the significant influence of evapotranspiration on agricultural drought severity. A comparison of clusters a and b, and c and d indicates that the leaves with higher evapotranspiration are more prone to experience severe drought. It is interesting to notice that in clusters c and d, the actual evapotranspiration threshold causes a notable difference in drought severity. While the leaf c, clustering subbasins with actual evapotranspiration below 1064 mm, presents the lowest median of months at severe category at the left branch of the tree, leaf d shows the highest median of months at the same category in the tree.* RM Ln 478 to Ln 483.

28. L446. "…Severity in leaves a, e, f and g was comparable…". Is this true? Are they more similar then the other classes?

Indeed, the agricultural drought severity in leaves a, e, f and g is comparable (more similar if compared with the classes). The authors highlight that the similarities specifically refer to the agricultural drought susceptibility observed in the subbasins clustered at these leaves.

From Figure 8, it is possible to see that subbasins clustered in leaves *a*, *e*, *f* and *g* experienced months in extreme drought category; the median of months for severe drought category is above ten months, and for moderate category, the median is above 20 months in leaves *e*, *f* and *g*. Although the values are not exactly the same, a similar trend is observed. Although the drought situation is comparable, the drivers are different, as presented in tree built for agricultural droughts (Figure 7). Our results show how the influence of different hydroclimatic parameters can lead to comparable drought susceptibility like leaves *a*, *e*, *f* and *g*. This is relevant information for understanding the drought-generating process in the Cesar River Basin.

29. L457. Is this true only for leaf e?

The authors agree with the reviewer that the statement is not only valid for leaf *e*. It is also true for leaves *f* and *g*. The paragraph is updated in the RM.

> *In contrast, the MVRT outcomes suggest that a lack of precipitation is not a primary driver of agricultural drought in the subbasins clustered at leaves e, f and g. Particularly, leaf e grouped the subbasins that experienced the most severe agricultural drought in the analysis period.* RM Ln 490 to Ln 492.

30. L457. "the most severe…" This is one example where the term severe (referring to severity) may be confused with severe as category.

Please see answer to general comment 1.

31. L485. I would avoid the use of "expose" here, as it refers to something different in risk analysis.

To prevent inadequate use of the terminology, the sentence t is updated in the RM

> *The subbasins clustered on the left branch of the tree were prone to hydrological drought (Figure 10a, b, c, d).* RM Ln 519.

Other paragraphs of the paper were the word was used are also corrected.

> *The scattering of the outputs in each leaf allows us to identify the susceptible subbasins to agricultural droughts.* RM Ln 363
>
> *This information allowed us to identify the clusters of subbasins prone to hydrological droughts.* RM Ln 423 to Ln 424.
>
> *Figure 10a shows that the subbasins in this terminal group repeatedly experienced to severe and extreme hydrological drought.* RM Ln 434.
>
> *The left branch of the MVRT clusters the subbasins susceptible to severe agricultural drought (Figure 8a, e, f and g).* RM Ln 470.

32. L563. Conclusions.

The authors apologize for the mistake. The typo error is corrected in the RM.

33. L572. "MVRTs indicate… course of the river". This can be inferred from the severity data (see Fig. 6) even without the need of MVRT. Please rephrase.

The authors agree with the reviewer. We updated the conclusion as follows:

> *The outcomes of the MVRT provide valuable information on the hydroclimatic parameters influencing the drought-generating process in the Cesar River basin. MVRTs indicate that severe agricultural and hydrological*

*drought conditions observed in the upper and middle course of the river are influenced by different hydroclimatic factors. The interaction between precipitation shortfalls and high potential evapotranspiration drives severe agricultural drought in the headwater. Conversely, severe hydrological drought condition is mostly caused by limited precipitation. In subbasins in the middle course, droughts' severity is linked to inadequate rainfall partitioning and an unbalanced water cycle favouring water loss through evapotranspiration and low percolation values. Notably, results suggest that poor soil structure enhances severe agricultural drought conditions, and high curve numbers seem to increase hydrological drought severity.* RM Ln 610 to Ln 617.

34. L574. Solely mostly.

We like the suggestion. In the RM, we use "mostly". Please see answer to the minor comment 33.

35. L588. Vulnerable is not the right term here. Also, as mentioned above, the area with most frequent drought can be inferred even without MVRT. Please rephase and emphasise the added value of MVRT is detecting the drivers.

The authors agree with the reviewer. The conclusion is updated as follows:

*It can also be concluded that the MVRT (and other machine learning techniques that generate 'explainable AI' models based on progressive tree-like data partitioning and simplified models in leaves) is a relevant tool for defining drought management strategies. The tool helps to identify drought-prone areas and design management strategies that contribute to maintaining the hydrological parameters influencing droughts above (or below) the thresholds that trigger severe and extreme drought conditions.* RM Ln 625 to Ln 629.

**Manuscript title:** Multivariate regression trees as an 'explainable machine learning' approach to exploring relationships between hydroclimatic characteristics and agricultural and hydrological drought severity

**Author's general response:**

The authors would like to thank the reviewer for thoroughly reviewing the manuscript. We are particularly grateful for the time dedicated to reviewing the Results section and finding many typo errors that were compromising the description of the outcomes of our analysis. We have considerably improved the trees' description and corrected the errors identified by the reviewer.

**General Comments.**

1.  Thanks for addressing my comments. However, I would like to raise a concern regarding the explanation provided on the use of the Standardized Soil Moisture Index (SSMI or SSI and ESMI in some papers) that is not correct. The authors stated that 1) SSMI is an agricultural drought index derived from daily satellite data, 2) the index is developed for short-term drought monitoring, and 3) there is no previous assessment of the index performance using simulated SM as the input parameter. All these three arguments are misleading. First, the SSMI is a standardized drought index like SPI (for precipitation), SSI (for streamflow), SGI (for groundwater). It uses monthly data instead of daily data (e.g., Ndehedehe et al., 2016; Carrão et al., 2016; Das et al., 2022). These publications used monthly soil moisture data to derive SSMI. Second, SSMI has been employed not only for drought monitoring but also for drought forecasting in some studies (e.g., AghaKouchak, 2014; Xu et al., 2018). Last, some of the aforementioned publications have indeed utilized models to simulate soil moisture variable used in drought identification (SSMI). The second paragraph is accepted.

The authors thank the reviewer for providing additional insights about the SSMI. We reviewed the references and realised some studies had used simulated soil moisture to compute the SSMI. We apologise for not conducting more exhaustive research on the SSMI applications and computation principle. However, we consider that SMDI is still a good choice since the authors of SMDI developed the index to use simulated soil moisture as input parameter. The good results for analysing the hydroclimatic drivers of agricultural droughts confirm that the index was appropriate for representing the severity of agricultural droughts.

2. Thanks for your explanation. The authors may consider to move the PCA analysis in the appendix or supplementary material instead of delete it. PCA analysis is still useful to indicate the variance.

The authors appreciate the reviewer's suggestion. However, we prefer to leave the results of the PCA analysis out. The authors consider that for this study, the most significant variance is the proportion of explained variance (EV) calculated for each tree. This parameter defines the explanatory power of the tree and is presented in the *Results* section and discussed in the section *Accuracy of the MVRT*.

**New line by line comments.**

1. P1L1: Suggestion for title: "Multivariate regression trees as an 'explainable machine learning' approach to explore relationships between hydroclimatic characteristics and agricultural and hydrological drought severity: Case of study Cesar River basin".

We like the suggestion. The title is updated accordingly in the RM.

2. P1L17: The authors may write "(SWAT)" here.

The abbreviation is included in the RM.

> *The Soil Water Assessment Tool (SWAT) is used for hydrological modelling.* RM Ln 17.

3. P1L18: Suggested text revision: ….the drought indices namely Soil Moisture….

We like the suggestion. The abstract is updated accordingly in the RM.

> *Model outputs, soil moisture and streamflow are used to calculate the drought indices, namely Soil Moisture Deficit Index and the Standardized Stream Flow Index.* RM Ln 17 to Ln 19.

4. P2L59-60: Rephrase this sentence. It is unclear.

The sentence is rephrased in the RM.

> *Using drought indices is a widespread methodology for drought characterisation.* RM Ln 61 to Ln 62.

5. P3L83: Mentioned -> The aforementioned research

The sentence is updated in the RM.

> *The aforementioned research shows the potential of machine learning techniques for drought-related analysis; nevertheless, it also suggests that assessing the parameters driving drought characteristics requires techniques*

*capable of simultaneously handling the different categories of drought characteristics.* RM Ln 89 to Ln 91.

6. P4L98: Between indices and Soil Moisture Deficit Index, the author my write either "which are" or "i.e.,"

We like the suggestion. The sentence is improved in the RM.

> *The Second is the analysis of droughts. SWAT outputs, soil moisture and streamflow are used to calculate the drought indices, i.e., Soil Moisture Deficit Index (SMDI) and the Standardized Stream Flow Index (SSI).* RM Ln 103 to Ln 104.

7. P10L231: Instead of "(the drought indices give categories)" -> "(moderate, severe, and extreme)"

We like the suggestion. The sentence is updated in the RM.

> *The multivariate response is the number of months observed in the three drought severity categories (moderate, severe and extreme) at each subbasin.* RM Ln 241 to Ln 243.

8. P10L234: What do the authors mean with four technique attributes are relevant to this study?

The authors thank the reviewer for the question since it allows us to notice that referring to "four technique attributes" can be misleading. In the sentence, we want to highlight the technique attributes that make it suitable for this study. The sentence is rephrased in the RM.

> *The following MVRT attributes are relevant for this study.* RM Ln 245.

9. P10L238: The authors may remove "The drought indicators give these three categories to represent the drought severity". It is redundant.

Agreed. The sentence is not included in the RM.

> *MVRT's capability to handle multiple outputs allowed us to evaluate the influence of the hydroclimatic parameters on moderate, severe and extreme drought conditions simultaneously (response variables).* RM Ln 247 to Ln 249.

10. P12L274: What is SS?

SS is the within-group sums of squared distances to the group means. To prevent readers confusion the paragraph is updated in the RM.

> *Second, for each partition, it was calculated the resulting sum of within-group sums of squared distances to the group means for the response data*

*(within-group SS). Within-group SS is equivalent to standard deviation.* RM Ln 282 to Ln 283.

11. P13L310: Please give a low flow definition here. How do the authors identify low flow? Is it using a threshold method?

The authors thank the reviewer for the question. We realised that using the term low flow is inaccurate. We did not develop an analysis to determine low flows. The results in Table 5 correspond to the model performance simulating the discharge in the dry season. To adequately describe the analysis developed, the terminology in the paragraph and the caption of Table 5 are updated in the RM.

*Since the study focuses on droughts, the model performance simulating streamflow in the dry season was analysed separately. Performance indicators were calculated for the period corresponding to the basin's dry season (December to March). The intermediate period of precipitation decrease from June to July was also included in this analysis. Table 5 summarises the calibration and validation performance indicators in the dry season. According to the rating guidelines, the model performance simulating streamflow in the dry season is satisfactory (ASABE, 2017).* RM Ln 323 to Ln 327.

*Table 1. SWAT model performance simulating flows in the dry season.*

12. P15L324: Figure 4. It is annual average right? Also, readers need an explanation about soil type A, B, C, and D. What are those?

Figure 4a to h presents the multi-annual average of the numerical hydroclimatic drivers of droughts at each subbasin. In the RM, we indicate that Figure 4 presents the multi-annual average.

Regarding soil types in the Table 2 (last row), we introduce the hydrologic soil groups and we clarify that they refer to soil's infiltration characteristics.

*Figure 4 presents the numerical and categorical hydroclimatic parameters used as potential drivers of droughts. Figure 4a to h presents the multi-annual average of the numerical hydroclimatic drivers of droughts at each subbasin.* RM Ln 332 to Ln 333.

*Table 2. Explanatory variables used in MVRT*

| Hydroclimatic parameter | Abbreviation | Unit | Definition |
|---|---|---|---|
| Precipitation | PRECP | mm | Average precipitation at each subbasin |
| Potential evapotranspiration | PET | mm | Average potential evapotranspiration at each subbasin |
| Evapotranspiration | ET | mm | Average actual evapotranspiration at each subbasin |
| Percolation | PERC | mm | Average percolation past the root zone |
| Surface runoff | SURFQ | mm | Average surface contribution to the streamflow at each subbasin |
| Groundwater | GRWQ | mm | Average groundwater contribution to the streamflow at each subbasin |
| Water yield | WYLD | mm | Average amount of water that leaves the subbasin and contributes to the streamflow at each subbasin |
| Sediment yield | SYLD | metric tons/ha | Average sediment from the subbasin transported into the reach |
| Curve number | CN | – | Dominant curve number at each subbasin |
| Slope | SLP | – | Dominant slope at each subbasin |
| Hydrologic soil group | STY | – | Dominant hydrologic soil group (A, B, C, and D) at each subbasin. The soil hydrologic groups refer to the soil's infiltration characteristics. Properties of each soil type can be found in USDA (2007) |

13. P15L328: What drought category is represented in Table 6? Is it severe drought, extreme, or moderate drought?

Table 6 does not present information on drought severity. Severity changes every month in each subbasin, and the table cannot show such variability. Figures 4 and 5 are used to present the severity variability. These figures present the total number of months observed for each drought category. Using that information, we build the dataset of explanatory variables to apply the MVRT technique.

**Starting from here, please read carefully and do comprehensive check.**

14. P17L357: Here the authors say: potential evapotranspiration (1,679 mm). However, I cannot see this number in Figure 7.

The authors regret for this mistake, potential evapotranspiration was not used at the fifth level of split. The sentence was *corrected* in the RM.

> Then, the left branch was recursively split as follows: at the third level, according to potential evapotranspiration (1,888 mm) and evapotranspiration (1,191 mm); at the fourth level, according to evapotranspiration (1,064 mm) and percolation (111 mm); and at the fifth level, according to sediment yield (101 tons/ha). RM Ln 367 to Ln 370.

15. P18L369: I think it is lower and not above.

The authors regret for this mistake, as the reviewer indicates the correct word is below not above. The paragraph is updated in the RM.

*For subbasins with actual evapotranspiration below the threshold, leaf c, the median of months in the severe drought category is below ten (Figure 8c).. RM Ln 386 to Ln 387.*

16. P18L372-373: It is not Figure 8b and also please check your statement about "highest median of months in the severe drought category"

The authors regret for this mistake, as the reviewer indicates referring to Figure 8b is incorrect. The statement is also corrected in the RM.

*For subbasins with actual evapotranspiration below the threshold, leaf c, the median of months in the severe drought category is below ten (Figure 8c). For subbasins with actual evapotranspiration above the threshold, leaf d, the median of months in the severe drought category is sixteen, one of the highest among the terminal groups (Figure 8d). RM Ln 386 to Ln 387.*

P18L375,376: It is not above but lower.

The authors thank the reviewer for this comment since we realized that there is room to improve the description of the leaves *e*, *f* and *g*. The paragraph is updated in the RM.

*Leaves e, f and g cluster twenty-four, six and twelve subbasins, respectively. Subbasins are located in the river valley and the basin's western part. In these subbasins, precipitation was below the basin average (1318 mm), and actual evapotranspiration was above the average (1191 mm). The percolation threshold to split leaves e and f from leaf g is 111 mm, a value considerably below the basin average. At the fifth level of split, the sediment yield threshold to split leaves e and f is 101 metric tons/ha, a value close to the average value in the basin. Figures 8e, f and g show that subbasins clustered in these leaves are prone to agricultural droughts. The median of months in the moderate drought category was above twenty months; the severe category was above ten months; and the three leaves exhibited months in the extreme drought category. RM Ln 382 to Ln 388.*

17. P18377: Here the authors stated that moderate drought category was above 20 months and severe category was above 10 motnhs -> but this is not for Figure 8h.

The authors regret for this mistake, in this paragraph we are describing leaves *e*, *f* and *g*. Referring to Figure 8h was a mistake. The paragraph is updated in the RM (Please see answer to specific comment 17).

18. P20L404: Check the number 1362 mm, cannot see this number.

The authors regret for this typo error. The correct number is 1632 mm. The error is corrected in the RM.

> *The subbasins were separated at the first split level according to precipitation (1632 mm).* RM Ln 426.

19. P20L421: I think it is higher and not lower.

The authors regret for this typo error. As the reviewer indicates, the correct word is higher. The paragraph is updated in the RM.

> *In leaves e (n = 72) and f (n = 18), precipitation and water yield exceeded the basin average. The actual evapotranspiration threshold to split leaves e and f is 833 mm, value below the basin average. Both terminal groups describe moderate exposure to hydrological drought. At leaf e, the median of months in the severe and extreme drought categories is below five, while the median of months in the moderate drought category is twenty (Figure 10e). The hydrological drought exposure of the subbasins clustered at leaf f is also mild. In these subbasins, actual evapotranspiration is above the threshold and close to the basin average. These subbasins present the lowest median of months for all drought categories (Figure 10f). Notably, the Zapatosa marsh and upstream subbasins are clustered in this terminal group (Figure 9f).* RM Ln XX to Ln XX.

20. P20L426: Check if it is Figure 9f?

The authors double-check and the Zapatosa marsh is clustered at leaf *f*; then referring to Figure 9f is correct. Please see answer to specific comment 19).

21. P20L428: Check if it is 37?

The authors regret for this typo error. As the reviewer indicates, the correct number is forty. The paragraph is updated in the RM.

> *Leaves g and h cluster seventy-one and forty subbasins, respectively.* RM Ln 456.

22. P21L442: Why figure 8e? 8d is higher

Reviewer's comment allowed us to notice that group *d* also experience a considerable number of months in severe drought category, accordingly the leaf *d* was included in the statement. Leaf *e* is also included in the sentence since subbasins clustered at that group experience a higher number of months in extreme drought category compared with leaf *d*.

23. P21L444: Clusters h, i, and j are seen in Figure 8i, j, and k?

The authors regret for this typo error. As the reviewer noticed, there is an error in the figure we are referring to. The error is corrected in the RM.

> *The subbasins in leaves h, i and j predominately experienced months in the moderate drought category (Figure 8h, i, and j).*RM Ln 471 to Ln 472.

24. P22L475: It think it is not groups I and j but h and i.

The authors regret for this typo error. As the reviewer indicates, the correct leaf is "*h*".

> *At terminal groups h and i, water yield was found to influence the severity of agricultural drought.* RM Ln 508 to Ln 509.

25. P23L479: This sentence is confusing. Subbasins grouped at leaf i showed in Figure 10b and c?

We double-checked the statement, and it is correct. The sentence indicates that subbasins in leaf i show low susceptibility to agricultural droughts but high exposure to hydrological droughts. We mention Figures 10b and c because they present the subbasins experiencing hydrological drought susceptibility.

26. P25L544: were located

The authors apologize for this mistake, the paragraph is updated in the RM.

> *Overall, these subbasins were located in the southern part of the basin.* RM Ln 578 to Ln 579.

27. P26L591: software -> model

The paragraph is updated in the RM.

> *The study's limitations include its simplified approach to modelling a complex phenomenon using SWAT model (e.g. representing the groundwater components that impact hydrological drought conditions) and using only a single ML technique to build explainable models.* RM Ln 631 to Ln 633.

---

## Author Response (AR3)

**Manuscript title:** Multivariate regression trees as an 'explainable machine learning' approach to explore relationships between hydroclimatic characteristics and agricultural and hydrological drought severity: Case of study Cesar River basin.

**Author's general response:**

We want to express our gratitude to the reviewers for their contributions to the review process of our paper. Your revision and constructive criticism have significantly improved the quality of our research. We appreciate the time and effort you invested in providing recommendations that have strengthened the clarity of our findings. In the following, you will find the answers to the general and specific comments. Some of them required a particular action or change in the manuscript. The changes we apply in the Revised Manuscript (RM) are in italics.

Response to Reviewer 1

**General Comments.**

1.  I understand that not all the explicit thresholds can be easily explained, and I am not asking for a full explanation of both trees, but an attempt to explains some of the selection is advisable to better clarify the added value of this methodology over other 'not-explainable' ML. Some examples are reported in the detailed comments, but comments on why a certain threshold is meaningful (e.g. what a precipitation of 1600 mm represents for the region compared to the common climatology) may still be added for some of the key thresholds.

The authors thank the reviewer for the remark. We consider that the manuscript provides the necessary information to relate key thresholds to the basin climatology. For example, the study case description indicates the mean annual precipitation in the three climatic regions defined by the basin topography. Additionally, in section *3.2 Hydroclimatic drivers of droughts*, Figure 4 shows the multi-annual average of the numerical hydroclimatic drivers of droughts at each subbasin. Figure 4 presents an overall picture of the basin climatology and allows the reader to identify the areas showing the lowest and the highest values of the hydroclimatic parameters used in the study. In sections *3.4.1 Drivers of agricultural drought* and *3.5.2 Drivers of hydrological drought*, the authors describe the MVRTs and indicate whether the value falls above or below the basin's average for the key thresholds. These two sections were substantially improved in the second round of review by following the reviewer's recommendation on refine the description of the MVRTs and relate the key thresholds to the climatology of the basin. Accordingly, we believe that it is not necessary to include additional values of reference to relate the thresholds to the basin climatology.

2. The second comment is a follow up (clarification) to the previous review on model validation. You are using SSI as a drought stream flow index, so why don't you compare the simulated SSI (from SWAT) with the observed SSI (from stream gauge records)? The comparison between simulated and observed stream flow data during the dry season is not a proxy variable of the performance of your drought index, as many events are outside the dry season (see Table 6). I suggest to revisit this analysis.

Why don't you compare the simulated SSI (from SWAT) with the observed SSI (from stream gauge records)?

The authors clarify that the SWAT model does not simulate the SSI or any other drought indicator. The SSI was calculated by the authors using the results of the SWAT model, particularly the simulated streamflow at each subbasin. Regarding the question, it is not possible to calculate the SSI using gauge records because there is no observed data (gauging stations) at each subbasin. Hence, it is not possible to develop the comparison proposed by the reviewer (comparing the SSI calculated using simulated streamflow with the SSI calculated using gauge records). In the absence of observed streamflow at each subbasin, we consider that using simulated streamflow from the calibrated model is a valid approach to calculate the hydrological drought index at each subbasin.

The comparison between simulated and observed stream flow data during the dry season is not a proxy variable of the performance of your drought index, as many events are outside the dry season

The authors clarify that this comparison does not intend to assess the accuracy of the SSI calculated using simulated data. The comparison between the simulated and the observed streamflow is part of the calibration and validation process to evaluate the model performance for simulating streamflow. Calibration and validation aim to minimise the difference between model simulations and the observed data. This procedure reduces the uncertainty of the model results, and it can be asserted that a model with good performance indicators correctly simulates the basin hydrology.

The performance indicators used in the calibration and validation of the Cesar River Basin indicate that the model adequately simulates streamflow (in the wet and dry seasons, see Table 4 and 5); accordingly, the model results can be utilised to study the basin hydrology or develop further analyses, such as using the simulated streamflow to calculate the SSI.

3. Finally, a careful read of the text is recommended, as there are still many typos in the text (some are highlighted below).

The authors thank the reviewer's suggestion. Accordingly, we conducted an exhaustive revision of the entire manuscript to correct the typo errors in the text.

**Minor comments**

1. L13. lower than normal precipitation

Agreed. The paragraph is updated in the RM.

*The typical drivers of drought events are lower than normal precipitation and/or higher than normal evaporation.* RM Ln 13.

2. L14-15. …influencing droughts.. influences droughts… Redundant. Please, reword.

Agreed. The paragraph is updated in the RM.

*Evaluating the combined effect of the multiple factors influencing droughts requires innovative approaches.* RM Ln 14 to Ln 15

3. L18. Two drought indices.

Agreed. The paragraph is updated in the RM.

*Model outputs, soil moisture and streamflow, are used to calculate two drought indices, namely the Soil Moisture Deficit Index and the Standardized Stream Flow Index.* RM Ln 17 to Ln 18.

4. L20-21. I suggest removing 'From the… Lastly,'

Agreed. The paragraph is updated in the RM.

*Then, drought indices are utilised to identify the agricultural and hydrological drought events during the analysis period, and the indices categories are employed to describe their severity. Finally, the Multivariate regression tree technique is applied to assess the relationship between hydroclimatic characteristics and the severity of agricultural and hydrological droughts.* RM Ln 19 to Ln 21.

5. L22. Not only simulated. Precipitation is not simulated by SWAT. I suggest removing the part in parentheses.

Agreed. Please see answer to specific comment 4.

6. L28 …agricultural drought, whereas only limited precipitation…

Agreed. The paragraph is updated in the RM.

*Precipitation shortfalls and high potential evapotranspiration drive severe agricultural drought, whereas limited precipitation influences severe hydrological drought.* RM Ln 24 to Ln 26.

7.  L31. In my opinion, exposure is not the right term here.

Agreed. The paragraph is updated in the RM.

*Moderate sensitivity to agricultural and hydrological droughts is related to the capacity of the subbasins to retain water, which lowers evapotranspiration losses and promotes percolation.* RM Ln 29 to Ln 30.

8.  L51. Lower than normal precipitation

Agreed. The paragraph is updated in the RM.

*Typically, droughts are triggered by atmospheric circulation and weather systems that combine to cause lower than normal precipitation and/or higher than normal evaporation in a region.* RM Ln 47 to Ln 48.

9.  L53-59. I still found quite strange this early emphasis on such factors, which albeit very important, and not accounted (or play a very minor role) in your study.

The authors agree with the reviewer that not all the drivers of droughts listed play a significant role in this study; nevertheless, it is relevant to mention them since these factors influence droughts in other areas. In the introduction, we aim to provide an overall picture of all the potential drivers of droughts. Previous studies have demonstrated that the listed factors can enhance or alleviate a drought.

10. L103. 's'econd

We apologize for the mistake. The typo error is corrected in the RM.

*The second is the analysis of droughts.* RM Ln 99 to Ln 100.

11. L103. …, soil moisture and stream flow, are…

We apologize for the mistake. The typo error is corrected in the RM.

*SWAT outputs, soil moisture and streamflow, are used to calculate the drought indices, i.e., the Soil Moisture Deficit Index (SMDI) and the Standardized Stream Flow Index (SSI).* RM Ln 100 to Ln 101.

12. L113. Topography (panel a) and land use (panel b).

Agreed. The paragraph is updated in the RM.

*Figure 1 presents the Cesar River basin's location, topography (Fig. 1a) and land use (Fig. 1b).* RM Ln 110.

13. L113. 72º53'W 74º04'W longitude and…

We apologize for the mistake. The typo error is corrected in the RM.

*The basin is located between 72º53'W 74º04'W longitude and 10º52'00'N 7º41'00''N latitude (Colombia).* RM Ln 110 to Ln 111.

14. L123-124. This description does not match. If the rainy season is April-May, how is it possible that the main rainfall events occur in August November. I guess there is a typo somewhere here.

To prevent readers confusion, we clarity that basin's annual rainfall pattern is bimodal.

*The basin's annual rainfall pattern is bimodal. The dry season occurs from December to April, followed by a rainy season from April to May. From June to July, precipitation decreases, and the main rainfall events occur between August and November.* RM Ln 119 to Ln 121.

15. L132. Maybe a reference to Ramsar sites can be useful here for non-experts.

Agreed. The reference is included in the RM.

*The Zapatosa marsh is recognised as one of the most important wetlands in the country, and considering the relevance of this ecosystem, it was declared a Ramsar site in 2018 (Ramsar sites are wetlands of international importance for containing rare or unique wetland types or for their relevance in conserving biological diversity).* RM Ln 129 to Ln 130.

16. L134. Compared to… (pre-industrial period 1850-1900 if this is the case). Please clarify.

Agreed. The paragraph is updated in the RM.

*In addition, climate change projections indicate that by 2070, the basin's temperature may increase by 2.7°C, and precipitation may reduce by 10 % compared to the reference period 1971-2000 (Universidad del Magdalena et al., 2017).* RM Ln 131 to Ln 133.

17. L146. The SWAT… as it seems to me that you are using the term to refer to one particular SWAT model (the USDA one). If you want to leave this as 'A SWAT', then you need to explicitly say that you are using one specific SWAT in the next sentence (i.e. the SWAT model developed by USDA is here used…).

Agreed. The paragraph is updated in the RM.

*The SWAT model with an ArcSWAT extension was used to simulate the hydrological balance of the Cesar River.* RM Ln 144 to Ln 145.

18. L146. To simulate the hydrological balance of the Cesar River basin.

Agreed. Please see answer to specific comment 17.

19.  L149. Groundwater,

Agreed. The paragraph is updated in the RM.

> *The model is designed to simulate the quality and quantity of surface and groundwater and predict the environmental impacts of land management and climate change (Neitsch et al., 2011).* RM Ln 146 to Ln 147.

20.  L155. Is this average area correct? It does not seem to add up to 22,500 km2.

We apologize for the mistake. The error is corrected in the RM.

> *The model was built for the period from 1987 to 2018. The Cesar River basin was divided into 313 subbasins with a median area of 70 km².* RM Ln 152 to Ln 153.

21.  Table 1. since you are using Hargreaves, I am assuming that min and max daily temperature are used. Please clarify.

Agreed. The Table 1 is updated in the RM.

| Data type | Details | Source |
|---|---|---|
| Digital elevation model | 25 × 25 m | Dataset ALOS PALSAR L1.0, Cartography 1:25000 Geographic Institute Agustín Codazzi (IGAC), Colombia |
| Soil map | 300 × 300 m | Soil profiles Project GEF Magdalena–Cauca VIVE, GEF, BID, Fundación Natura, Colombia |
| Land use map | 25 × 25 m | Land use map Geographic Institute Agustin Codazzi (IGAC), Colombia |
| Daily precipitation and daily minimum and maximum temperature | Period 1985–2018 (34 years) | Institute of Hydrology, Meteorology and Environmental Studies (IDEAM), Colombia |

22.  L169. I am not familiar with this tool, but I am assuming that these parameters are calibrated spatially for each HRU or subbasins. Please add a short sentence to clarify how it works.

Agreed. We include a short comment on the model calibration in the RM.

> *We used the SWAT-CUP software package with Sequential Uncertainty Fitting version 2 (SUFI-2) for automatic model calibration and validation. SUFI-2 operates by performing several iterations. The calibration parameters are sampled in each iteration using the Latin hypercube technique against the objective function values (Abbaspour et al., 2018).* RM Ln 161 to Ln 163.

> *In the calibration process, a physically meaningful range is set for each parameter in each iteration. Then, a new parameter value (within the range) is selected and applied at each HRU or subbasin.* RM Ln 171 to Ln 172.

23.  L185. Represents… representing. Please reword.

Agreed. The paragraph is updated in the RM.

*The NSE is a dimensionless indicator ranging from -∞ to 1, with 1 representing a perfect match between the observed and simulated values (Moriasi et al., 2007).* RM Ln 184 to Ln 185.

24. L187. Low (in absolute value). Or closer to zero.

The following sentence was included in the RM.

*The ideal PBIAS is 0, with low-magnitude values indicating accurate model simulation (Moriasi et al., 2007).* RM Ln 186 to Ln 187.

25. L200. If the index is scaled between -4 and 4, how is it possible to have a category -4 or less? Please check and clarify.

The authors deeply regret this mistake. As the reviewer points out, the minimum value that the SMDI can take is -4. It occurs when the monthly soil water available in the soil profile ($SW_j$) corresponds to the long-term minimum soil water available in the soil profile ($minSW_j$). We corrected the mistake in the RM.

*and extreme drought (SMDI -4).* RM Ln 199.

26. L215. To reiterate on the previous comment. Here you correctly say 'mainly' ranges between -2 and 2. Anyway, I would reword as: zero average and unitary standard deviation.

We appreciate the reviewer's suggestion, but we consider that it is not necessary introduce changes into the sentence since it adequately indicates the SSI range.

27. L229. Short

We apologize for the mistake. The typo error is corrected in the RM.

*The temporal threshold, it was used to avoid including short-term droughts (i.e., daily or weekly) in the analysis (Li et al., 2020).* RM Ln 229 to Ln 230.

28. L229. And minor and flash drought… This sentence (and also the next) is awkwardly worded. Also, you should check the actual definition of flash drought, which is not fitting what you are saying here (quickly developing not shortly lasting).

The reference to flash droughts is not included in the RM to avoid the readers' confusion, and the sentence is updated as presented in the answer to specific comment 27.

29. L279. As there are already many sub-sub-sub-sections, I suggest merging the next two (constrained…, and cross-validation) with the previous one (building the MVRT) and to reduce the text. You are here describing standard procedures, that are not specific of your study.

We like the reviewer's suggestion. The sections are merged in the RM. Regarding the description of the methodology, the authors consider that the description of the methodology is appropriate and presents the main elements of the technique applied. Accordingly, we do not introduce changes in the text.

> ***Building the MVRT: Constrained partitioning of the data and cross-validation***
>
> *Building the MVRT consisted of two processes: (1) the constrained partitioning of the data, and (2) the cross-validation of the results. The mvpart package run both processes in parallel. The two procedures are briefly explained below, and a more detailed description can be found in Borcard et al. (2018).* RM Ln 276 to Ln 278.

30. L322. The ability of SSI to correctly capture drought events depends on the accuracy of your SSI compared to the observations, and not the accuracy of modelled stream flow during the dry season (i.e., ability to capture the fluctuations across the entire year vs. ability to reproduce on average the flow during the dry period).

Please see answer to General Comment 2.

31. L358. Figure 7. The sum of the sub-basins classified does not match the total number of sub basins. Is this due to the wetlands? Please clarify. This is not the case for stream flow drought.

The reviewer's observation is correct. The wetland subbasins are not included in the agricultural drought analysis, which is why the sum is not equal to 313 subbasins. In section 2.2.2 Agricultural and hydrological drought analysis, we indicate: "*SMDI was not calculated for the subbasins that correspond to the Zapatosa marsh. In these subbasins, the predominant land cover is water. See Figure 5*" (RM Ln 211 to Ln 212). Additionally, in Figure 5's caption, the authors clarify that: "*SMDI was not calculated in the wetland subbasins (i.e. hatched area)*". We include the following sentence in Figure 7's caption.

> *Figure 7 MVRT of hydroclimatic drivers of agricultural droughts at the Cesar River basin and spatial distribution of the subbasins clustered at each leaf. Tree leaves are named from a to l, and n indicates the number of subbasins clustered at each leaf. The wetland subbasins are not included in the analysis for agricultural drought.*

32. L478. b, c and d should be in italic. Please check the entire text.

We apologize for the mistake. The error is corrected in the RM.

Leaves *b*, *c*, and *d* corroborate the significant influence of evapotranspiration on agricultural drought severity. RM Ln 484 to Ln 485.

33. L482. How this value compares with the average basin actual ET. Are these subbasins with low or high actual ET?

In the section *3.4.1 Drivers of agricultural drought* we indicate: *The actual evapotranspiration threshold to split leaves c and d is 1,064 mm, value above the basin average (Figure 4c).* RM Ln 388 to Ln 389.

34. L491. Leaves shouldn't be in italic.

We apologize for the mistake. The error is corrected in the RM.

In contrast, the MVRT outcomes suggest that a lack of precipitation is not a primary driver of agricultural drought in the subbasins clustered at leaves *e, f* and *g*. RM Ln 496 to Ln 497.

35. L500. Enhance the characteristics… I suggest rewording this sentence.

The sentence is updated as follows.

*This agrees with earlier findings concluding that soil degradation enhances agricultural drought characteristics (Masroor et al., 2022; Santra & Santra Mitra, 2020; Trnka et al., 2016).* RM Ln 505 to Ln 506.

36. L546-547. This sentence is unclear to me.

The paragrap is updated in the RM.

*The subbasins in terminal group g experienced the lowest median number of months for all hydrological drought categories (Fig. 10g). Low water yield indicates good water retention capacity in these subbasins. It can be explained by the proximity of the subbasins to the marsh (which acted as a natural control), the low slope in the area (which reduced streamflow velocity) and the presence of water bodies (which collected and stored runoff during the rainy season). The runoff threshold indicates that part of rainfall water reaches the streamflow; nevertheless, the subbasins in cluster g have one of the lowest runoff potentials in the basin (Fig. 4e). On the contrary, in these subbasins, percolation is considerably high (Fig. 4d). This seems to confirm that low susceptibility to hydrological droughts is linked to subbasins water retention capacity. The present findings suggest that the water storage capacity of the Zapatosa marsh*

*can compensate for the increased evaporation that occurs during drought events, thereby alleviating hydrological drought severity upstream. Our results concur well with previous analyses concluding that wetlands (located in different climatic regions) significantly alleviate hydrological drought severity when direct evaporation from the water body does not significantly reduce water storage (Wu et al., 2023).* RM Ln 551 to Ln 561.

37. L548. By the proximity of the subbasins to…

The paragraph is updated in the RM. Please see answer to specific comment 36.

38. L551. Is this referring to studies on this basin or studies on wetlands in general? Please clarify.

The paragraph is updated in the RM. Please see answer to specific comment 36.

39. L558. If the runoff is 'negligible', the reliability of SSI as drought index is very limited, as it is a drought index designed for rivers with flow. Please clarify the meaning of 'negligible' and highlight this limitation if needed.

The authors highlight that low runoff values do not necessarily imply streamflow ceasing during dry periods. It is important to keep in mind that runoff and baseflow are the two major components of streamflow. Low runoff values indicate that rainfall water infiltrates, and the baseflow sustains the streamflow between precipitation events. In the area of study, the streams hold water throughout the year; therefore, SSI is an adequate index to analyse hydrological droughts.

40. L571. Drought severity.

We like reviewer's suggestion. The sentence is updated in the RM.

*In this area, drought severity was linked to inadequate rainfall partitioning and an unbalanced water cycle that favours water loss through evapotranspiration and low percolation values (Figs.7d, e, f and g, and Figs. 9b, c and d).* RM Ln 576 to Ln 578.

41. L575. How high CNs increase hydrological drought? Streamflow should be higher for high CN, is this an effect of increased evaporation? Please argument.

We agree with the reviewer that high CN numbers are associated with high runoff potential. High CN values also indicate impervious surfaces and high runoff potential. This means less time is available for the water to percolate and recharge the groundwater. As a result, less water is available for sustaining baseflow in streams, contributing to lower streamflow during dry periods, when droughts are more severe and more likely to occur.

High CN numbers reduce the basin's water retention capacity and are linked to rapid streamflow increase during wet periods (flash floods) and lower streamflow during dry periods.

42. L604. Please add MVRT.

We like reviewer's suggestion. The sentence is updated in the RM.

> *In this study a machine learning technique, namely multivariate regression tree (MVRT), was applied.* RM Ln 609.

43. L607. I would stress that this is particularly true for agricultural drought.

Agreed.

> *Notably, the MVRT built for agricultural drought shows a good explanatory power.* RM Ln 613.

44. L614. In the subbasins in the middle course, drought severity…

We like reviewer's suggestion. The sentence is updated in the RM.

> *In subbasins in the middle course, drought severity is linked to inadequate rainfall partitioning and an unbalanced water cycle favouring water loss through evapotranspiration and low percolation values.* RM Ln 620 to Ln 622

45. L610-624. I suggest to further summarize this section, as many sentences are 1:1 re-proposition of previous paragraphs.

We appreciate the reviewer's suggestion. We consider that the paragraph adequately presents the main findings of this study, and we prefer not to summarise it.

46. L629. I suggest rewording to something like: …hydrological parameters influencing droughts based on threshold values that discriminate between different drought severity conditions.

We appreciate the reviewer's suggestion. The authors consider the sentence clear and prefer not to introduce the proposed change.

**Response to Reviewer 2**

**General comment.**

1. I acknowledge the responses from the authors for line-by-line comments. However, there are still very minor typos found in the revised manuscript. I urge the authors to carefully check it again.

The authors thank the reviewer's suggestion. Following the recommendation, we conducted an exhaustive revision of the entire manuscript to correct the typo errors in the text.

**Specific comments.**

1. P2L38: The authors may remove comma in between words significant and large-scale.

We apologize for the mistake. The error is corrected in the RM.

> *Upcoming soil moisture drought scenarios predict statically significant large-scale drying, especially in scenarios with strong radiative forcing in Central America and tropical South America (Lu et al., 2019).* RM Ln 35 to Ln 36.

2. P8L191-194: Redundant statement about 3 soil profiles. It was already explained before.

The sentence is reworded as follows.

> *SWAT calculates the soil water content of the entire soil profile. Three soil layers were identified in the Cesar River basin. The first layer thickness (vertical distance from the surface) reaches up to 350 mm, the second 1000 mm, and the third 1500 mm.* RM Ln 190 to Ln 192.

3. P10L230: J. Shah et al., 2022 -> Shah et al., 2022.

The reference is not included in the RM.

4. P10L234: Missing spacing for reference De'ath (2022).

We apologize for the mistake. The error is corrected in the RM.

> *MVRT is an extension of the popular regression tree (Breiman, 2001), but it differs in that it allows for multiple outputs (see De'ath (2002)).* RM Ln 232 to Ln 233.

5. P10L235: The authors may choose either technique or approach

We appreciate the reviewer's suggestion. Since technique and approach are synonyms, we prefer using both to prevent using the same word too much.

6. P13L297: Y here should be not capital.

To prevent readers confusion, we update the equation description in the RM.

> *and the denominator is the overall sum of squares of the response data.* RM Ln 298.

7. P13: Table 4: Move the words calibration and validation to the middle.

Agreed. The Table 4 is updated in the RM.

| Gauging station | Calibration | | Validation | |
|---|---|---|---|---|
| | NSE | PBIAS [%] | NSE | PBIAS [%] |
| Puente Salguero | 0.61 | 4.28 | 0.52 | -8.3 |
| Puente Carretera | 0.50 | -5.34 | 0.52 | 7.6 |
| Cantaclaro | 0.58 | -11.30 | 0.50 | -11.7 |
| Puente Canoas | 0.70 | -1.34 | 0.57 | 10.64 |

8. P16: Figure 4. Again please explain in the figure caption, what are soil types A, B, C, and D.

We apologize for the mistake. The caption is updated in the RM.

> *and k) soil type, the soil hydrologic groups A, B, C and D refer to the soil's infiltration characteristics.*RM Ln 340 to Ln 341.

9. P16: Table 6. Please explain if the drought duration here is total from moderate to severe or only for severe?

Agreed.

> *We identified the drought events and estimated their duration following the definition of droughts presented in 2.2.2. A month was summed to the duration of an event when a number of subbasins, covering at least 30 % of the basin's total area, were in a drought state (moderate, severe or extreme).* RM Ln 343 to Ln 345.

10. P18L365: Since it is a new paragraph, please indicate again that you refer to Figure 7.

Agreed.

> *The MVRT indicated that actual evapotranspiration was a strong driver of agricultural droughts; it appeared three times at different tree levels of split (Fig.7).* RM Ln 368 to Ln 369.